# Incremental Aggregated Asynchronous SGD for Arbitrarily Heterogeneous Data

## Abstract

We consider the distributed learning problem with data dispersed across multiple workers under the orchestration of a central server. Asynchronous Stochastic Gradient Descent (SGD) has been widely explored in such a setting to reduce the synchronization overhead associated with parallelization. However, prior works have shown that the performance of asynchronous SGD algorithms depends on a bounded dissimilarity condition among the workers' local data, a condition that can drastically affect their efficiency when the workers' data are highly heterogeneous. To overcome this limitation, we introduce the Incremental Aggregated Asynchronous SGD (IA$^2$SGD) algorithm. With a server-side buffer, IA$^2$SGD makes full use of stale stochastic gradients from all workers to neutralize the adverse effects of data heterogeneity. In an asynchronous implementation setting, the algorithm entails two distinct time lags in the model parameters and data samples utilized in the server's iterations. Furthermore, by adopting an incremental aggregation strategy, IA$^2$SGD maintains a per-iteration computational cost that is on par with traditional asynchronous SGD algorithms. Our analysis demonstrates that IA$^2$SGD achieves a consistent convergence rate for smooth nonconvex problems for arbitrarily heterogeneous data. Numerical experiments indicate that IA$^2$SGD compares favorably with existing asynchronous and synchronous SGD-based algorithms.

## 1 Introduction

In traditional machine learning, training often occurs on a single machine. This approach can be restrictive when handling large datasets or complex models that demand substantial computational resources. Distributed machine learning overcomes this constraint by utilizing multiple machines that work in parallel. This method distributes the computational workload and data across several nodes or workers, enabling faster and more scalable training.

We focus on a common distributed machine learning paradigm, known as the *data parallelism* approach. In this setup, the training data are distributed among multiple workers, with each worker independently conducting computations on its local data. As an extension of stochastic gradient descent (SGD) used on a single machine, *synchronous SGD* (Cotter et al., 2011; Dekel et al., 2012; Chen et al., 2016; Goyal et al., 2017) stands as a prominent example of data-parallel training algorithms. In synchronous SGD, the server broadcasts the latest model to all workers, who then simultaneously compute stochastic gradients using their respective datasets. After local computation, these workers send their stochastic gradients back to the central server. The server then aggregates these stochastic gradients and updates the model accordingly.

However, variations in computation speeds and communication bandwidths across workers, typically due to the differences in hardware, are common. In synchronous SGD, this disparity forces all workers to wait for the slowest one to complete its computations before proceeding to the next iteration. This issue, often referred to as the *straggler effect*, leads to significant idle times, severely limiting the efficiency and scalability of the approach. To address this problem, *asynchronous SGD (ASGD)* algorithms have been extensively studied to mitigate the synchronization overhead among workers. Since nodes operate independently, each can proceed at its own pace without waiting for others. This attribute is especially beneficial in ad-hoc clusters or cloud environments where hardware heterogeneity is prevalent (Assran et al., 2020).

The primary challenge faced by asynchronous training is that its efficiency can be compromised by *data heterogeneity*. This issue arises because fast workers are able to send more frequent updates to the server, while slower workers participate less frequently. Consequently, the training process may become biased, as the data from fast and slow workers are not equally represented in the server's model updates. Recent research efforts have addressed the problem of data heterogeneity in ASGD (Gao et al., 2021; Mishchenko et al., 2022; Koloskova et al., 2022; Islamov et al., 2024). These studies focus on the convergence properties of ASGD

under conditions where the dissimilarity of local objective functions is bounded. However, if the local datasets are highly heterogeneous, leading to significant differences in local objective functions, then the convergence performance of these algorithms can be substantially reduced.

## 1.1 OUR CONTRIBUTIONS

The aim of this paper is to tackle the above limitations of existing ASGD algorithms in handling heterogeneous data. Our main contributions are summarized as follows:

1) We propose the incremental aggregated ASGD (IA$^2$SGD) algorithm for distributed training, with the following key features:

   - IA$^2$SGD handles the data heterogeneity issue through using an aggregated stochastic gradients from *all* workers, which are computed based on both stale models and stale data samples. This leads to a *dual-delayed* aggregated stochastic gradient at the server, contrasting sharply with existing ASGD algorithms that use stale models but fresh data samples for each iteration.
   - IA$^2$SGD can operate in a *fully asynchronous* manner, meaning that the server updates the model as soon as it receives a stochastic gradient from any worker, without the need to wait for other workers.
   - Although IA$^2$SGD requires aggregation of stochastic gradients from all workers in every iteration, it can be implemented *incrementally* by storing each worker's latest aggregated stochastic gradient at a server-side buffer, ensuring a per-iteration computational cost comparable to existing ASGD algorithms.

2) Through a careful analysis accounting for the time lags inherent in the dual-delayed system, we demonstrate that IA$^2$SGD achieves a sublinear convergence rate for general smooth but nonconvex optimization problems under mild assumptions. Our analysis does not depend on bounded function dissimilarity conditions, indicating that IA$^2$SGD can achieve rapid and consistent convergence on arbitrarily heterogeneous data.

3) We perform experiments comparing IA$^2$SGD with other ASGD and aggregation-based algorithms in training deep neural networks on the CIFAR-10 dataset. We show that IA$^2$SGD delivers competitive runtime performance relative to asynchronous and synchronous SGD-based algorithms, validating its effectiveness and efficiency in practical applications.

To our best knowledge, the proposed IA$^2$SGD algorithm is among the first ASGD algorithms with guaranteed exact convergence without the assumption of bounded function dissimilarity.

## 2 PROBLEM SETUP AND PRIOR ART

Consider a distributed machine learning setting involving $n$ workers and a server. Our goal is to tackle the following stochastic optimization problem:

$$\min_{\boldsymbol{w} \in \mathbb{R}^d} F(\boldsymbol{w}) := \frac{1}{n} \sum_{i=1}^{n} F_i(\boldsymbol{w}), \text{ where } F_i(\boldsymbol{w}) := \mathbb{E}_{\boldsymbol{\xi}_i \sim \mathbb{P}_i} \left[ f_i(\boldsymbol{w}; \boldsymbol{\xi}_i) \right]. \tag{1}$$

Here, $d \in \mathbb{Z}_+$ denotes the dimension of the model parameters, and $\boldsymbol{\xi}_i \in \Xi_i$ is a data sample from worker $i$, following a probability distribution $\mathbb{P}_i$ supported on the sample space $\Xi_i$. Each local loss function $f_i(\cdot; \boldsymbol{\xi}_i)$, defined for $i \in [n] := \{1, \dots, n\}$ and $\boldsymbol{\xi}_i \in \Xi_i$, is continuously differentiable and accessible to worker $i$. Problem (1) shall be solved collaboratively by $n$ workers under the coordination of a central server. Our focus is on the heterogeneous data scenarios, where the local data distributions $\mathbb{P}_i$ differ significantly. This setting is particularly relevant in contexts such as data-parallel distributed training (Verbraeken et al., 2020) and horizontal federated learning (Yang et al., 2019).

In vanilla ASGD (Nedić et al., 2001; Agarwal & Duchi, 2011), every computed stochastic gradient at a worker triggers a model update at the server. Given some initial model $\boldsymbol{w}^0 \in \mathbb{R}^d$, this results in the following iterations performed by the server:

$$\boldsymbol{w}^t = \boldsymbol{w}^{t-1} - \eta \nabla f_{j_t}(\boldsymbol{w}^{t-\tau_{j_t}(t)}; \boldsymbol{\xi}_{j_t}^t), \ t = 1, 2, \dots, \tag{2}$$

where $j_t \in [n]$ denotes the index of the worker that contributes to the server's iteration $t$ and $\tau_i(t) \in [1, t]$ represents the delay of the model used to compute the stochastic gradient by worker $i$ at server iteration $t$. The updated model, $\boldsymbol{w}^t$, is then transmitted back to worker $j_t$ for subsequent local computations. It is important to note that $\boldsymbol{\xi}_i^t \sim \mathbb{P}_i$ is indexed by $t$ to indicate that this particular data sample has not been utilized by the server prior to iteration $t$.

The iterative process (2) allows faster workers to participate in the server's model updates more frequently. However, when dealing with data heterogeneity where $F_i$ are different, the stochastic gradient $\nabla f_{j_t}(\boldsymbol{w}^{t-\tau_{j_t}(t)}; \boldsymbol{\xi}_i^t)$ can significantly deviate from $\nabla F(\boldsymbol{w}^t)$ on average, which can impede the model's convergence. To be more specific, we assume that $j_t$ follows some distribution $\{p_1, \ldots, p_n\}$ over $[n]$, where $p_i$ is the probability that $j_t = i$ for $i \in [n]$. To have an intuitive understanding on the effects of data heterogeneity, we consider a hypothetical scenario where the algorithm operates synchronously—that is, $\tau_i(t) = 1$ for all $i \in [n]$. Then, the stochastic gradient remains a *biased estimate* of the exact gradient:

$$\mathbb{E}\left[\nabla f_{j_t}(\boldsymbol{w}^{t-1}; \boldsymbol{\xi}_{j_t}^t)\right] = \sum_{i=1}^n p_i \nabla F_i(\boldsymbol{w}^{t-1}) \neq \nabla F(\boldsymbol{w}^{t-1}).$$

This scenario serves to highlight that even under such a simpler condition, non-uniform participation by workers can lead to biased gradient estimates solely due to data heterogeneity. When we transition to asynchronous operation, as in ASGD, the situation becomes more complex. The inherent delays in ASGD amplify the biases introduced by data heterogeneity. This results in gradient estimates that deviate further from the true gradient, complicating convergence. The convergence analysis of vanilla ASGD on heterogeneous data has been attempted by (Mishchenko et al., 2022). However, vanilla ASGD *may not converge to a stationary point of Problem (1)* and the asymptotic bias is proportional to the level of data heterogeneity, as reported in Table 1.

To address the disparity between fast and slow workers, Koloskova et al. (2022) integrates a random worker scheduling scheme within the ASGD framework. In this approach, after executing iteration (2), the server sends the updated model $\boldsymbol{w}^t$ to a worker sampled from the set of all workers *uniformly at random*. This method promotes more uniform contribution of workers and ensures the convergence of the iterates to a stationary point of Problem (1), achieving the best-known convergence rate for ASGD on heterogeneous data. However, as data heterogeneity increases, the convergence rate is adversely affected, as detailed in Table 1. Additionally, there is a potential issue with this scheduling method: a worker may be chosen multiple times consecutively before it completes its current tasks, leading to a backlog of models in the worker's buffer. This accumulation can reduce the overall efficiency of the algorithm, as workers may struggle to process a queue of pending models. In contrast to strategies that employ uniformly random worker sampling, Leconte et al. (2024a) introduces a non-uniform worker sampling scheme in ASGD to balance the accumulation of queued tasks among both fast and slow workers. The analysis involves specific assumptions about the processing time distributions, which facilitate the accurate determination of the stationary distribution of the number of tasks currently being processed. Additionally, Islamov et al. (2024) has proposed the *Shuffled ASGD*, which shuffles the sampling order of workers after a specified number of iterations. This approach aims to further enhance the fairness and efficiency of task distribution, ensuring that no single worker consistently benefits or suffers from its position in the sampling sequence. Nevertheless, these state-of-the-art ASGD methods all require the dissimilarity among local functions $F_i$ to be bounded. Their performance tends to deteriorate in the presence of high data heterogeneity. Further discussion on other works related to asynchronous training methods can be found in Appendix A.

## 3 INCREMENTAL AGGREGATED ASGD (IA$^2$SGD)

Given the challenges of managing data heterogeneity while ensuring rapid convergence in ASGD, we introduce the Incremental Aggregated ASGD (IA$^2$SGD) method. Our key idea behind the method is to enhance updates by incorporate a *full gradient aggregation* step, which utilizes the stale stochastic gradients from all the $n$ workers. Specifically, given some initial model $\boldsymbol{w}^0 \in \mathbb{R}^d$, the iterative formula of IA$^2$SGD is given by

$$\boldsymbol{w}^t = \boldsymbol{w}^{t-1} - \eta \boldsymbol{g}^t, \ t = 1, 2, \ldots. \tag{3}$$

Importantly, $\boldsymbol{g}^t$ is taken as the *aggregated stochastic gradient* given by

$$\boldsymbol{g}^t := \begin{cases} \dfrac{1}{n}\sum_{i=1}^n \nabla f_i(\boldsymbol{w}^0; \boldsymbol{\xi}_i^1), & t = 1, \\ \dfrac{1}{n}\sum_{i \neq j_t} \nabla f_i(\boldsymbol{w}^{t-\tau_i(t)}; \boldsymbol{\xi}_i^{t-\rho_i(t)}) + \dfrac{1}{n}\nabla f_{j_t}(\boldsymbol{w}^{t-\tau_{j_t}(t)}; \boldsymbol{\xi}_{j_t}^t), & t = 2, 3, \ldots, \end{cases} \tag{4}$$

where $\tau_i(t) \in [1, t]$ and $\rho_i(t) \in [0, t-1]$ are the delays of the models and data samples of worker $i$ in iteration $t$.

We justify that the above recursions can be implemented in the context of asynchronous training. In iteration $t \geq 2$, the server updates the model according to (3) once it receives the stochastic gradient $\nabla f_{j_t}(\boldsymbol{w}^{t-\tau_{j_t}(t)}; \boldsymbol{\xi}_{j_t}^t)$

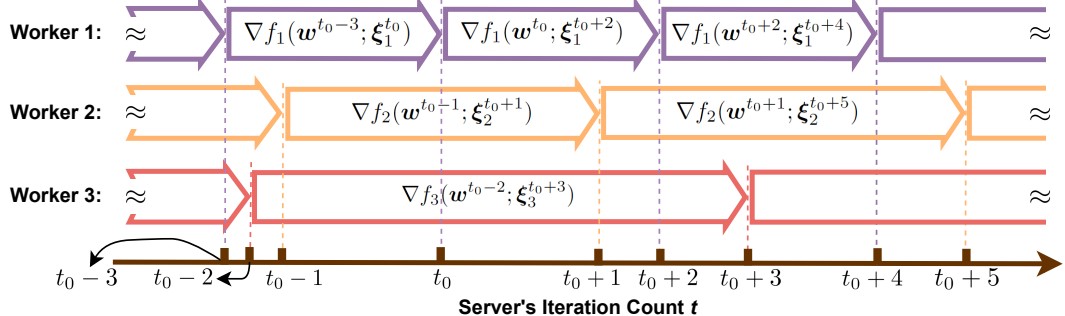

Figure 1: Parallel execution of IA$^2$SGD in a system with 3 workers. The length of each colored arrow box represents the time a worker spends computing a stochastic gradient. To simplify the illustration, we assume that the computation time for each worker is constant throughout the training process. Additionally, the time required for communication between the workers and the server, as well as the server's computation time, are not considered in this visualization. Focusing on the server's iteration $t = t_0 + 4$, we have $j_{t_0+4} = 1$ and the aggregated stochastic gradient used in IA$^2$SGD is expressed as: $g^{t_0+4} = \frac{1}{3}\left(\nabla f_1(w^{t_0+2}; \xi_1^{t_0+4}) + \nabla f_2(w^{t_0-1}; \xi_2^{t_0+1}) + \nabla f_3(w^{t_0-2}; \xi_3^{t_0+3})\right)$. Except for $\xi_1^{t_0+4}$ from worker 1, all the models and data samples utilized within $g^{t_0+4}$ are stale. Besides, the staleness of the model $w$ and data sample $\xi$ used by each worker are different, exhibiting dual delays. In contrast, the vanilla ASGD algorithm would only use the stochastic gradient $\nabla f_1(w^{t_0+2}; \xi_1^{t_0+4})$ from worker 1.

from worker $j_t$—which is called the *participating* worker—without the need to wait for any other workers. For each non-participating worker $i \neq j_t$, the superscript of $\xi_i^{t-\rho_i(t)}$ represents the most recent iteration during which worker $i$ participated in the server's model update before iteration $t$. For each newly computed stochastic gradient $\nabla f_{j_t}(w^{t-\tau_{j_t}(t)}; \xi_{j_t}^t)$, the model used may be outdated, even though the data $\xi_{j_t}^t$ is freshly sampled. This implies that within the aggregated gradient $g^t$, the delay of the model is always larger than that of the data sample by at least 1, i.e., for all $i \in [n]$ and $t \geq 1$,

$$\tau_i(t) \geq \rho_i(t) + 1. \tag{5}$$

After performing iteration (3), the server sends $w^t$ to worker $j_t$ for its subsequent gradient computation using this model.

**Dual-Delay Characterization.** Unlike the ASGD algorithm (2) that uses only the stochastic gradient $\nabla f_{j_t}(w^{t-\tau_{j_t}(t)}; \xi_{j_t}^t)$ of the participating worker $j_t$ in iteration $t$, the IA$^2$SGD method further incorporates the stale stochastic gradients $\nabla f_i(w^{t-\tau_i(t)}; \xi_i^{t-\rho_i(t)})$ of all other workers $i \neq j_t$. In addition, different from (2) where only the model experiences delay while the data sample remains current, the updates of IA$^2$SGD involves two distinct types of delays—in both the models and the data samples—which we call *dual-delayed*. This dual consideration is crucial for rigorously characterizing asynchrony in our system. To clarify, suppose that worker $j_t$ send a stochastic gradient to the server in iteration $t$. We use $\nabla f_{j_t}(w^{t-\tau_{j_t}(t)}; \xi_{j_t}^t)$ to denote this stochastic gradient, because the model $w^{t-\tau_{j_t}(t)}$ was created in a prior iteration $t - \tau_{j_t}(t)$ while the data sample $\xi_{j_t}^t$ is first utilized by the server (the data sample is still fresh at time $t$). As the computation progresses and another worker sends its gradient at iteration $t + 1$, the previous gradient $\nabla f_{j_t}(w^{t-\tau_{j_t}(t)}; \xi_{j_t}^t)$ remains as a component in $g^{t+1}$, rendering both the model and data used by worker $j_t$ *increasingly stale* at time $t + 1$. In Figure 1, we present a simplified algorithmic process of IA$^2$SGD in a system with 3 workers to illustrate the dual-delay property.

**Intuition behind the Effect of Full Aggregation.** Traditional ASGD can lead to a bias in the model towards the data characteristics of faster workers. This is particularly problematic when local data distributions are heterogeneous, as it can skew the convergence of the model and necessitate assumptions about bounded data heterogeneity to control this bias. Our IA$^2$SGD tackles this issue by incorporating outdated stochastic gradients from *all* workers in every iteration, similar to a synchronous aggregation model. In synchronous SGD, the gradient at each iteration is the average of gradients computed concurrently across all workers: $\frac{1}{n}\sum_{i=1}^n \nabla f_i(w^{t-1}; \xi_i^t)$. This full aggregation naturally mitigates the bias caused by data heterogeneity among workers and does not require the assumption of bounded heterogeneity. In IA$^2$SGD, the term $g^t$ serves as an approximation of this synchronous gradient. By ensuring that every worker, regardless of their speed, contributes to the global update within a bounded delay (Assumption 5), we control the approximation error of $g^t$. This controlled approximation

---

**Algorithm 1** Incremental Aggregated Asynchronous SGD (IA$^2$SGD)

---

1: **Input:** $n, T \in \mathbb{Z}_{++}, \eta > 0, \boldsymbol{w}^0 \in \mathbb{R}^d$
2: **Initialization:** For worker $i \in [n]$, it computes $\boldsymbol{G}_i^1 = \nabla f_i(\boldsymbol{w}^0; \boldsymbol{\xi}_i^1)$, stores it in the worker's buffer, and sends it to the server. The server computes $\boldsymbol{g}^1 = \frac{1}{n} \sum_{i=1}^n \boldsymbol{G}_i^1$ and $\boldsymbol{w}^1 = \boldsymbol{w}^0 - \eta \boldsymbol{g}^1$, stores them in the server's buffers, and broadcasts $\boldsymbol{w}^1$ to all workers
3: **for** $t = 2, 3, \ldots, T$ **do**
4:     Once some worker $j_t$ finishes computing $\boldsymbol{G}_{j_t}^t = \nabla f_{j_t}(\boldsymbol{w}^{t-\tau_{j_t}(t)}; \boldsymbol{\xi}_{j_t}^t)$, it sends $\boldsymbol{\delta}^t = \boldsymbol{G}_{j_t}^t - \boldsymbol{G}_{j_t}^{t-1}$ to the server and write $\boldsymbol{G}_{j_t}^t$ into its local buffer
5:     The server computes the aggregated stochastic gradient as $\boldsymbol{g}^t = \boldsymbol{g}^{t-1} + \boldsymbol{\delta}^t/n$ and writes $\boldsymbol{g}^t$ into its gradient buffer
6:     The server computes the new model as $\boldsymbol{w}^t = \boldsymbol{w}^{t-1} - \eta \boldsymbol{g}^t$, sends $\boldsymbol{w}^t$ to worker $j_t$, and writes $\boldsymbol{w}^t$ into its model buffer
7: **end for**
8: **Output:** $\boldsymbol{w}^{r-1}$, where $r$ selected uniformly random from $[T]$

---

allows us to effectively balance between the benefits of asynchrony and the robustness of synchronous updates, improving convergence rates despite the use of potentially outdated gradients.

### 3.1 INCREMENTAL AGGREGATED IMPLEMENTATIONS

We define $\tau_i(1) = 1$ and $\rho_i(1) = 0$ for all $i \in [n]$. Note that $\tau_{j_t}(t) \geq 1$ and $\rho_{j_t}(t) = 0$ for all $t \geq 1$, then the aggregated stochastic gradient $\boldsymbol{g}^t$ defined in (4) can be expressed more compactly as

$$\boldsymbol{g}^t = \frac{1}{n} \sum_{i=1}^n \underbrace{\nabla f_i(\boldsymbol{w}^{t-\tau_i(t)}; \boldsymbol{\xi}_i^{t-\rho_i(t)})}_{=: \boldsymbol{G}_i^t}, \ t = 1, 2, \ldots.$$

The evolution of the delays associated with the data samples is described by:

$$\rho_i(t) = \begin{cases} 0, & \text{if } i = j_t \\ \rho_i(t-1) + 1, & \text{if } i \neq j_t \end{cases}, \ t = 2, 3, \ldots, \tag{6}$$

and the delay $\tau_i(t)$ associated with the model satisfies $\tau_i(t) = \tau_i(t-1) + 1$ for all $i \neq j_t$. Therefore, the aggregated stochastic gradient $\boldsymbol{g}^t$ utilized by IA$^2$SGD can be updated in an *incremental* manner:

$$\boldsymbol{g}^t = \boldsymbol{g}^{t-1} - \frac{1}{n} \underbrace{\nabla f_{j_t}(\boldsymbol{w}^{t-1-\tau_{j_t}(t-1)}; \boldsymbol{\xi}_{j_t}^{t-1-\rho_{j_t}(t-1)})}_{\boldsymbol{G}_{j_t}^{t-1}} + \frac{1}{n} \underbrace{\nabla f_{j_t}(\boldsymbol{w}^{t-\tau_{j_t}(t)}; \boldsymbol{\xi}_{j_t}^t)}_{\boldsymbol{G}_{j_t}^t}, \ t = 2, 3, \ldots, \tag{7}$$

thus we call IA$^2$SGD an *incremental aggregated* method. By letting each worker to maintain a record of the last stochastic gradient $\boldsymbol{G}_i^{t-1}$ and the server to maintain the aggregated stochastic gradient $\boldsymbol{g}^{t-1}$ computed in the last iteration, computing the new $\boldsymbol{g}^t$ can be implemented rather efficiently. Specifically, once worker $j_t$ has completed computing $\boldsymbol{G}_{j_t}^t$, it uploads $\boldsymbol{\delta}^t := \boldsymbol{G}_{j_t}^t - \boldsymbol{G}_{j_t}^{t-1}$ to the server. Then, the server updates the aggregated stochastic gradient as $\boldsymbol{g}^t = \boldsymbol{g}^{t-1} + \boldsymbol{\delta}^t/n$ according to (7). In view of this, the server's per-iteration computational complexity is $\mathcal{O}(d)$ in IA$^2$SGD, which is independent of the number of workers $n$ and thus aligns with that of traditional ASGD algorithms (Mishchenko et al., 2022; Koloskova et al., 2022; Leconte et al., 2024a; Islamov et al., 2024). The overall procedures of IA$^2$SGD are described in Algorithm 1 and a pictorial comparison between traditional ASGD and IA$^2$SGD during a single communication round is illustrated in Figure 2.

**Server-Side Memory Management.** As depicted in Figure 2, compared to traditional ASGD, IA$^2$SGD requires only an additional memory allocation for a $d$-dimensional vector at the server and each worker. The additional $\mathcal{O}(d)$ memory requirement per worker, indeed, does not scale with the number of workers or the mini-batch size, which generally keeps the memory overhead manageable. However, this could still be a limitation in environments where clients have extremely restricted memory capabilities. To address such scenarios, our algorithm design includes provisions for flexible implementation strategies that can accommodate varying memory capacities among the workers. For example, using the system architecture illustrated in Figure 2, we can adopt the following *server-side memory management* strategy: For a memory-restricted worker (e.g., worker 2), we can adapt the system such that the server temporarily stores the previous gradient $\boldsymbol{G}_2^{t-1}$. When worker 2

computes the new gradient $\boldsymbol{G}_2^t$, it only needs to send this new gradient to the server. The server then computes the difference $\boldsymbol{\delta}^t = \boldsymbol{G}_2^t - \boldsymbol{G}_2^{t-1}$, reducing the memory load on worker 2.

**Mini-Batch Variant.** For conciseness of presentation, each worker computes a stochastic gradient using just one data sample at a time in Algorithm 1. However, to better balance the stochastic gradient noise and per-iteration computation/memory costs, it is advantageous to employ multiple samples. With a slight abuse of notation, we define $\nabla f_i(\boldsymbol{w}^{t-\tau_i(t)}; \mathcal{D}_i^{t-\rho_i(t)}) := \frac{1}{b_i} \sum_{k=1}^{b_i} \nabla f_i(\boldsymbol{w}^{t-\tau_i(t)}; \boldsymbol{\xi}_{i,k}^{t-\rho_i(t)})$, where $\mathcal{D}_i$ is a set of data samples independently drawn from $\mathbb{P}_i$ with batch size $b_i \geq 1$. Using $\boldsymbol{g}^t = \frac{1}{n} \sum_{i=1}^n \nabla f_i(\boldsymbol{w}^{t-\tau_i(t)}; \mathcal{D}_i^{t-\rho_i(t)})$ as the aggregated stochastic gradient in iteration (3) yields the *mini-batch* version of IA$^2$SGD.

## 4 THEORETICAL ANALYSIS

This section delves into the theoretical underpinnings of convergence behaviors of IA$^2$SGD. For clarity in our discussion, we present the convergence analysis of Algorithm 1 without mini-batching. The technical results discussed herein can be readily generalized to the mini-batch variant of IA$^2$SGD.

To prepare for the analysis, we introduce the following standard assumptions that are instrumental to our analysis.

**Assumption 1.** *There exists $F^* > -\infty$ such that $F(\boldsymbol{w}) \geq F^*$ for all $\boldsymbol{w} \in \mathbb{R}^d$.*

**Assumption 2.** *$F_i$ is $L$-smooth, i.e., $F_i$ is continuously differentiable and there exists $L \geq 0$ such that*

$$\|\nabla F_i(\boldsymbol{w}) - \nabla F_i(\boldsymbol{w}')\|_2 \leq L \|\boldsymbol{w} - \boldsymbol{w}'\|_2, \ \ \forall \, \boldsymbol{w}, \boldsymbol{w}' \in \mathbb{R}^d.$$

**Assumption 3.** *Let $\boldsymbol{w}^r$ and $\boldsymbol{w}^s$ with $s, r \geq 0$ be iterates generated by Algorithm 1, $\boldsymbol{\xi}_i^t \in \Xi_i$ with $i \in [n]$ and $t \geq 1$ be a data sample drawn from $\mathbb{P}_i$, and $\mathcal{F}_s$ be the sigma algebra generated by $\boldsymbol{w}^1, \ldots, \boldsymbol{w}^s$. If $r \leq s < t$, then*

$$\mathbb{E}\left[\nabla f_i(\boldsymbol{w}^r; \boldsymbol{\xi}_i^t) \mid \mathcal{F}_s\right] = \nabla F_i(\boldsymbol{w}^r). \tag{8}$$

Assumption 3 specifies that the stochastic gradient estimate is unbiased. It is important to note that the iterate $\boldsymbol{w}^r$ may depend on $\boldsymbol{\xi}_i^t$ for different times $r$ and $t$, rendering $\nabla f_i(\boldsymbol{w}^r; \boldsymbol{\xi}_i^t)$ a potentially biased estimate of $\nabla F_i(\boldsymbol{w}^r)$. To maintain the unbiasedness as defined in equation (8), it is critical to ensure that $r \leq s < t$. This condition guarantees that $\mathcal{F}_s$ encompasses all information present in $\boldsymbol{w}^r$ and that $\boldsymbol{\xi}_i^t$ is independent of $\mathcal{F}_s$.

Furthermore, we impose upper bounds on the conditional variance of stochastic gradients:

**Assumption 4.** *Let $\boldsymbol{w}^r$ and $\boldsymbol{w}^s$ with $s, r \geq 0$ be iterates generated by Algorithm 1, $\boldsymbol{\xi}_i^t \in \Xi_i$ with $i \in [n]$ and $t \geq 1$ be a data sample drawn from $\mathbb{P}_i$, and $\mathcal{F}_s$ be the sigma algebra generated by $\boldsymbol{w}^1, \ldots, \boldsymbol{w}^s$. If $r \leq s < t$, then there exists a constant $\sigma \geq 0$ such that*

$$\mathbb{E}\left[\|\nabla f_i(\boldsymbol{w}^r; \boldsymbol{\xi}_i^t) - \nabla F_i(\boldsymbol{w}^r)\|_2^2 \mid \mathcal{F}_s\right] \leq \sigma^2.$$

Lastly, we assume that each worker participates in the server's model updates within a bounded number of iterations, encapsulated by the following assumption regarding the maximum delay of model parameters:

**Assumption 5.** *There exists $\tau_{\max} \geq 1$ such that $\tau_i(t) \leq \tau_{\max}$ for all $i \in [n]$ in Algorithm 1.*

### 4.1 CONVERGENCE ANALYSIS OF IA$^2$SGD

As we focus on the non-convex optimization setting where the loss functions are only assumed to the lower bounded and smooth (cf. Assumption 1, 2), our aim is to analyze the convergence rate of IA$^2$SGD towards a stationary solution of (1), i.e., the number of iterations required to find $\hat{\boldsymbol{w}}$ so that $\mathbb{E}\|\nabla F(\hat{\boldsymbol{w}})\|^2 \leq \epsilon$ for some $\epsilon > 0$. In the following, we outline the analysis idea and highlight on the technical challenges that have to be overcomed.

Our first step is to observe the following descent lemma that holds under Assumption 2, i.e., for any $t \geq 1$,

$$\mathbb{E}[F(\boldsymbol{w}^t)] - \mathbb{E}[F(\boldsymbol{w}^{t-1})] \leq \mathbb{E}[\langle \nabla F(\boldsymbol{w}^{t-1}), \boldsymbol{w}^t - \boldsymbol{w}^{t-1} \rangle] + \frac{L}{2} \mathbb{E}\|\boldsymbol{w}^t - \boldsymbol{w}^{t-1}\|_2^2$$

$$= -\eta \mathbb{E}\langle \nabla F(\boldsymbol{w}^{t-1}), \boldsymbol{g}^t \rangle + \frac{L\eta^2}{2} \mathbb{E}\|\boldsymbol{g}^t\|_2^2. \tag{9}$$

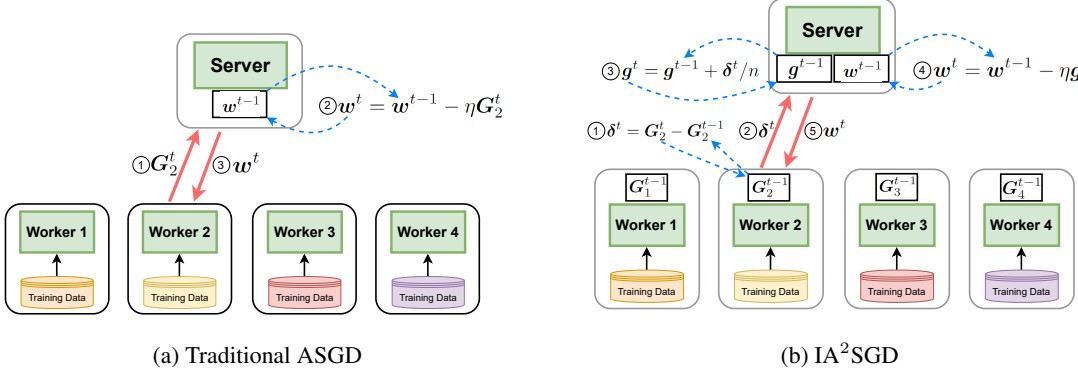

(a) Traditional ASGD  (b) IA$^2$SGD

Figure 2: Comparison of a single communication round of traditional ASGD and IA$^2$SGD. Suppose that worker 2 participates in the server's model update in iteration $t$. In traditional ASGD, each worker directly sends the freshly computed stochastic gradient $\boldsymbol{G}_2^t = \nabla f_2(\boldsymbol{w}^{t-\tau_2(t)}; \boldsymbol{\xi}_2^t)$ to the server. While in IA$^2$SGD, each worker maintains a memory of the most recently evaluated stochastic gradient $\boldsymbol{G}_2^{t-1}$ and sends the gradient difference $\boldsymbol{\delta}^t = \boldsymbol{G}_2^t - \boldsymbol{G}_2^{t-1}$ to the server.

Intuitively, both $\mathbb{E}\langle\nabla F(\boldsymbol{w}^{t-1}), \boldsymbol{g}^t\rangle$ and $\mathbb{E}\|\boldsymbol{g}^t\|_2^2$ can be regarded as biased estimates of $\|\nabla F(\boldsymbol{w}^{t-1})\|_2^2 \geq 0$. Subsequently, selecting an appropriate step size $\eta$ can make the right-hand side of (9) negative, thereby ensuring a sufficient decrease in the expected function value in each iteration.

The above intuition is indeed correct when $\tau_{\max} = 1$, i.e., the algorithm is synchronous and there are no delays. However, due to the dual-delayed property of the information encapsulated in $\boldsymbol{g}^t$, handling the inner product term is not trivial. To describe the challenge, note that

$$\langle\nabla F(\boldsymbol{w}^{t-1}), \boldsymbol{g}^t\rangle = \frac{1}{n}\sum_{i=1}^n \langle\nabla F(\boldsymbol{w}^{t-1}), \nabla f_i(\boldsymbol{w}^{t-\tau_i(t)}; \boldsymbol{\xi}_i^{t-\rho_i(t)})\rangle.$$

According to the iterative formula (3), $\boldsymbol{w}^{t-1}$ is a function of $\boldsymbol{\xi}_i^{t-\rho_i(t)}$ for all $i \in [n]$ so that $t - \rho_i(t) \leq t - 1$. Consequently, we cannot apply conditional expectation to simplify the summand in the above, i.e.,

$$\mathbb{E}\langle\nabla F(\boldsymbol{w}^{t-1}), \nabla f_i(\boldsymbol{w}^{t-\tau_i(t)}; \boldsymbol{\xi}_i^{t-\rho_i(t)})\rangle \neq \mathbb{E}\langle\nabla F(\boldsymbol{w}^{t-1}), \nabla F_i(\boldsymbol{w}^{t-\tau_i(t)})\rangle.$$

Thus we can no longer obtain a simple expression for the expectation of the inner product.[1] To address this challenge, our idea is to decompose the inner product into two terms:

$$\langle\nabla F(\boldsymbol{w}^{t-1}), \boldsymbol{g}^t\rangle = \langle\nabla F(\boldsymbol{w}^{[t-\tau_{\max}]_+}), \boldsymbol{g}^t\rangle + \langle\nabla F(\boldsymbol{w}^{t-1}) - \nabla F(\boldsymbol{w}^{[t-\tau_{\max}]_+}), \boldsymbol{g}^t\rangle, \quad (10)$$

where $[x]_+ := \max\{x, 0\}$ for $x \in \mathbb{R}$. For the first term in the right-hand-side (RHS) of (10), utilizing Assumption 5 and considering the expectation conditioned on the most outdated model, we have the desired property:

$$\mathbb{E}\langle\nabla F(\boldsymbol{w}^{[t-\tau_{\max}]_+}), \boldsymbol{g}^t\rangle = \frac{1}{n}\sum_{i=1}^n \mathbb{E}\langle\nabla F(\boldsymbol{w}^{[t-\tau_{\max}]_+}), \nabla F_i(\boldsymbol{w}^{t-\tau_i(t)})\rangle$$

Then, through carefully controlling the second error term in the RHS of (10), we arrive at a lower bound for the inner product:

**Proposition 1.** *Suppose that Assumptions 2–5 hold. If the stepsize satisfies $\eta \leq \frac{1}{16L\tau_{\max}}$, then it holds for all $t \geq 1$ that*

$$\mathbb{E}\langle\nabla F(\boldsymbol{w}^{t-1}), \boldsymbol{g}^t\rangle \geq \frac{1}{8}\mathbb{E}\|\nabla F(\boldsymbol{w}^{t-1})\|_2^2 - 2L\eta\sum_{s=1+[t-\tau_{\max}]_+}^t \mathbb{E}\|\nabla F(\boldsymbol{w}^{s-1})\|_2^2 - 3L\tau_{\max}\eta\frac{\sigma^2}{n}$$

$$+ \frac{1}{2}\mathbb{E}\left\|\frac{1}{n}\sum_{i=1}^n \nabla F_i(\boldsymbol{w}^{t-\tau_i(t)})\right\|_2^2 - 6L^2\tau_{\max}\eta^2\sum_{s=1+[t-2\tau_{\max}]_+}^t \mathbb{E}\left\|\frac{1}{n}\sum_{j=1}^n \nabla F_j(\boldsymbol{w}^{s-\tau_j(s)})\right\|_2^2,$$

---

[1]We remark that prior studies (Lian et al., 2018; Avdiukhin & Kasiviswanathan, 2021; Zhang et al., 2023; Wang et al., 2023b) involved similar inner product terms in their analysis and have treated them with $\mathbb{E}\langle\nabla F(\boldsymbol{w}^r), \nabla f_i(\boldsymbol{w}^s; \boldsymbol{\xi}_i^t)\rangle = \mathbb{E}\langle\nabla F(\boldsymbol{w}^r), \nabla F_i(\boldsymbol{w}^s)\rangle$ for $s < t$. The latter identity may not hold in the asynchronous setting. This nuanced but crucial issue may not have been adequately emphasized in these works.

The proof of Proposition 1 is deferred to Appendix B.2. Equipped with this, we establish the following convergence bound of IA$^2$SGD:

**Theorem 1.** *Suppose that Assumptions 1–5 hold. Let $\{\boldsymbol{w}^t\}_{t=1}^T$ be the sequence generated by Algorithm 1. If the step size $\eta$ satisfies $\eta \leq \frac{1}{64L\tau_{\max}}$ with $\Delta := F(\boldsymbol{w}^0) - F^*$, then it holds that*

$$\frac{1}{T}\sum_{t=1}^T \mathbb{E}\|\nabla F(\boldsymbol{w}^{t-1})\|_2^2 \leq \frac{32\Delta}{T\eta} + 128L\tau_{\max}\eta\frac{\sigma^2}{n} + 128L^3\tau_{\max}^2\eta^3\frac{\sigma^2}{n}. \tag{11}$$

The proof of Theorem 1 is deferred to Appendix B.3. By property choosing the step size $\eta$ in Theorem 1, we can obtain the specific convergence rate of IA$^2$SGD, as stated in the following Corollary:

**Corollary 1.** *Suppose that Assumptions 1–5 hold. Let $\{\boldsymbol{w}^t\}_{t=1}^T$ be the sequence generated by Algorithm 1 and the step size be $\eta = \frac{1}{2}\sqrt{\frac{n\Delta}{L\tau_{\max}T}}$. Then, for $T = \Omega(L\Delta n\tau_{\max})$, it holds that*

$$\frac{1}{T}\sum_{t=1}^T \mathbb{E}\|\nabla F(\boldsymbol{w}^{t-1})\|_2^2 = \mathcal{O}\left((1+\sigma^2)\sqrt{\frac{L\Delta\tau_{\max}}{nT}} + \frac{L\sigma^2\Delta^{3/2}\sqrt{n\tau_{\max}}}{T^{3/2}}\right).$$

Corollary 1 demonstrates that IA$^2$SGD converges to a stationary point of Problem (1) at a rate of $\mathcal{O}\left((1+\sigma^2)\sqrt{L\Delta\tau_{\max}/(nT)}\right)$ and the *transient time* $T = \Omega(n\tau_{\max})$ required for convergence exhibits moderate *linear* dependence on both the number of workers and the maximum model delay. Critically, the convergence rate of IA$^2$SGD is achieved without imposing any assumptions on upper bounds for data heterogeneity or dissimilarity among individual functions $F_i$. This indicates that IA$^2$SGD is well-suited for distributed environments with highly heterogeneous data. For sufficiently small $\epsilon > 0$, we can deduce that after acquiring $\mathcal{O}\left((1+\sigma^2)^2L\Delta\tau_{\max}/(n\epsilon^2)\right)$ samples, the output of Algorithm 1, $\boldsymbol{w}^{r-1}$, satisfies $\mathbb{E}\|\nabla F(\boldsymbol{w}^{r-1})\|_2^2 = \frac{1}{T}\sum_{t=1}^T \mathbb{E}\|\nabla F(\boldsymbol{w}^{t-1})\|_2^2 \leq \epsilon$.

Additionally, Corollary 1 implies that when $\tau_{\max} = \mathcal{O}(n)$, the dependence on $\tau_{\max}$ in the dominant term of the convergence bound is offset by the speedup factor $1/n$. Consequently, the convergence rate of IA$^2$SGD is $\mathcal{O}\left((1+\sigma^2)\sqrt{\frac{L\Delta}{T}} + \frac{L\sigma^2\Delta^{3/2}n}{T^{3/2}}\right)$.

**Remark 1.** *The step size selection is not unique. A simpler choice can be $\eta = \sqrt{n/T}$, which does not rely on any unknown quantities. However, this yields a rate of $\mathcal{O}\left(\frac{\Delta+L\tau_{\max}\sigma^2}{\sqrt{nT}}\right)$, which is suboptimal in terms of the constant factor. Moreover, the lower bound on $T$ in Corollary 1 can be relaxed by selecting a smaller step size: $\eta = \min\left\{\frac{1}{64L\tau_{\max}}, \frac{1}{2}\left(\frac{n\Delta}{L\sigma^2\tau_{\max}T}\right)^{1/2}, \left(\frac{n\Delta}{4L^3\sigma^2\tau_{\max}^2T}\right)^{1/4}\right\}$. Substituting this into (11) yields*

$$\frac{1}{T}\sum_{t=1}^T \mathbb{E}\|\nabla F(w^{t-1})\|_2^2 \leq \mathcal{O}\left(\frac{L\Delta\tau_{\max}}{T} + \left(\frac{L\Delta\sigma^2\tau_{\max}}{nT}\right)^{1/2} + \left(\frac{\sqrt[3]{L^3\sigma^2\tau_{\max}^2}\Delta}{\sqrt[3]{nT}}\right)^{3/4}\right)$$

*for all $T \geq 1$, as derived in Lemma 17 from Koloskova et al. (2020). However, this approach results in a degradation of the non-dominant term from $\mathcal{O}(1/T^{\frac{3}{2}})$ as shown in Corollary 1 to $\mathcal{O}(1/T^{\frac{3}{4}})$.*

**Remark 2.** *Theorem 1 covers the deterministic setting (i.e., $\sigma^2 = 0$) as a special case. By setting $\sigma^2 = 0$ and choosing $\eta = \frac{1}{64L\tau_{\max}}$, inequality (11) simplifies to $\frac{1}{T}\sum_{t=1}^T \mathbb{E}[\|\nabla F(\boldsymbol{w}^{t-1}\|_2^2)] \leq \mathcal{O}\left(\frac{L\Delta\tau_{\max}}{T}\right)$. This surpasses the rate of $\mathcal{O}(\sqrt{\frac{(1+\sigma^2)\tau_{\max}}{nT}})$ achieved by IA$^2$SGD in the stochastic setting. Furthermore, if $\tau_{\max} = \mathcal{O}(n)$, then the sample complexity of this deterministic algorithm to reach an $\epsilon$-stationary point is $\mathcal{O}\left(\frac{n}{\epsilon}\right)$, which aligns with that of the nonconvex full gradient descent (see, e.g., Reddi et al. (2016)).*

**Comparisons with Prior Theoretical Results.** We compare the theoretical performance of IA$^2$SGD with several representative distributed SGD-based algorithms as detailed in Table 1. Representative SGD-based algorithms that employ full aggregation strategies include sIAG (Wang et al., 2023a) and MIFA (Gu et al., 2021), while both of which operate synchronously. The analysis of sIAG is applicable only to strongly convex objectives. The convergence rate of MIFA is established based on the Lipschitz continuity of Hessians and boundedness of gradient noise, and the transient time of order $\Omega(n\tau_{\max}^2)$ exhibits a quadratic dependence on the maximum model delay. By contrast, our analysis is conducted under less restrictive conditions and achieves a reduced transient time. FedBuff (Nguyen et al., 2022), a notable *semi-asynchronous*[2] federated learning algorithm, incorporates

---

[2]Semi-asynchrony refers to the property that the server needs to wait for *multiple* participating workers to synchronize before performing an update, while the non-participating workers continue their on-going jobs asynchronously.

Table 1: Convergence rates of representative distributed SGD-based algorithms for smooth nonconvex problems with heterogeneous data. (Shorthand notation: **Async.** = Asynchronous, **Agg.** = Aggregation-based, **Add. Assump.** = Additional assumptions aside from Assumptions 1–4, BDH = Bounded Data Heterogeneity[3], BN = Bounded Noise[4], LH = Lipschitz Hessian[5], UWP = Uniform Worker Participation[6], BG = Bounded Gradients[7])

| Algorithms | Async.? | Agg.? | Convergence Rates | Add. Assump. |
|---|---|---|---|---|
| Synchronous SGD (Khaled & Richtárik, 2023) | No | **Yes** | $\mathcal{O}\left(\sqrt{\frac{\sigma^2}{nT}} + \frac{1}{T}\right)$ | – |
| MIFA (Gu et al., 2021) | No | **Yes** | $\mathcal{O}\left(\sqrt{\frac{1+\tau_{\mathrm{avg}}}{nKT}}\sigma^2 + \frac{nK\sigma\tau_{\max}\zeta + \sigma^2\tau_{\max}\delta\rho}{T}\right)$[8] | BDH, BN, LH |
| FedBuff (Nguyen et al., 2022) | Semi | Partial | $\mathcal{O}\left(\frac{\sigma^2+K\zeta^2}{\sqrt{mKT}} + \frac{K\tau_{\mathrm{avg}}\tau_{\max}\zeta^2 + \tau_{\max}\sigma^2}{T}\right)$[9] | BDH, UWP |
| Vanilla ASGD (Mishchenko et al., 2022) | **Yes** | No | $\mathcal{O}\left(\sqrt{\frac{\sigma^2}{T}} + \frac{n}{T} + \zeta_{\max}^2\right)$ | BDH |
| Uniform ASGD (Koloskova et al., 2022) | **Yes** | No | $\mathcal{O}\left(\sqrt{\frac{\sigma^2+\zeta^2}{T}} + \frac{\sqrt[3]{\tau_{\mathrm{avg}}\frac{1}{n}\sum_{i=1}^{n}\tau_{\mathrm{avg}}^i\zeta_i^2}}{T^{2/3}}\right)$ | BDH |
| Shuffled ASGD (Islamov et al., 2024) | **Yes** | No | $\mathcal{O}\left(\sqrt{\frac{\sigma^2}{T}} + \frac{(\sqrt{n}\zeta)^{2/3}+(nG)^{2/3}}{T^{2/3}} + \frac{n}{T}\right)$ | BDH, BG |
| IA$^2$SGD (This Paper) | **Yes** | **Yes** | $\mathcal{O}\left((1+\sigma^2)\sqrt{\frac{\tau_{\max}}{nT}} + \frac{\sigma^2\sqrt{n\tau_{\max}}}{T^{3/2}}\right)$ | – |

partial aggregation where only a subset of delayed local updates are considered during each model update of the server. There has been several convergence analyses for FedBuff (Nguyen et al., 2022; Toghani & Uribe, 2022; Wang et al., 2023b), while they all assume equal probability of worker's participation in the server's global update—an idealistic scenario that rarely holds in practical systems. In the context of asynchronous learning, existing ASGD algorithms (Mishchenko et al., 2022; Koloskova et al., 2022; Islamov et al., 2024) all require the data heterogeneity to be bounded, while this assumption is eliminated in our analysis for IA$^2$SGD.

## 5 NUMERICAL EXPERIMENTS

We simulate a distributed system comprising $n$ workers. To model the hardware variations across different workers, we employ the *fixed-computation-speed model* described in (Mishchenko et al., 2022). Specifically, each worker $i$ consistently takes fixed units of time, $s_i$, to compute a stochastic gradient. For each $i \in [n]$, $s_i$ is drawn from the truncated normal distribution $\mathcal{TN}(\mu, \mathtt{std})$ with a mean $\mu = 1$ and standard deviation $\mathtt{std} = 1$ and 5, ensuring all time values are greater than 0. A higher $\mathtt{std}$ indicates more significant hardware variation, leading to a greater maximum delay in the models during the training process. Furthermore, we assume that the communication time between the server and workers, as well as the server's computation time, are negligible. We implement IA$^2$SGD with mini-batching, along with other distributed SGD-based algorithms listed in Table 1. Additionally, we compare the Melenia SGD (Tyurin & Richtárik, 2023) that achieves the optimal time complexity under the fixed-computation-speed model. Each mini-batch comprises 64 samples, uniformly drawn from the local datasets allocated to the workers. We allow MIFA and FedBuff to perform one local step each, ensuring that the workers in these two algorithms have a computation workload comparable to that of the other algorithms.

---

[3]There exists $\zeta_i > 0$ such that $\|\nabla F_i(\boldsymbol{w}) - \nabla F(\boldsymbol{w})\|_2^2 \leq \zeta_i^2$ for all $i \in [n]$ and $\boldsymbol{w} \in \mathbb{R}^d$. Define $\zeta^2 := \frac{1}{n}\sum_{i=1}^{n}\zeta_i^2$ and $\zeta_{\max} := \max_{i\in[n]}\{\zeta_i\}$, which characterize the heterogeneity of data distributions.

[4]There exists $\delta > 0$ such that $\|\nabla f_i(\boldsymbol{w}; \boldsymbol{\xi}_i) - \nabla F_i(\boldsymbol{w})\|_2 \leq \delta$ almost surely for all $i \in [n]$, $\boldsymbol{w} \in \mathbb{R}^d$, and $\boldsymbol{\xi}_i \sim \mathbb{P}_i$.

[5]There exists $\rho > 0$ such that $\|\nabla^2 F_i(\boldsymbol{w}) - \nabla^2 F_i(\boldsymbol{w}')\|_2 \leq \rho\|\boldsymbol{w} - \boldsymbol{w}'\|_2$ for all $\boldsymbol{w}, \boldsymbol{w}' \in \mathbb{R}^d$.

[6]Every worker participates in each iteration of the server with equal probability.

[7]There exists $G \geq 0$ such that $\|\nabla F_i(\boldsymbol{w})\|_2^2 \leq G^2$ for all $i \in [n]$.

[8]$K$ is the number of local updates and $\tau_{\mathrm{avg}} := \frac{1}{n(T-1)}\sum_{t=1}^{T-1}\sum_{i=1}^{n}\tau_i(t)$.

[9]$m$ is the number of workers participating in each iteration of the server. We report the best-known convergence rate of FedBuff established in (Wang et al., 2023b).

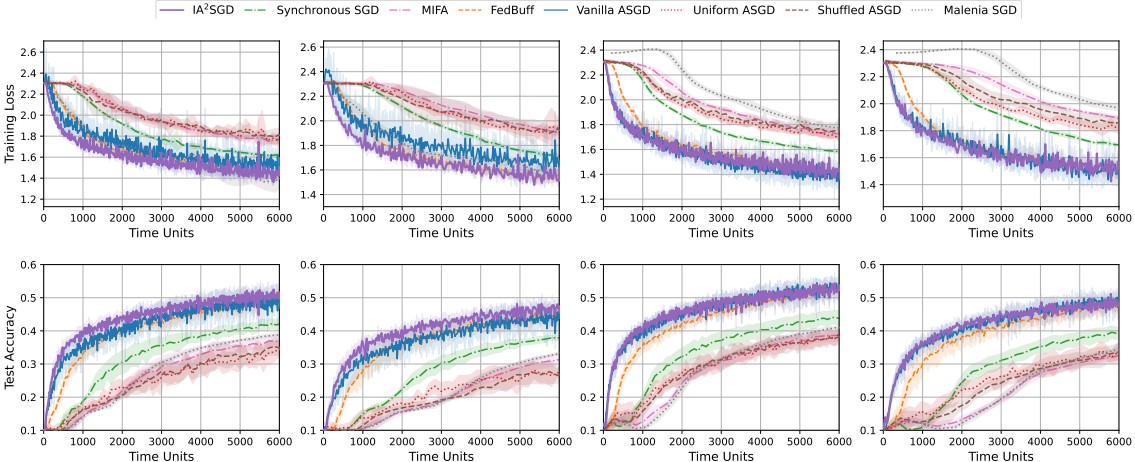

Figure 3: Convergence curves displaying training losses and test accuracies over time with $n = 10$ workers. (1st column: $\alpha = 0.1$, std $= 1$; 2nd column: $\alpha = 0.1$, std $= 5$; 3rd column: $\alpha = 0.5$, std $= 1$; 4th column: $\alpha = 0.5$, std $= 5$)

We evaluate the performance of these algorithms using the CIFAR-10 image dataset (Krizhevsky et al., 2009) by training a convolutional neural network with two convolutional layers for image classification. Following the approach described in Yurochkin et al. (2019), we allocate the dataset to the workers based on the Dirichlet distribution with concentration parameter $\alpha$. A lower $\alpha$ results in greater data heterogeneity among the workers. The step sizes for the algorithms under comparison are selected from the set $\{0.001, 0.005, 0.01\}$, based on which they achieve the fastest convergence.

## 5.1 NUMERICAL RESULTS

Each experiment is independently repeated three times using different random seeds, and the mean and standard deviation of the numerical performance for a configuration of $n = 10$ workers are shown in Figure 3. In scenarios of high data heterogeneity, specifically $\alpha = 0.1$, IA$^2$SGD achieves a better convergence rate in training loss and test accuracy. Additionally, IA$^2$SGD maintains consistent performance even as the computation speeds of the workers vary significantly, as indicated by an increasing std. This manifests its robustness to hardware variations. On the other hand, under conditions of low data heterogeneity, where $\alpha = 0.5$, the performance of IA$^2$SGD aligns closely with that of vanilla ASGD. This similarity supports the theoretical convergence rate of vanilla ASGD, which includes an additive $\zeta_{\max}^2$ term that becomes less significant in the low data heterogeneity regime. The convergence rate of synchronous SGD is theoretically invariant across different levels of data heterogeneity. However, its practical runtime performance suffers from the slowest worker, particularly as std increases. Furthermore, the Uniform ASGD does not deliver satisfactory outcomes, potentially because the repeated sampling of a slow worker before it completes its task can impair performance.

## 6 CONCLUSIONS

This paper introduces the Incremental Aggregated Asynchronous SGD (IA$^2$SGD), a novel approach to distributed machine learning that effectively counteracts the challenges posed by data heterogeneity across workers. By leveraging an asynchronous mechanism that utilizes stale gradients from all workers, IA$^2$SGD not only alleviates synchronization overheads but also balances the contributions of diverse worker datasets to the learning process. Our comprehensive theoretical analysis shows that IA$^2$SGD achieves an $\mathcal{O}((1 + \sigma^2)\sqrt{\tau_{\max}/nT})$ convergence rate for solving nonconvex problems, regardless of variations in data distribution across workers. This significant advancement highlights the robustness and efficiency of IA$^2$SGD in handling highly heterogeneous data scenarios, which are prevalent in modern distributed environments. Experiments on real-world datasets validate its superior runtime performance under high data heterogeneity in comparison to other leading distributed SGD-based algorithms, thereby confirming its potential as an effective tool for large-scale machine learning tasks. While our approach was to focus comprehensively on data heterogeneity, there can be room for improvement in reducing the dependence of our convergence rate on maximum delay. Whether our algorithmic framework is able to achieve both unbounded data heterogeneity and unbounded delay is worth further exploring.

**Reproducibility.** The source code is available at this anonymized link: `https://anonymous.4open.science/r/asgd/`.

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

## A ADDITIONAL RELATED WORKS

**Other Variants of ASGD.** While numerous variants of ASGD have been developed, they mostly address simpler scenarios with homogeneous data (Agarwal & Duchi, 2011; Lian et al., 2015; Feyzmahdavian et al., 2016; Leblond et al., 2018; Stich & Karimireddy, 2020; Arjevani et al., 2020; Dutta et al., 2021), where all workers operate on the same loss function and possess data with an identical probability distribution. In this setup, $F_1 = F_2 = \cdots = F_n$ and thus Problem (1) reduces to

$$\min_{\boldsymbol{w} \in \mathbb{R}^d} F_1(\boldsymbol{w}).$$

Another line of research explores ASGD for homogeneous data without parameter servers (Recht et al., 2011; De Sa et al., 2015; Mania et al., 2017), where multiple threads perform updates to the shared model parameters simultaneously without explicit synchronization. This means that while one thread is updating certain parameters, another thread can compute gradients and update other parameters concurrently. These algorithms operates most effectively when the data is sparse, so that concurrent updates are unlikely to interfere with each other significantly. An independent line of works have also considered ASGD in decentralized networks (Lian et al., 2018; Even et al., 2024), contrasting with the centralized architectures, as depicted in Figure 2, that rely on a central server.

**Asynchronous Federated Learning.** Federated learning (FL) is an emerging distributed machine learning paradigm that pays particular attention to data privacy and heterogeneity. Asynchronous federated learning algorithms (Xie et al., 2019; Chen et al., 2020; Nguyen et al., 2022; Zakerinia et al., 2022; Wang et al., 2023b; Fraboni et al., 2023; Wang et al., 2024; Leconte et al., 2024b) share similarities with ASGD but typically include a local update strategy that potentially reduces the communication frequency between the server and the workers. Among these works, FedBuff (Nguyen et al., 2022) serves as a representative algorithm where workers operate independently, and the server waits for a subset of workers $\mathcal{C}_t$ to submit their local updates in each global iteration. The local and global updates of FedBuff proceeds as follows:

$$\boldsymbol{w}_i^{\tau_i(t),k} = \boldsymbol{w}_i^{\tau_i(t),k-1} - \eta_\ell \nabla f_i(\boldsymbol{w}_i^{\tau_i(t),k-1}; \boldsymbol{\xi}_i^{t,k}), \ k = 1, 2, \ldots, K, \ i \in \mathcal{C}_t,$$
$$\boldsymbol{w}^t = \boldsymbol{w}^{t-1} - \frac{\eta_g}{|\mathcal{C}_t|} \sum_{i \in \mathcal{C}_t} (\boldsymbol{w}_i^{\tau_i(t),0} - \boldsymbol{w}_i^{\tau_i(t),K}), \ t = 1, 2, \ldots,$$

where $\eta_\ell$ and $\eta_g$ are the local and global step sizes, respectively. This approach is a *semi-asynchronous* algorithm designed to decrease communication frequency at the expense of increased waiting time. Nevertheless, the local update strategy in federated learning leads to the *client drift* phenomenon (Karimireddy et al., 2020; Sun et al., 2023), where local models at each worker tend to deviate from the global model. Therefore, asynchronous federated learning algorithms typically require either data heterogeneity or function dissimilarity to be bounded, so that local models remain closely aligned with the global model throughout the training process.

**Incremental Aggregated Gradient (IAG)-Type Methods:** The algorithmic concept of IA$^2$SGD is rooted in the well-established IAG methods (Blatt et al., 2007; Gurbuzbalaban et al., 2017; Vanli et al., 2018), which update the model parameters by using a combination of new and previously computed gradients. When solving Problem (1), the iterative formula of IAG can be expressed as:

$$\boldsymbol{w}^t = \boldsymbol{w}^{t-1} - \frac{\eta_t}{n} \sum_{i=1}^{n} \nabla F_i(\boldsymbol{w}^{t-\tau_i(t)}), \ t = 1, 2, \ldots,$$

where $\{\eta_t\}_{t \geq 1}$ are step sizes. IAG is suitable for asynchronous distributed implementations. SAG (Roux et al., 2012; Schmidt et al., 2017) and SAGA (Defazio et al., 2014) are randomized versions of IAG, where the index $i$

of the component function to be updated is selected at random in each iteration. Glasgow & Wootters (2022) developed ADSAGA, an extension of SAGA to the asynchronous setting, assuming a stochastic delay model and that the server is aware of the delay distribution. Nevertheless, these algorithms all assume that the *exact gradients* $\nabla F_i(\cdot)$ can be evaluated by each worker. This is a fundamental difference from our IA$^2$SGD, which relies solely on *stochastic gradients* $\nabla f_i(\cdot, \boldsymbol{\xi}_i)$ for any $\boldsymbol{\xi}_i \in \Xi_i$.

**Time Complexity Analyses:** Recent research has delved into the time complexities of parallel optimization algorithms, with notable examples including Melania SGD (Tyurin & Richtárik, 2023), Shadowheart SGD Tyurin et al. (2024b), and Freya PAGE Tyurin et al. (2024a). Melania SGD achieves optimal time complexity under the fixed-computation-time model, which is theoretically sound while may not fit real-world scenarios where computational speeds fluctuate. This method requires adherence to a global update criterion, incorporating an extra hyperparameter, which could lead to less frequent updates. Conversely, our IA$^2$SGD adapts more flexibly to these fluctuations with its greedy global update approach, updating server parameters as soon as any stochastic gradient is received, irrespective of delays. Shadowheart SGD (Tyurin et al., 2024b) and Freya PAGE (Tyurin et al., 2024a) extend these discussions by incorporating communication delays and shared dataset considerations, respectively, offering a broader view under both computation and communication heterogeneity.

# B  PROOFS OF MAIN RESULTS

For random variables $P, Q$ and function $h$, we denote by

$$\mathbb{E}_P[h(P,Q)] := \mathbb{E}[h(P,Q) \mid Q]$$

the *conditional expectation* with respect to $P$ while holding $Q$ constant.

## B.1  TECHNICAL LEMMAS

**Lemma 1.** *Suppose that Assumptions 3 and 4 hold. Then, it holds for all $t \geq 1$ that*

$$\mathbb{E}\left\|\frac{1}{n}\sum_{i=1}^{n}\left(\nabla f_i(\boldsymbol{w}^{t-\tau_i(t)}; \boldsymbol{\xi}_i^{t-\rho_i(t)}) - \nabla F_i(\boldsymbol{w}^{t-\tau_i(t)})\right)\right\|_2^2 \leq \frac{\sigma^2}{n}.$$

*Proof.* Expanding the squared norm gives

$$\mathbb{E}\left\|\frac{1}{n}\sum_{i=1}^{n}\left(\nabla f_i(\boldsymbol{w}^{t-\tau_i(t)}; \boldsymbol{\xi}_i^{t-\rho_i(t)}) - \nabla F_i(\boldsymbol{w}^{t-\tau_i(t)})\right)\right\|_2^2$$

$$= \frac{1}{n^2}\sum_{i=1}^{n}\mathbb{E}\|\nabla f_i(\boldsymbol{w}^{t-\tau_i(t)}; \boldsymbol{\xi}_i^{t-\rho_i(t)}) - \nabla F_i(\boldsymbol{w}^{t-\tau_i(t)})\|_2^2$$

$$+ \frac{1}{n^2}\sum_{i\neq j}\underbrace{\mathbb{E}\left\langle\nabla f_i(\boldsymbol{w}^{t-\tau_i(t)}; \boldsymbol{\xi}_i^{t-\rho_i(t)}) - \nabla F_i(\boldsymbol{w}^{t-\tau_i(t)}), \nabla f_j(\boldsymbol{w}^{t-\tau_j(t)}; \boldsymbol{\xi}_j^{t-\rho_j(t)}) - \nabla F_j(\boldsymbol{w}^{t-\tau_j(t)})\right\rangle}_{Y_{ij}} \quad (12)$$

To simplify $Y_{ij}$ for $i, j \in [n]$ such that $i \neq j$, we assume without loss of generality that $\rho_i(t) \geq \rho_j(t)$. Then, $t - \tau_i(t) \leq t - \rho_j(t)$ and thus $\boldsymbol{\xi}_j^{t-\rho_j(t)}$ is independent of $\boldsymbol{w}^{t-\tau_i(t)}$. Hence, using the law of total expectation and Assumption 3, we have

$$Y_{ij} = \mathbb{E}\left\langle\nabla f_i(\boldsymbol{w}^{t-\tau_i(t)}; \boldsymbol{\xi}_i^{t-\rho_i(t)}) - \nabla F_i(\boldsymbol{w}^{t-\tau_i(t)}), \nabla f_j(\boldsymbol{w}^{t-\tau_j(t)}; \boldsymbol{\xi}_j^{t-\rho_j(t)}) - \nabla F_j(\boldsymbol{w}^{t-\tau_j(t)})\right\rangle$$

$$= \mathbb{E}\left[\mathbb{E}_{\boldsymbol{\xi}_j^{t-\rho_j(t)}\sim\mathbb{P}_j}\left\langle\nabla f_i(\boldsymbol{w}^{t-\tau_i(t)}; \boldsymbol{\xi}_i^{t-\rho_i(t)}) - \nabla F_i(\boldsymbol{w}^{t-\tau_i(t)}), \nabla f_j(\boldsymbol{w}^{t-\tau_j(t)}; \boldsymbol{\xi}_j^{t-\rho_j(t)}) - \nabla F_j(\boldsymbol{w}^{t-\tau_j(t)})\right\rangle\right]$$

$$= \mathbb{E}\left[\left\langle\nabla f_i(\boldsymbol{w}^{t-\tau_i(t)}; \boldsymbol{\xi}_i^{t-\rho_i(t)}) - \nabla F_i(\boldsymbol{w}^{t-\tau_i(t)}), \mathbb{E}_{\boldsymbol{\xi}_j^{t-\rho_j(t)}}\left[\nabla f_j(\boldsymbol{w}^{t-\tau_j(t)}; \boldsymbol{\xi}_j^{t-\rho_j(t)}) - \nabla F_j(\boldsymbol{w}^{t-\tau_j(t)})\right]\right\rangle\right]$$

$$= 0.$$

Substituting this back into (12) gives

$$\mathbb{E}\left\|\frac{1}{n}\sum_{i=1}^{n}\left(\nabla f_i(\boldsymbol{w}^{t-\tau_i(t)};\boldsymbol{\xi}_i^{t-\rho_i(t)})-\nabla F_i(\boldsymbol{w}^{t-\tau_i(t)})\right)\right\|_2^2$$

$$=\frac{1}{n^2}\sum_{i=1}^{n}\mathbb{E}\|\nabla f_i(\boldsymbol{w}^{t-\tau_i(t)};\boldsymbol{\xi}_i^{t-\rho_i(t)})-\nabla F_i(\boldsymbol{w}^{t-\tau_i(t)})\|_2^2$$

$$=\frac{1}{n^2}\sum_{i=1}^{n}\mathbb{E}\left[\mathbb{E}\left[\|\nabla f_i(\boldsymbol{w}^{t-\tau_i(t)};\boldsymbol{\xi}_i^{t-\rho_i(t)})-\nabla F_i(\boldsymbol{w}^{t-\tau_i(t)})\|_2^2 \mid \boldsymbol{w}^{t-\tau_i(t)}\right]\right]$$

$$\leq \frac{\sigma^2}{n}. \tag{13}$$

where the second equality holds due to the law of total expectation and the inequality follows from Assumption 4. $\qquad\square$

**Lemma 2.** *Suppose that Assumptions 3 and 4 hold. Then, it holds for all $i \in [n]$ and $t \geq 1$ that*

$$\mathbb{E}\|\boldsymbol{w}^t-\boldsymbol{w}^{t-\tau_i(t)}\|_2^2 \leq 2\tau_{\max}^2\eta^2\frac{\sigma^2}{n}+2\tau_{\max}\eta^2\sum_{s=1+[t-\tau_{\max}]_+}^{t}\mathbb{E}\left\|\frac{1}{n}\sum_{j=1}^{n}\nabla F_j(\boldsymbol{w}^{s-\tau_j(s)})\right\|_2^2.$$

*Additionally, we have*

$$\mathbb{E}\|\boldsymbol{w}^t-\boldsymbol{w}^{[t-\tau_{\max}]_+}\|_2^2 \leq 2\tau_{\max}^2\eta^2\frac{\sigma^2}{n}+2\tau_{\max}\eta^2\sum_{s=1+[t-\tau_{\max}]_+}^{t}\mathbb{E}\left\|\frac{1}{n}\sum_{j=1}^{n}\nabla F_j(\boldsymbol{w}^{s-\tau_j(s)})\right\|_2^2.$$

*Proof.* For all $i \in [n]$ and $t \geq 1$, it follows from the telescoping sum

$$\sum_{s=1+t-\tau_i(t)}^{t}\left(\boldsymbol{w}^s-\boldsymbol{w}^{s-1}\right)=\boldsymbol{w}^t-\boldsymbol{w}^{t-\tau_i(t)}$$

and the iterative formula (3) that

$$\mathbb{E}\|\boldsymbol{w}^t-\boldsymbol{w}^{t-\tau_i(t)}\|_2^2$$

$$=\mathbb{E}\left\|\sum_{s=1+t-\tau_i(t)}^{t}\left(\boldsymbol{w}^s-\boldsymbol{w}^{s-1}\right)\right\|_2^2$$

$$=\mathbb{E}\left\|\sum_{s=1+t-\tau_i(t)}^{t}\eta\boldsymbol{g}^s\right\|_2^2$$

$$=\mathbb{E}\left\|\sum_{s=1+t-\tau_i(t)}^{t}\frac{\eta}{n}\sum_{j=1}^{n}\nabla f_j(\boldsymbol{w}^{s-\tau_j(s)};\boldsymbol{\xi}_j^{s-\rho_j(s)})\right\|_2^2$$

$$=\frac{\eta^2}{n^2}\mathbb{E}\left\|\sum_{s=1+t-\tau_i(t)}^{t}\sum_{j=1}^{n}\left(\nabla f_j(\boldsymbol{w}^{s-\tau_j(s)};\boldsymbol{\xi}_j^{s-\rho_j(s)})-\nabla F_j(\boldsymbol{w}^{s-\tau_j(s)})+\nabla F_j(\boldsymbol{w}^{s-\tau_j(s)})\right)\right\|_2^2$$

$$\leq\frac{2\eta^2}{n^2}\underbrace{\mathbb{E}\left\|\sum_{s=1+t-\tau_i(t)}^{t}\sum_{j=1}^{n}\left(\nabla f_j(\boldsymbol{w}^{s-\tau_j(s)};\boldsymbol{\xi}_j^{s-\rho_j(s)})-\nabla F_j(\boldsymbol{w}^{s-\tau_j(s)})\right)\right\|_2^2}_{\Phi_1}$$

$$+\frac{2\eta^2}{n^2}\underbrace{\mathbb{E}\left\|\sum_{s=1+t-\tau_i(t)}^{t}\sum_{j=1}^{n}\nabla F_j(\boldsymbol{w}^{s-\tau_j(s)})\right\|_2^2}_{\Phi_2}, \tag{14}$$

where the inequality uses the fact that $\|\boldsymbol{x} + \boldsymbol{y}\|_2^2 \leq 2\|\boldsymbol{x}\|_2^2 + 2\|\boldsymbol{y}\|_2^2$ for vectors $\boldsymbol{x}$ and $\boldsymbol{y}$. Subsequently, we upper bound $\Phi_1$ and $\Phi_2$, respectively.

Upper bounding $\Phi_1$: Expanding $\Phi_1$, we have

$$
\Phi_1 = \sum_{s=1+t-\tau_i(t)}^{t} \mathbb{E} \left\| \sum_{j=1}^{n} \left( \nabla f_j(\boldsymbol{w}^{s-\tau_j(s)}; \boldsymbol{\xi}_i^{s-\rho_j(s)}) - \nabla F_j(\boldsymbol{w}^{s-\tau_j(s)}) \right) \right\|_2^2
$$
$$
+ \sum_{\substack{s,s':s\neq s', \\ 1+t-\tau_i(t) \leq s,s' \leq t}} \mathbb{E} \underbrace{\left\langle \sum_{j=1}^{n} \left( \nabla f_j(\boldsymbol{w}^{s-\tau_j(s)}; \boldsymbol{\xi}_j^{s-\rho_j(s)}) - \nabla F_j(\boldsymbol{w}^{s-\tau_j(s)}) \right), \sum_{j=1}^{n} \left( \nabla f_j(\boldsymbol{w}^{s'-\tau_j(s')}; \boldsymbol{\xi}_j^{s'-\rho_j(s')}) - \nabla F_j(\boldsymbol{w}^{s'-\tau_j(s')}) \right) \right\rangle}_{Z^{ss'}}.
$$
(15)

Inspecting the inner product terms in (15), we note that for all $s, s' \in [1+t-\tau_i(t), t]$ such that $s \neq s'$,

$$
Z^{ss'} = \mathbb{E} \left\langle \sum_{j=1}^{n} \left( \nabla f_j(\boldsymbol{w}^{s-\tau_j(s)}; \boldsymbol{\xi}_j^{s-\rho_j(s)}) - \nabla F_j(\boldsymbol{w}^{s-\tau_j(s)}) \right), \sum_{j=1}^{n} \left( \nabla f_j(\boldsymbol{w}^{s'-\tau_j(s')}; \boldsymbol{\xi}_j^{s'-\rho_j(s')}) - \nabla F_j(\boldsymbol{w}^{s'-\tau_j(s')}) \right) \right\rangle
$$
$$
= \mathbb{E} \left[ \sum_{i=1}^{n} \sum_{j=1}^{n} \left\langle \nabla f_i(\boldsymbol{w}^{s-\tau_i(s)}; \boldsymbol{\xi}_i^{s-\rho_i(s)}) - \nabla F_i(\boldsymbol{w}^{s-\tau_i(s)}), \nabla f_j(\boldsymbol{w}^{s'-\tau_j(s')}; \boldsymbol{\xi}_j^{s'-\rho_j(s')}) - \nabla F_j(\boldsymbol{w}^{s'-\tau_j(s')}) \right\rangle \right]
$$
$$
= \sum_{i=1}^{n} \sum_{j=1}^{n} \mathbb{E} \left\langle \nabla f_i(\boldsymbol{w}^{s-\tau_i(s)}; \boldsymbol{\xi}_i^{s-\rho_i(s)}) - \nabla F_i(\boldsymbol{w}^{s-\tau_i(s)}), \nabla f_j(\boldsymbol{w}^{s'-\tau_j(s')}; \boldsymbol{\xi}_j^{s'-\rho_j(s')}) - \nabla F_j(\boldsymbol{w}^{s'-\tau_j(s')}) \right\rangle
$$
$$
= \sum_{j=1}^{n} \mathbb{E} \left\langle \nabla f_j(\boldsymbol{w}^{s-\tau_j(s)}; \boldsymbol{\xi}_j^{s-\rho_j(s)}) - \nabla F_j(\boldsymbol{w}^{s-\tau_j(s)}), \nabla f_j(\boldsymbol{w}^{s'-\tau_j(s')}; \boldsymbol{\xi}_j^{s'-\rho_j(s')}) - \nabla F_j(\boldsymbol{w}^{s'-\tau_j(s')}) \right\rangle
$$
$$
+ \sum_{i,j:i\neq j} \mathbb{E} \underbrace{\left\langle \nabla f_i(\boldsymbol{w}^{s-\tau_i(s)}; \boldsymbol{\xi}_i^{s-\rho_i(s)}) - \nabla F_i(\boldsymbol{w}^{s-\tau_i(s)}), \nabla f_j(\boldsymbol{w}^{s'-\tau_j(s')}; \boldsymbol{\xi}_j^{s'-\rho_j(s')}) - \nabla F_j(\boldsymbol{w}^{s'-\tau_j(s')}) \right\rangle}_{Z_{ij}^{ss'}}
$$
(16)

To simplify $Z_{ij}^{ss'}$ for all $i, j \in [n]$ and $i \neq j$, we assume with out loss of generality that $s - \rho_i(s) \geq s' - \rho_j(s')$. This, together with the fact that $\rho_j(s') \leq \tau_j(s')$ by (5), implies that $s - \rho_i(s) \geq s' - \tau_j(s')$. Thus, $\boldsymbol{\xi}_j^{s-\rho_i(s)}$ is independent of $\boldsymbol{w}^{s'-\rho_j(s')}$. Further using the law of total expectation and Assumption 3, we have

$$
Z_{ij}^{ss'} = \mathbb{E} \left[ \mathbb{E}_{\boldsymbol{\xi}_i^{s-\rho_i(s)}} \left\langle \nabla f_i(\boldsymbol{w}^{s-\tau_i(s)}; \boldsymbol{\xi}_j^{s-\rho_i(s)}) - \nabla F_i(\boldsymbol{w}^{s-\tau_i(s)}), \nabla f_j(\boldsymbol{w}^{s'-\tau_i(s')}; \boldsymbol{\xi}_j^{s'-\rho_j(s')}) - \nabla F_j(\boldsymbol{w}^{s'-\tau_i(s')}) \right\rangle \right]
$$
$$
= \mathbb{E} \left[ \left\langle \mathbb{E}_{\boldsymbol{\xi}_i^{s-\rho_i(s)}} \left[ \nabla f_i(\boldsymbol{w}^{s-\tau_i(s)}; \boldsymbol{\xi}_i^{s-\rho_i(s)}) - \nabla F_i(\boldsymbol{w}^{s-\tau_i(s)}) \right], \nabla f_j(\boldsymbol{w}^{s'-\tau_i(s')}; \boldsymbol{\xi}_j^{s'-\rho_j(s')}) - \nabla F_j(\boldsymbol{w}^{s'-\tau_i(s')}) \right\rangle \right]
$$
$$
= 0
$$

Substituting this back into (16) and using Assumption 4, we have

$$
Z^{ss'} = \sum_{j=1}^{n} \mathbb{E} \left\langle \nabla f_j(\boldsymbol{w}^{s-\tau_j(s)}; \boldsymbol{\xi}_j^{t-\rho_j(s)}) - \nabla F_j(\boldsymbol{w}^{s-\tau_j(s)}), \nabla f_j(\boldsymbol{w}^{s'-\tau_i(s')}; \boldsymbol{\xi}_j^{s'-\rho_j(s')}) - \nabla F_j(\boldsymbol{w}^{s'-\tau_i(s')}) \right\rangle
$$
$$
\leq \frac{1}{2} \sum_{j=1}^{n} \mathbb{E} \|\nabla f_j(\boldsymbol{w}^{s-\tau_j(s)}; \boldsymbol{\xi}_j^{t-\rho_j(s)}) - \nabla F_j(\boldsymbol{w}^{s-\tau_j(s)})\|_2^2
$$
$$
+ \frac{1}{2} \sum_{j=1}^{n} \mathbb{E} \|\nabla f_j(\boldsymbol{w}^{s'-\tau_i(s')}; \boldsymbol{\xi}_j^{s'-\rho_j(s')}) - \nabla F_j(\boldsymbol{w}^{s'-\tau_i(s')})\|_2^2
$$
$$
\leq n\sigma^2.
$$

Plugging this back into (15) and using Lemma 1 yield

$$
\begin{aligned}
\Phi_1 &\leq \sum_{s=1+t-\tau_i(t)}^{t} n\sigma^2 + \sum_{\substack{s,s':s\neq s',\\ 1+t-\tau_i(t)\leq s,s'\leq t}} n\sigma^2 \\
&= \tau_i(t)n\sigma^2 + (\tau_i(t)^2 - \tau_i(t))n\sigma^2 \\
&= \tau_i(t)^2 n\sigma^2 \\
&\leq n\tau_{\max}^2\sigma^2.
\end{aligned}
\tag{17}
$$

Upper bounding $\Phi_2$: Following the fact that $\|\sum_{i=1}^{m} \boldsymbol{x}_i\|_2^2 \leq m \sum_{i=1}^{m} \|\boldsymbol{x}_i\|_2^2$ for vectors $\boldsymbol{x}_1, \ldots, \boldsymbol{x}_m$, we have

$$
\begin{aligned}
\Phi_2 &= \mathbb{E}\left\|\sum_{s=1+t-\tau_i(t)}^{t}\sum_{j=1}^{n}\nabla F_j(\boldsymbol{w}^{s-\tau_j(s)})\right\|_2^2 \\
&\leq \tau_i(t)\sum_{s=1+t-\tau_i(t)}^{t}\mathbb{E}\left\|\sum_{j=1}^{n}\nabla F_j(\boldsymbol{w}^{s-\tau_j(s)})\right\|_2^2. \\
&\leq \tau_{\max}\sum_{s=1+[t-\tau_{\max}]_+}^{t}\mathbb{E}\left\|\sum_{j=1}^{n}\nabla F_j(\boldsymbol{w}^{s-\tau_j(s)})\right\|_2^2.
\end{aligned}
\tag{18}
$$

Substituting (17) and (18) back into (14) gives

$$
\mathbb{E}\|\boldsymbol{w}^t - \boldsymbol{w}^{t-\tau_i(t)}\|_2^2 \leq \frac{2\sigma^2}{n}\tau_{\max}^2\eta^2 + 2\tau_{\max}\eta^2\sum_{s=1+[t-\tau_{\max}]_+}^{t}\mathbb{E}\left\|\frac{1}{n}\sum_{j=1}^{n}\nabla F_j(\boldsymbol{w}^{s-\tau_j(s)})\right\|_2^2,
$$

as desired. $\qquad\square$

**Lemma 3.** *Suppose that Assumptions 2–4 hold. Then, it holds for all $i \in [n]$ and $t \geq 1$ that*

$$
\mathbb{E}\|\boldsymbol{g}^t\|_2^2 \leq \left(2 + 8L^2\tau_{\max}^2\eta^2\right)\frac{\sigma^2}{n} + 4\mathbb{E}\|\nabla F(\boldsymbol{w}^{t-1})\|_2^2
$$

$$
+ 8L^2\tau_{\max}\eta^2\sum_{s=1+[t-\tau_{\max}]_+}^{t}\mathbb{E}\left\|\frac{1}{n}\sum_{j=1}^{n}\nabla F_j(\boldsymbol{w}^{s-\tau_j(s)})\right\|_2^2.
$$

*Proof.* Following the fact that $\|\boldsymbol{x}+\boldsymbol{y}\|_2^2 \leq 2\|\boldsymbol{x}\|_2^2 + 2\|\boldsymbol{y}\|_2^2$ for vectors $\boldsymbol{x}$ and $\boldsymbol{y}$, we have

$$
\begin{aligned}
&\mathbb{E}\|\boldsymbol{g}^t\|_2^2 \\
&= \mathbb{E}\left\|\frac{1}{n}\sum_{i=1}^{n}\nabla f_i(\boldsymbol{w}^{t-\tau_i(t)};\boldsymbol{\xi}_i^{t-\rho_i(t)})\right\|_2^2 \\
&= \mathbb{E}\left\|\frac{1}{n}\sum_{i=1}^{n}\left(\nabla f_i(\boldsymbol{w}^{t-\tau_i(t)};\boldsymbol{\xi}_i^{t-\rho_i(t)}) - \nabla F_i(\boldsymbol{w}^{t-\tau_i(t)})\right) + \frac{1}{n}\sum_{i=1}^{n}\nabla F_i(\boldsymbol{w}^{t-\tau_i(t)})\right\|_2^2 \\
&\leq 2\mathbb{E}\left\|\frac{1}{n}\sum_{i=1}^{n}\left(\nabla f_i(\boldsymbol{w}^{t-\tau_i(t)};\boldsymbol{\xi}_i^{t-\rho_i(t)}) - \nabla F_i(\boldsymbol{w}^{t-\tau_i(t)})\right)\right\|_2^2 + 2\mathbb{E}\left\|\frac{1}{n}\sum_{i=1}^{n}\nabla F_i(\boldsymbol{w}^{t-\tau_i(t)})\right\|_2^2 \\
&\leq \frac{2\sigma^2}{n} + 2\underbrace{\mathbb{E}\left\|\frac{1}{n}\sum_{i=1}^{n}\nabla F_i(\boldsymbol{w}^{t-\tau_i(t)})\right\|_2^2}_{\Psi}.
\end{aligned}
\tag{19}
$$

where the last inequality holds due to Lemma 1. It suffices to upper bound $\Psi$. We observe that

$$
\begin{aligned}
\Psi &= \mathbb{E}\left\|\frac{1}{n}\sum_{i=1}^{n}\nabla F_i(\boldsymbol{w}^{t-\tau_i(t)})\right\|_2^2 \\
&= \mathbb{E}\left\|\frac{1}{n}\sum_{i=1}^{n}\left(\nabla F_i(\boldsymbol{w}^{t-\tau_i(t)}) - \nabla F_i(\boldsymbol{w}^t)\right) + \frac{1}{n}\sum_{i=1}^{n}\nabla F_i(\boldsymbol{w}^{t-1})\right\|_2^2 \\
&\leq 2\mathbb{E}\left\|\frac{1}{n}\sum_{i=1}^{n}\left(\nabla F_i(\boldsymbol{w}^{t-\tau_i(t)}) - \nabla F_i(\boldsymbol{w}^t)\right)\right\|_2^2 + 2\mathbb{E}\left\|\frac{1}{n}\sum_{i=1}^{n}\nabla F_i(\boldsymbol{w}^{t-1})\right\|_2^2 \\
&\leq \frac{2}{n}\sum_{i=1}^{n}\mathbb{E}\|\nabla F_i(\boldsymbol{w}^{t-\tau_i(t)}) - \nabla F_i(\boldsymbol{w}^t)\|_2^2 + 2\mathbb{E}\|\nabla F(\boldsymbol{w}^{t-1})\|_2^2 \\
&\leq \frac{2L^2}{n}\sum_{i=1}^{n}\mathbb{E}\|\boldsymbol{w}^{t-\tau_i(t)} - \boldsymbol{w}^t\|_2^2 + 2\mathbb{E}\|\nabla F(\boldsymbol{w}^{t-1})\|_2^2,
\end{aligned}
\tag{20}
$$

where the second inequality uses the fact that $\|\sum_{i=1}^{m}\boldsymbol{x}_i\|_2^2 \leq m\sum_{i=1}^{m}\|\boldsymbol{x}_i\|_2^2$ for vectors $\boldsymbol{x}_1,\ldots,\boldsymbol{x}_m$. Substituting Lemma 2 into (20) gives

$$
\begin{aligned}
\Psi &\leq 2L^2\left(\frac{2\sigma^2}{n}\tau_{\max}^2\eta^2 + 2\tau_{\max}\eta^2\sum_{s=1+[t-\tau_{\max}]_+}^{t}\mathbb{E}\left\|\frac{1}{n}\sum_{j=1}^{n}\nabla F_j(\boldsymbol{w}^{s-\tau_j(s)})\right\|_2^2\right) + 2\mathbb{E}\|\nabla F(\boldsymbol{w}^{t-1})\|_2^2 \\
&= \frac{4\sigma^2}{n}L^2\tau_{\max}^2\eta^2 + 4L^2\tau_{\max}\eta^2\sum_{s=1+[t-\tau_{\max}]_+}^{t}\mathbb{E}\left\|\frac{1}{n}\sum_{j=1}^{n}\nabla F_j(\boldsymbol{w}^{s-\tau_j(s)})\right\|_2^2 + 2\mathbb{E}\|\nabla F(\boldsymbol{w}^{t-1})\|_2^2.
\end{aligned}
\tag{21}
$$

Plugging (21) back into (19) gives

$$
\begin{aligned}
\mathbb{E}\|\boldsymbol{g}^t\|_2^2 &\leq \left(2 + 8L^2\tau_{\max}^2\eta^2\right)\frac{\sigma^2}{n} + 4\mathbb{E}\|\nabla F(\boldsymbol{w}^{t-1})\|_2^2 \\
&\quad + 8L^2\tau_{\max}\eta^2\sum_{s=1+[t-\tau_{\max}]_+}^{t}\mathbb{E}\left\|\frac{1}{n}\sum_{j=1}^{n}\nabla F_j(\boldsymbol{w}^{s-\tau_j(s)})\right\|_2^2,
\end{aligned}
$$

as desired. $\qquad\square$

### B.2 PROOF OF PROPOSITION 1

*Proof.* We first decompose the inner product into two terms:

$$
\mathbb{E}\langle\nabla F(\boldsymbol{w}^{t-1}),\boldsymbol{g}^t\rangle = \underbrace{\mathbb{E}\left\langle\nabla F(\boldsymbol{w}^{[t-\tau_{\max}]_+}),\boldsymbol{g}^t\right\rangle}_{A} + \underbrace{\mathbb{E}\left\langle\nabla F(\boldsymbol{w}^{t-1}) - \nabla F(\boldsymbol{w}^{[t-\tau_{\max}]_+}),\boldsymbol{g}^t\right\rangle}_{B}.
\tag{22}
$$

Subsequently, we lower bound $A$ and $B$, respectively.

**Lower bounding $A$:** Since $\rho_i(t) \le \tau_i(t) \le \tau_{\max}$ for all $i \in [n]$, then $t - \rho_i(t) \ge t - \tau_{\max}$ for all $i \in [n]$, which implies that $\boldsymbol{\xi}_1^{t-\rho_1(t)}, \ldots, \boldsymbol{\xi}_n^{t-\rho_n(t)}$ are independent of $\boldsymbol{w}^{[t-\tau_{\max}]+}$. Then, we have

$$
\begin{aligned}
A &= \mathbb{E}\left[\left\langle \nabla F(\boldsymbol{w}^{[t-\tau_{\max}]+}), \boldsymbol{g}^t \right\rangle\right] \\
&\overset{(a)}{=} \mathbb{E}\left[\mathbb{E}_{\boldsymbol{\xi}_1^{t-\rho_1(t)}, \ldots, \boldsymbol{\xi}_n^{t-\rho_n(t)}}\left[\left\langle \nabla F(\boldsymbol{w}^{[t-\tau_{\max}]+}), \boldsymbol{g}^t \right\rangle\right]\right] \\
&= \mathbb{E}\left\langle \nabla F(\boldsymbol{w}^{[t-\tau_{\max}]+}), \mathbb{E}_{\boldsymbol{\xi}_i^{t-\rho_i(t)}}\left[\frac{1}{n}\sum_{i=1}^{n} \nabla f_i(\boldsymbol{w}^{t-\tau_i(t)}; \boldsymbol{\xi}_i^{t-\rho_i(t)})\right]\right\rangle \\
&\overset{(b)}{=} \mathbb{E}\left\langle \nabla F(\boldsymbol{w}^{[t-\tau_{\max}]+}), \frac{1}{n}\sum_{i=1}^{n} \nabla F_i(\boldsymbol{w}^{t-\tau_i(t)})\right\rangle \\
&= \underbrace{\mathbb{E}\left\langle \nabla F(\boldsymbol{w}^{[t-\tau_{\max}]+}) - \nabla F(\boldsymbol{w}^{t-1}), \frac{1}{n}\sum_{i=1}^{n} \nabla F_i(\boldsymbol{w}^{t-\tau_i(t)})\right\rangle}_{A_1} \\
&\quad + \underbrace{\mathbb{E}\left\langle \nabla F(\boldsymbol{w}^{t-1}), \frac{1}{n}\sum_{i=1}^{n} \nabla F_i(\boldsymbol{w}^{t-\tau_i(t)})\right\rangle}_{A_2},
\end{aligned}
\tag{23}
$$

where $(a)$ use the law of total expectation, and $(b)$ holds due to Assumption 3. Then, we lower bound $A_1$ as follows:

$$
\begin{aligned}
A_1 &= \mathbb{E}\left\langle \nabla F(\boldsymbol{w}^{[t-\tau_{\max}]+}) - \nabla F(\boldsymbol{w}^{t-1}), \frac{1}{n}\sum_{i=1}^{n}\left(\nabla F_i(\boldsymbol{w}^{t-\tau_i(t)}) - \nabla F_i(\boldsymbol{w}^{t-1})\right)\right\rangle \\
&\quad + \mathbb{E}\left\langle \nabla F(\boldsymbol{w}^{[t-\tau_{\max}]+}) - \nabla F(\boldsymbol{w}^{t-1}), \nabla F(\boldsymbol{w}^{t-1})\right\rangle \\
&\ge -\frac{1}{2}\mathbb{E}\|\nabla F(\boldsymbol{w}^{[t-\tau_{\max}]+}) - \nabla F(\boldsymbol{w}^{t-1})\|_2^2 - \frac{1}{2}\mathbb{E}\left\|\frac{1}{n}\sum_{i=1}^{n}\left(\nabla F_i(\boldsymbol{w}^{t-\tau_i(t)}) - \nabla F_i(\boldsymbol{w}^{t-1})\right)\right\|_2^2 \\
&\quad - \mathbb{E}\|\nabla F(\boldsymbol{w}^{[t-\tau_{\max}]+}) - \nabla F(\boldsymbol{w}^{t-1})\|_2^2 - \frac{1}{4}\mathbb{E}\|\nabla F(\boldsymbol{w}^{t-1})\|_2^2 \\
&= -\frac{1}{4}\mathbb{E}\|\nabla F(\boldsymbol{w}^{t-1})\|_2^2 - \frac{3}{2}\mathbb{E}\|\nabla F(\boldsymbol{w}^{[t-\tau_{\max}]+}) - \nabla F(\boldsymbol{w}^{t-1})\|_2^2 \\
&\quad - \frac{1}{2}\mathbb{E}\left\|\frac{1}{n}\sum_{i=1}^{n}\left(\nabla F_i(\boldsymbol{w}^{t-\tau_i(t)}) - \nabla F_i(\boldsymbol{w}^{t-1})\right)\right\|_2^2,
\end{aligned}
\tag{24}
$$

where the inequality uses the fact that $\langle \boldsymbol{x}, \boldsymbol{y} \rangle \ge -\frac{1}{2}\|\boldsymbol{x}\|_2^2 - \frac{1}{2}\|\boldsymbol{y}\|_2^2$ and $\langle \boldsymbol{x}, \boldsymbol{y} \rangle \ge -\|\boldsymbol{x}\|_2^2 - \frac{1}{4}\|\boldsymbol{y}\|_2^2$ for vectors $\boldsymbol{x}$ and $\boldsymbol{y}$. Using the identity $\langle \boldsymbol{x}, \boldsymbol{y} \rangle = \frac{1}{2}\|\boldsymbol{x}\|_2^2 + \frac{1}{2}\|\boldsymbol{y}\|_2^2 - \frac{1}{2}\|\boldsymbol{x} - \boldsymbol{y}\|_2^2$ for vectors $\boldsymbol{x}$ and $\boldsymbol{y}$, we can express $A_2$ as

$$
\begin{aligned}
A_2 &= \frac{1}{2}\mathbb{E}\|\nabla F(\boldsymbol{w}^{t-1})\|_2^2 + \frac{1}{2}\mathbb{E}\left\|\frac{1}{n}\sum_{i=1}^{n} \nabla F_i(\boldsymbol{w}^{t-\tau_i(t)})\right\|_2^2 \\
&\quad - \frac{1}{2}\mathbb{E}\left\|\nabla F(\boldsymbol{w}^{t-1}) - \frac{1}{n}\sum_{i=1}^{n} \nabla F_i(\boldsymbol{w}^{t-\tau_i(t)})\right\|_2^2.
\end{aligned}
\tag{25}
$$

Putting (24) and (25) back into (23) gives

$$
\begin{aligned}
A &\geq \frac{1}{4}\mathbb{E}\|\nabla F(\boldsymbol{w}^{t-1})\|_2^2 - \frac{3}{2}\mathbb{E}\|\nabla F(\boldsymbol{w}^{t-1}) - \nabla F(\boldsymbol{w}^{[t-\tau_{\max}]+})\|_2^2 \\
&\quad - \mathbb{E}\left\|\frac{1}{n}\sum_{i=1}^{n}\left(\nabla F_i(\boldsymbol{w}^{t-1}) - \nabla F_i(\boldsymbol{w}^{t-\tau_i(t)})\right)\right\|_2^2 + \frac{1}{2}\mathbb{E}\left\|\frac{1}{n}\sum_{i=1}^{n}\nabla F_i(\boldsymbol{w}^{t-\tau_i(t)})\right\|_2^2 \\
&\overset{(a)}{\geq} \frac{1}{4}\mathbb{E}\|\nabla F(\boldsymbol{w}^{t-1})\|_2^2 - \frac{3L^2}{2}\mathbb{E}\|\boldsymbol{w}^t - \boldsymbol{w}^{[t-\tau_{\max}]+}\|_2^2 \\
&\quad - \frac{L^2}{n}\sum_{i=1}^{n}\mathbb{E}\|\boldsymbol{w}^t - \boldsymbol{w}^{t-\tau_i(t)}\|_2^2 + \frac{1}{2}\mathbb{E}\left\|\frac{1}{n}\sum_{i=1}^{n}\nabla F_i(\boldsymbol{w}^{t-\tau_i(t)})\right\|_2^2 \\
&\overset{(b)}{\geq} \frac{1}{4}\mathbb{E}\|\nabla F(\boldsymbol{w}^{t-1})\|_2^2 + \frac{1}{2}\mathbb{E}\left\|\frac{1}{n}\sum_{i=1}^{n}\nabla F_i(\boldsymbol{w}^{t-\tau_i(t)})\right\|_2^2 \\
&\quad - \frac{5L^2}{2}\left(2\tau_{\max}^2\eta^2\frac{\sigma^2}{n} + 2\tau_{\max}\eta^2\sum_{s=1+[t-\tau_{\max}]+}^{t}\mathbb{E}\left\|\frac{1}{n}\sum_{j=1}^{n}\nabla F_j(\boldsymbol{w}^{s-\tau_j(s)})\right\|_2^2\right) \\
&= \frac{1}{4}\mathbb{E}\|\nabla F(\boldsymbol{w}^{t-1})\|_2^2 - 5L^2\tau_{\max}^2\eta^2\frac{\sigma^2}{n} + \frac{1}{2}\mathbb{E}\left\|\frac{1}{n}\sum_{i=1}^{n}\nabla F_i(\boldsymbol{w}^{t-\tau_i(t)})\right\|_2^2 \\
&\quad - 5L^2\tau_{\max}\eta^2\sum_{s=1+[t-\tau_{\max}]+}^{t}\mathbb{E}\left\|\frac{1}{n}\sum_{j=1}^{n}\nabla F_j(\boldsymbol{w}^{s-\tau_j(s)})\right\|_2^2,
\end{aligned}
\tag{26}
$$

where $(a)$ uses Assumption 2 and $(b)$ uses Lemma 2.

Lower bounding $B$: We observe that

$$
\begin{aligned}
B &= \mathbb{E}\left\langle\nabla F(\boldsymbol{w}^{t-1}) - \nabla F(\boldsymbol{w}^{[t-\tau_{\max}]+}), \boldsymbol{g}^t\right\rangle \\
&\overset{(a)}{\geq} -\mathbb{E}\left[\|\nabla F(\boldsymbol{w}^{t-1}) - \nabla F(\boldsymbol{w}^{[t-\tau_{\max}]+})\|_2\|\boldsymbol{g}^t\|_2\right] \\
&\overset{(b)}{\geq} -L\mathbb{E}\left[\|\boldsymbol{w}^t - \boldsymbol{w}^{[t-\tau_{\max}]+}\|_2\|\boldsymbol{g}^t\|_2\right] \\
&\overset{(c)}{=} -L\mathbb{E}\left[\left\|\sum_{s=1+[t-\tau_{\max}]+}^{t}\eta\boldsymbol{g}^s\right\|_2\|\boldsymbol{g}^t\|_2\right] \\
&\overset{(d)}{\geq} -L\mathbb{E}\left[\sum_{s=1+[t-\tau_{\max}]+}^{t}\eta\|\boldsymbol{g}^s\|_2\|\boldsymbol{g}^t\|_2\right] \\
&\overset{(e)}{\geq} -L\eta\sum_{s=1+[t-\tau_{\max}]+}^{t}\frac{1}{2}\left(\mathbb{E}\|\boldsymbol{g}^s\|_2^2 + \mathbb{E}\|\boldsymbol{g}^t\|_2^2\right) \\
&= -\frac{L\eta}{2}\sum_{s=1+[t-\tau_{\max}]+}^{t}\mathbb{E}\|\boldsymbol{g}^s\|_2^2 - \frac{L\eta}{2}\tau_{\max}\mathbb{E}\|\boldsymbol{g}^t\|_2^2,
\end{aligned}
\tag{27}
$$

where $(a)$ follows from the Cauchy-Schwartz inequality, $(b)$ follows from Assumption 2, $(c)$ uses the telescoping sum $\boldsymbol{w}^t - \boldsymbol{w}^{[t-\tau_{\max}]+} = \sum_{s=1+[t-\tau_{\max}]+}^{t}\left(\boldsymbol{w}^s - \boldsymbol{w}^{s-1}\right) = \sum_{s=1+[t-\tau_{\max}]+}^{t}\eta\boldsymbol{g}^s$, $(d)$ uses the triangle

inequality, and $(e)$ is due to the Young's inequality. Combining (27) with Lemma 3, we have

$$
B \geq -\frac{L\eta}{2} \sum_{s=1+[t-\tau_{\max}]_+}^{t} \left( \left(2 + 8L^2\tau_{\max}^2\eta^2\right)\frac{\sigma^2}{n} + 4\mathbb{E}\|\nabla F(\boldsymbol{w}^{s-1})\|_2^2 \right.
$$

$$
\left. +8L^2\tau_{\max}\eta^2 \sum_{s'=1+[s-\tau_{\max}]_+}^{s} \mathbb{E}\left\|\frac{1}{n}\sum_{j=1}^{n}\nabla F_j(\boldsymbol{w}^{s'-\tau_i(s')})\right\|_2^2 \right)
$$

$$
-\frac{L\eta}{2}\tau_{\max}\left( \left(2 + 8L^2\tau_{\max}^2\eta^2\right)\frac{\sigma^2}{n} + 4\mathbb{E}\|\nabla F(\boldsymbol{w}^{t-1})\|_2^2 \right.
$$

$$
\left. +8L^2\tau_{\max}\eta^2 \sum_{s=1+[t-\tau_{\max}]_+}^{t} \mathbb{E}\left\|\frac{1}{n}\sum_{j=1}^{n}\nabla F_j(\boldsymbol{w}^{s-\tau_j(s)})\right\|_2^2 \right)
$$

$$
= -\left(2L\tau_{\max}\eta + 8L^3\tau_{\max}^3\eta^3\right)\frac{\sigma^2}{n} - 2L\eta \sum_{s=1+[t-\tau_{\max}]_+}^{t} \mathbb{E}\|\nabla F(\boldsymbol{w}^{s-1})\|_2^2 - 2L\tau_{\max}\eta\mathbb{E}\|\nabla F(\boldsymbol{w}^{t-1})\|_2^2
$$

$$
- 4L^3\tau_{\max}\eta^3 \sum_{s=1+[t-\tau_{\max}]_+}^{t} \sum_{s'=1+[s-\tau_{\max}]_+}^{s} \mathbb{E}\left\|\frac{1}{n}\sum_{j=1}^{n}\nabla F_j(\boldsymbol{w}^{s'-\tau_i(s')})\right\|_2^2
$$

$$
- 4L^3\tau_{\max}^2\eta^3 \sum_{s=1+[t-\tau_{\max}]_+}^{t} \mathbb{E}\left\|\frac{1}{n}\sum_{j=1}^{n}\nabla F_j(\boldsymbol{w}^{s-\tau_j(s)})\right\|_2^2
$$

$$
\geq -\left(2L\tau_{\max}\eta + 8L^3\tau_{\max}^3\eta^3\right)\frac{\sigma^2}{n} - 2L\eta \sum_{s=1+[t-\tau_{\max}]_+}^{t} \mathbb{E}\|\nabla F(\boldsymbol{w}^{s-1})\|_2^2 - 2L\tau_{\max}\eta\mathbb{E}\|\nabla F(\boldsymbol{w}^{t-1})\|_2^2
$$

$$
- 8L^3\tau_{\max}^2\eta^3 \sum_{s=1+[t-2\tau_{\max}]_+}^{t} \mathbb{E}\left\|\frac{1}{n}\sum_{j=1}^{n}\nabla F_j(\boldsymbol{w}^{s-\tau_j(s)})\right\|_2^2, \tag{28}
$$

where the last inequality uses the fact that $\sum_{s=1+[t-K]_+}^{t}\sum_{s'=1+[s-K]_+}^{s} a_{s'} \leq K\sum_{s=1+[t-2K]_+}^{t} a_s$ for $a_1,\ldots,a_t \geq 0$ and $K \geq 1$.

Substituting (26) and (28) into (22) and simplifying it, we have

$$
\mathbb{E}\langle\nabla F(\boldsymbol{w}^{t-1}), \boldsymbol{g}^t\rangle
$$

$$
\geq \left(\frac{1}{4} - 2L\tau_{\max}\eta\right)\mathbb{E}\|\nabla F(\boldsymbol{w}^{t-1})\|_2^2 - 2L\eta \sum_{s=1+[t-\tau_{\max}]_+}^{t} \mathbb{E}\|\nabla F(\boldsymbol{w}^{s-1})\|_2^2
$$

$$
- \left(2L\tau_{\max}\eta + 5L^2\tau_{\max}^2\eta^2 + 8L^3\tau_{\max}^3\eta^3\right)\frac{\sigma^2}{n} + \frac{1}{2}\mathbb{E}\left\|\frac{1}{n}\sum_{i=1}^{n}\nabla F_i(\boldsymbol{w}^{t-\tau_i(t)})\right\|_2^2
$$

$$
- \left(5L^2\tau_{\max}\eta^2 + 8L^3\tau_{\max}^2\eta^3\right) \sum_{s=1+[t-2\tau_{\max}]_+}^{t} \mathbb{E}\left\|\frac{1}{n}\sum_{j=1}^{n}\nabla F_j(\boldsymbol{w}^{s-\tau_j(s)})\right\|_2^2
$$

$$
\geq \frac{1}{8}\mathbb{E}\|\nabla F(\boldsymbol{w}^{t-1})\|_2^2 - 2L\eta \sum_{s=1+[t-\tau_{\max}]_+}^{t} \mathbb{E}\|\nabla F(\boldsymbol{w}^{s-1})\|_2^2 - 3L\tau_{\max}\eta\frac{\sigma^2}{n}
$$

$$
+ \frac{1}{2}\mathbb{E}\left\|\frac{1}{n}\sum_{i=1}^{n}\nabla F_i(\boldsymbol{w}^{t-\tau_i(t)})\right\|_2^2 - 6L^2\tau_{\max}\eta^2 \sum_{s=1+[t-2\tau_{\max}]_+}^{t} \mathbb{E}\left\|\frac{1}{n}\sum_{j=1}^{n}\nabla F_j(\boldsymbol{w}^{s-\tau_j(s)})\right\|_2^2,
$$

where the last inequality holds because the stepsize condition $\eta \leq 1/(16L\tau_{\max})$ implies the following:

$$\frac{1}{4} - 2L\tau_{\max}\eta \geq \frac{1}{8},$$

$$2L\tau_{\max}\eta + 5L^2\tau_{\max}^2\eta^2 + 8L^3\tau_{\max}^3\eta^3 \leq 2L\tau_{\max}\eta + 6L^2\tau_{\max}^2\eta^2 \leq 3L\tau_{\max}\eta,$$

$$5L^2\tau_{\max}\eta^2 + 8L^3\tau_{\max}^2\eta^3 \leq 6L^2\tau_{\max}\eta^2.$$

This completes the proof. $\qquad\square$

### B.3   PROOF OF THEOREM 1

*Proof.* Since $F$ is $L$-smooth, it follows from the descent lemma that

$$\mathbb{E}[F(\boldsymbol{w}^t)] - \mathbb{E}[F(\boldsymbol{w}^{t-1})] \leq \mathbb{E}[\langle \nabla F(\boldsymbol{w}^{t-1}), \boldsymbol{w}^t - \boldsymbol{w}^{t-1}\rangle] + \frac{L}{2}\mathbb{E}\|\boldsymbol{w}^t - \boldsymbol{w}^{t-1}\|_2^2$$

$$= -\eta\mathbb{E}\langle \nabla F(\boldsymbol{w}^{t-1}), \boldsymbol{g}^t\rangle + \frac{L\eta^2}{2}\mathbb{E}\|\boldsymbol{g}^t\|_2^2.$$

Applying Lemma 3 and Proposition 1, we obtain

$$\mathbb{E}[F(\boldsymbol{w}^t)] - \mathbb{E}[F(\boldsymbol{w}^{t-1})]$$

$$\leq -\frac{1}{8}\eta\mathbb{E}\|\nabla F(\boldsymbol{w}^{t-1})\|_2^2 + 2L\eta^2 \sum_{s=1+[t-\tau_{\max}]_+}^{t} \mathbb{E}\|\nabla F(\boldsymbol{w}^{s-1})\|_2^2 + 3L\tau_{\max}\eta^2\frac{\sigma^2}{n}$$

$$- \frac{1}{2}\eta\mathbb{E}\left\|\frac{1}{n}\sum_{i=1}^{n}\nabla F_i(\boldsymbol{w}^{t-\tau_i(t)})\right\|_2^2 + 6L^2\tau_{\max}\eta^3 \sum_{s=1+[t-2\tau_{\max}]_+}^{t} \mathbb{E}\left\|\frac{1}{n}\sum_{j=1}^{n}\nabla F_j(\boldsymbol{w}^{s-\tau_j(s)})\right\|_2^2$$

$$+ \left(L\eta^2 + 4L^3\tau_{\max}^2\eta^4\right)\frac{\sigma^2}{n} + 4L^3\tau_{\max}\eta^4 \sum_{s=1+[t-\tau_{\max}]_+}^{t} \mathbb{E}\left\|\frac{1}{n}\sum_{j=1}^{n}\nabla F_j(\boldsymbol{w}^{s-\tau_j(s)})\right\|_2^2$$

$$+ 2L\eta^2\mathbb{E}\|\nabla F(\boldsymbol{w}^{t-1})\|_2^2$$

$$\leq -\left(\frac{1}{8}\eta - 2L\eta^2\right)\mathbb{E}\|\nabla F(\boldsymbol{w}^{t-1})\|_2^2 + 2L\eta^2 \sum_{s=1+[t-\tau_{\max}]_+}^{t} \mathbb{E}\|\nabla F(\boldsymbol{w}^{s-1})\|_2^2$$

$$+ (L\eta^2 + 3L\tau_{\max}\eta^2 + 4L^3\tau_{\max}^2\eta^4)\frac{\sigma^2}{n} - \frac{1}{2}\eta\mathbb{E}\left\|\frac{1}{n}\sum_{i=1}^{n}\nabla F_i(\boldsymbol{w}^{t-\tau_i(t)})\right\|_2^2$$

$$+ (6L^2\tau_{\max}\eta^3 + 4L^3\tau_{\max}\eta^4) \sum_{s=1+[t-2\tau_{\max}]_+}^{t} \mathbb{E}\left\|\frac{1}{n}\sum_{j=1}^{n}\nabla F_j(\boldsymbol{w}^{s-\tau_j(s)})\right\|_2^2$$

$$\leq -\frac{1}{16}\eta\mathbb{E}\|\nabla F(\boldsymbol{w}^{t-1})\|_2^2 + 2L\eta^2 \sum_{s=1+[t-\tau_{\max}]_+}^{t} \mathbb{E}\|\nabla F(\boldsymbol{w}^{s-1})\|_2^2$$

$$+ (4L\tau_{\max}\eta^2 + 4L^3\tau_{\max}^2\eta^4)\frac{\sigma^2}{n}$$

$$- \frac{1}{2}\eta\mathbb{E}\left\|\frac{1}{n}\sum_{i=1}^{n}\nabla F_i(\boldsymbol{w}^{t-\tau_i(t)})\right\|_2^2 + 7L^2\tau_{\max}\eta^3 \sum_{s=1+[t-2\tau_{\max}]_+}^{t} \mathbb{E}\left\|\frac{1}{n}\sum_{j=1}^{n}\nabla F_j(\boldsymbol{w}^{s-\tau_j(s)})\right\|_2^2, \qquad (29)$$

where the last inequality holds because $\tau_{\max} \geq 1 \Rightarrow L\eta^2 + 3L\tau_{\max}\eta^2 \leq 4L\tau_{\max}\eta$ and by requiring the following stepsize conditions:

$$\eta \leq \frac{1}{32L} \iff \frac{1}{8}\eta - 2L\eta^2 \geq \frac{1}{16}\eta, \qquad (30)$$

$$\eta \leq \frac{1}{4L} \implies 6L^2\tau_{\max}\eta^3 + 4L^3\tau_{\max}\eta^4 \leq 7L^2\tau_{\max}\eta^3. \qquad (31)$$

Summing up both sides of inequality (29) for $t = 1, \ldots, T$ yields

$$\mathbb{E}[F(\boldsymbol{w}^T)] - F(\boldsymbol{w}^0)$$

$$\leq -\frac{1}{16}\eta\sum_{t=1}^{T}\mathbb{E}\|\nabla F(\boldsymbol{w}^{t-1})\|_2^2 + 2L\eta^2\sum_{t=1}^{T}\sum_{s=1+[t-\tau_{\max}]_+}^{t}\mathbb{E}\|\nabla F(\boldsymbol{w}^{s-1})\|_2^2$$

$$+ \sum_{t=1}^{T}(4L\tau_{\max}\eta^2 + 4L^3\tau_{\max}^2\eta^4)\frac{\sigma^2}{n}$$

$$- \frac{1}{2}\eta\sum_{t=1}^{T}\mathbb{E}\left\|\frac{1}{n}\sum_{j=1}^{n}\nabla F_j(\boldsymbol{w}^{t-\tau_j(t)})\right\|_2^2 + 7L^2\tau_{\max}\eta^3\sum_{t=1}^{T}\sum_{s=1+[t-2\tau_{\max}]_+}^{t}\mathbb{E}\left\|\frac{1}{n}\sum_{j=1}^{n}\nabla F_j(\boldsymbol{w}^{s-\tau_j(s)})\right\|_2^2$$

$$\leq -\frac{1}{16}\eta\sum_{t=1}^{T}\mathbb{E}\|\nabla F(\boldsymbol{w}^{t-1})\|_2^2 + 2L\tau_{\max}\eta^2\sum_{t=1}^{T}\mathbb{E}\|\nabla F(\boldsymbol{w}^{t-1})\|_2^2$$

$$+ \sum_{t=1}^{T}(4L\tau_{\max}\eta^2 + 4L^3\tau_{\max}^2\eta^4)\frac{\sigma^2}{n}$$

$$- \frac{1}{2}\eta\sum_{t=1}^{T}\mathbb{E}\left\|\frac{1}{n}\sum_{j=1}^{n}\nabla F_j(\boldsymbol{w}^{t-\tau_j(t)})\right\|_2^2 + 14L^2\tau_{\max}^2\eta^3\sum_{t=1}^{T}\mathbb{E}\left\|\frac{1}{n}\sum_{j=1}^{n}\nabla F_j(\boldsymbol{w}^{t-\tau_j(t)})\right\|_2^2$$

$$= -\left(\frac{1}{16}\eta - 2L\tau_{\max}\eta^2\right)\sum_{t=1}^{T}\mathbb{E}\|\nabla F(\boldsymbol{w}^{t-1})\|_2^2 + \sum_{t=1}^{T}(4L\tau_{\max}\eta^2 + 4L^3\tau_{\max}^2\eta^4)\frac{\sigma^2}{n}$$

$$- \left(\frac{1}{2}\eta - 14L^2\tau_{\max}^2\eta^3\right)\sum_{t=1}^{T}\mathbb{E}\left\|\frac{1}{n}\sum_{j=1}^{n}\nabla F_j(\boldsymbol{w}^{t-\tau_j(t)})\right\|_2^2$$

$$\leq -\frac{1}{32}\eta\sum_{t=1}^{T}\mathbb{E}\|\nabla F(\boldsymbol{w}^{t-1})\|_2^2 + \sum_{t=1}^{T}(4L\tau_{\max}\eta^2 + 4L^3\tau_{\max}^2\eta^4)\frac{\sigma^2}{n}, \tag{32}$$

where the second inequality uses the fact that $\sum_{t=1}^{T}\sum_{s=1+[t-K]_+}^{t}a_s \leq K\sum_{t=1}^{T}a_t$ for $a_1, \ldots, a_T \geq 0$ and $K \geq 1$, and the last inequality holds by requiring the following stepsize conditions:

$$\eta \leq \frac{1}{64L\tau_{\max}} \iff \frac{1}{16}\eta - 2L\tau_{\max}\eta^2 \geq \frac{1}{32}\eta, \tag{33}$$

$$\eta \leq \frac{1}{\sqrt{28}L\tau_{\max}} \iff \frac{1}{2}\eta - 14L^2\tau_{\max}^2\eta^3 \geq 0. \tag{34}$$

Note that the stepsize conditions (30), (31), (33), and (34) are uniformly implied by $\eta \leq \frac{1}{64L\tau_{\max}}$. Rearranging (32) and using Assumption 1, we obtain

$$\frac{1}{T}\sum_{t=1}^{T}\mathbb{E}\|\nabla F(\boldsymbol{w}^{t-1})\|_2^2 \leq \frac{32(F(\boldsymbol{w}^0) - F^*)}{T\eta} + 128L\tau_{\max}\eta\frac{\sigma^2}{n} + 128L^3\tau_{\max}^2\eta^3\frac{\sigma^2}{n}. \tag{35}$$

which completes the proof of Theorem 1. $\qquad\square$

### B.4 PROOF OF COROLLARY 1

*Proof.* Taking $\eta = \frac{1}{2}\sqrt{\frac{n(F(\boldsymbol{w}^0)-F^*)}{L\tau_{\max}T}}$ in the right-hand-side of (11) in Theorem 1, we obtain

$$\frac{1}{T}\sum_{t=1}^{T}\mathbb{E}\|\nabla F(\boldsymbol{w}^{t-1})\|_2^2 \leq 64(1+\sigma^2)\sqrt{\frac{L\tau_{\max}(F(\boldsymbol{w}^0)-F^*)}{nT}} + \frac{16L\sigma^2((F(\boldsymbol{w}^0)-F^*))^{3/2}\sqrt{n\tau_{\max}}}{T^{3/2}}.$$

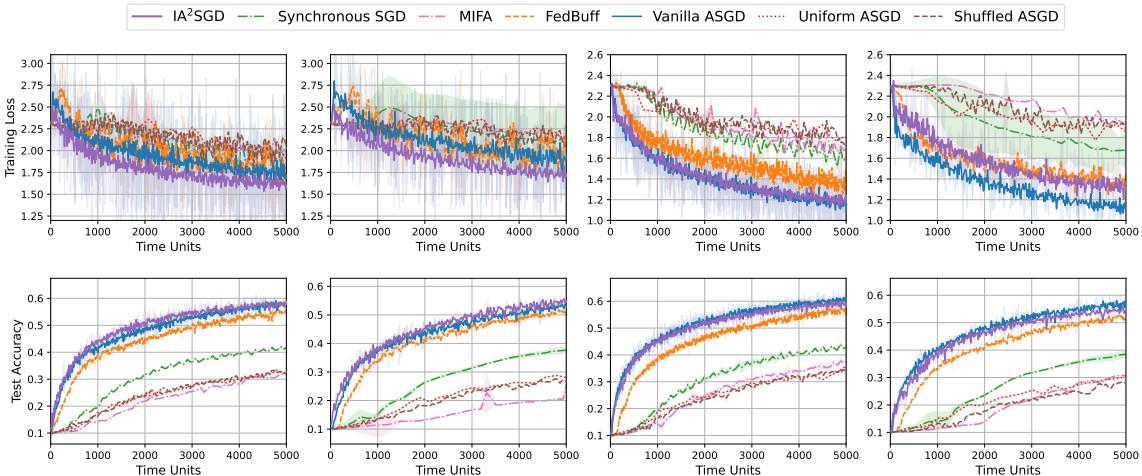

Figure 4: Convergence curves displaying training losses and test accuracies over time with $n = 30$ workers. (1st column: $\alpha = 0.05$, std $= 1$; 2nd column: $\alpha = 0.05$, std $= 5$; 3rd column: $\alpha = 0.1$, std $= 1$; 4th column: $\alpha = 0.1$, std $= 5$)

Note that Theorem 1 holds under the stepsize condition $\eta \leq \frac{1}{64L\tau_{\max}}$. If we take $\eta = \frac{1}{2}\sqrt{\frac{n(F(\boldsymbol{w}^0) - F^*)}{L\tau_{\max}T}}$, then the stepsize conditions can be satisfied when

$$\frac{1}{2}\sqrt{\frac{n(F(\boldsymbol{w}^0) - F^*)}{L\tau_{\max}T}} \leq \frac{1}{64L\tau_{\max}} \iff T \geq 1024L(F(\boldsymbol{w}^0) - F^*)n\tau_{\max},$$

which completes the proof. □

## C ADDITIONAL EXPERIMENTAL DETAILS AND NUMERICAL RESULTS

**Data Partitioning.** Following the approaches adopted in many works (Yurochkin et al., 2019; Hsu et al., 2019; Li et al., 2022), we use Dirichlet distribution to split the CIFAR-10 dataset into $S$ subsets. The training set in CIFAR-10 consists of 50,000 images with 10 different classes. For each class $k \in [10]$, we generate a generate a vector $\boldsymbol{p}_k \in \mathbb{R}^S$ from the $S$-dimensional Dirichlet distribution with concentration parameter $\alpha$, whose probability density is given by

$$\text{Dir}_S(\boldsymbol{p}_k; \alpha) := \frac{1}{B(\alpha)} \prod_{i=1}^{S} p_{k,i}^{\alpha-1}.$$

Here, $B(\alpha) := \frac{\prod_{i=1}^{S} \Gamma(\alpha)}{\Gamma(S\alpha)}$ is the Beta function, $\Gamma(\cdot)$ is the Gamma function, and $\boldsymbol{p}_k$ satisfies $p_{k,i} \in [0, 1]$ and $\sum_{i=1}^{S} p_{k,i} = 1$. After generating $\boldsymbol{p}_1, \ldots, \boldsymbol{p}_{10}$, each instance of class $k$ is assigned to worker $i$ with probability $p_{k,i}$.

**Numerical Results for $n = 30$ Workers.** We conduct experiments with a configuration of $n = 30$ workers. Increasing the number of workers $n$ in the Dirichlet distribution with a given concentration parameter $\alpha$ tends to result in more balanced data partitioning. For our experiments, we select $\alpha = 0.05$ and $0.1$ to observe the effects with $n = 30$. Each experiment is independently conducted three times using different random seeds. We report the mean and standard deviation of the numerical performance for this configuration in Figure 4. We observe that IA$^2$SGD and other algorithms display similar performance patterns, as previously shown in Figure 3, across different levels of data heterogeneity.

