# OpenReview forum: "Incremental Aggregated Asynchronous SGD for Arbitrarily Heterogeneous Data"
_ICLR.cc/2025/Conference — Submitted to ICLR 2025_

### Official Review · Reviewer_nXSQ · 2024-10-28

**Soundness:** 3
**Presentation:** 2
**Contribution:** 2
**Rating:** 5
**Confidence:** 4

**Summary:**

The paper introduces a new asynchronous SGD-based algorithm. Allowing each worker and server to have a memory term (buffer), the algorithm can provably converge without data heterogeneity bounds. The authors provide convergence guarantees for smooth non-convex functions and demonstrate the practical performance in training CNN model on CIFAR10 dataset simulating gradient computation time from truncated normal distribution and data heterogeneity by allocating data according to Dirichlet distribution.

**Strengths:**

- The authors provide a new asynchronous algorithm that provably converges without imposing data heterogeneity which is essential for applications such as Federated Learning
- The rate improves with the number of workers $n$
- The authors demonstrate the performance in training CNN model on CIFAR10 dataset varying data heterogeneity by Dirichlet data allocation

**Weaknesses:**

- Bad dependency on the maximum delay $\tau_{\max}$ (we can create some example with $\Delta=L=1$ for that to satisfy the requirement on $T$ from Theorem 1; although this lower bound on $T$ is not needed to derive the convergence rate for the proposed algorithm). If the slowest worker responds once in the training, i.e. $\tau_{\max} \sim T$ then the theory shows no convergence while some previous algorithms can still achieve the convergence in this extreme case.
- The empirical results show that the proposed algorithm might be affected by the heterogeneity (the accuracy after the same number of iterations/time increases when the heterogeneity decreases, i.e. the problem becomes more homogeneous).
- The experiments do not show the improvement of the proposed algorithm over other algorithms (those convergence guarantees are affected by the heterogeneity bounds). The proposed algorithm performs similarly to FedBuff and vanilla ASGD even when the heterogeneity is high. These observations contradict the theory
- Some important related works are missing [1,2]. The methods from these papers also do not require data heterogeneity bounds but the comparison is not provided in this work.

[1] Alexander Tyurin and Peter Richtárik, Optimal time complexities of parallel stochastic optimization methods under a fixed computation model, NeurIPS 2023
[2] Alexander Tyurin, Marta Pozzi, Ivan Ilin, and Peter Richtárik, Shadowheart SGD: Distributed asynchronous SGD with optimal time complexity under arbitrary computation and communication heterogeneity, NeurIPS 2024

**Questions:**

- It is not clear why the analysis needs two sequences $\tau_i(t)$ and $\rho_i(t)$. Isn't there a bijection between them? I guess it would be needed if we were allowed to assign a new job to a worker different from the one who finished the computations.
- Could the authors show that the proposed method can have a better dependency on $\tau_{\max}$ under stronger assumptions (e.g., bounded gradients)? Can we get both better dependency on $\tau_{\max}$ and no requirement to bound the data heterogeneity simultaneously?
- Could the authors provide a detailed comparison with algorithms from [1,2]?
- The first line in Figure 4 is not readable. It is hard to distinguish the variance of algorithms. Could the authors make them more readable? Could you please clarify why there is no variance in accuracy plots for some algorithms (FedBuff, Synchronous SGD, Uniform ASGD)? Why does the red shaded area (variance) around the red dashed line in Figure 4 (second line) look so weird? Sometimes red dashed line is not inside red shaded area.
- I find it weird that uniform/shuffled ASGD performs worse than vanilla ASGD. Is there any particular reason for that since from theory vanilla ASGD converges to $\zeta^2$-neighbourhood of the solution while uniform/shuffled ASGD should converge exactly?
- The rate of the proposed algorithm in the table looks weird: why is there $\sigma$ in the denominator of the second term?
- Why is the accuracy so low? All algorithms barely achieved 50 % accuracy which I found low for the CIFAR10 dataset. Could you present the results when the algorithms' accuracy stops increasing after a sufficient amount of ''time''?
- Could the authors provide the empirical comparison against Melania SGD?

---

> ### Author Response · Authors · 2024-11-20
>
> > Bad dependency on the maximum delay $\tau_{\max}$ (we can create some example with $\Delta=L=1$ for that to satisfy the requirement on  from Theorem 1; although this lower bound on  is not needed to derive the convergence rate for the proposed algorithm). If the slowest worker responds once in the training, i.e., $\tau_{\max} \sim T$, then the theory shows no convergence while some previous algorithms can still achieve the convergence in this extreme case.
> >
>
> **Response:**
>
> Theorem 1 states that as long as the slowest worker respond once no more than $\tau_{\max}$ iterations, the algorithm converges as $T \rightarrow \infty$. Here, $\tau_{\max}$ is treated as a constant that **does not scale with $T$**.
>
> Additionally, Theorem 1 implies that when $\tau_{\max} = \mathcal{O}(n)$, the dependence on $\tau_{\max}$ in the dominant term of the convergence bound is offset by the speedup factor $1/n$. Consequently, the convergence rate of IA$^2$SGD is $\mathcal{O}\left( (1+\sigma^2)\sqrt{\frac{1}{T}} + \frac{\sigma^2n}{T^{3/2}}\right)$.
>
> > The empirical results show that the proposed algorithm might be affected by the heterogeneity (the accuracy after the same number of iterations/time increases when the heterogeneity decreases, i.e. the problem becomes more homogeneous).
> >
>
> **Response:**
>
> It is important to clarify that changes in dataset heterogeneity imply a different data partitioning across workers, thereby altering each worker’s local objective function $F_i$. While our theoretical framework suggests that IA$^2$SGD is designed to minimize dependency on dataset heterogeneity, this does not imply that IA$^2$SGD will exhibit identical convergence behaviors across varying levels of heterogeneity. The theoretical convergence bound indicates that IA$^2$SGD is less affected by data heterogeneity compared to other methods. Nonetheless, the bound applies even in scenarios with the most challenging $F_i$ configurations, indicating that some variation in performance as heterogeneity levels change is both anticipated and consistent with our empirical results.
>
> > The experiments do not show the improvement of the proposed algorithm over other algorithms (those convergence guarantees are affected by the heterogeneity bounds). The proposed algorithm performs similarly to FedBuff and vanilla ASGD even when the heterogeneity is high. These observations contradict the theory.
> >
>
> **Response:**
>
> I would like to draw attention to the specific results presented in Figure 3. The high-data-heterogeneity scenario ($\alpha=0.1$), shown in the 1st and 2nd columns of Figure 3, IA$^2$SGD demonstrates a faster drop in training loss and achieves higher test accuracy compared to the other algorithms. In contrast, the low-data-heterogeneity scenario ($\alpha=0.5$), depicted in the 3rd and 4th columns of Figure 3, shows IA$^2$SGD performing comparably to vanilla ASGD. Although the performance is similar, IA$^2$SGD still ranks among the top performers and exceeds that of FedBuff. Our empirical findings show that IA$^2$SGD maintains robust performance across varying levels of data heterogeneity, aligning with our theoretical predictions.

---

> > ### Author Response · Authors · 2024-11-20
> >
> > > Some important related works are missing [1, 2]. The methods from these papers also do not require data heterogeneity bounds but the comparison is not provided in this work.
> > [1] Alexander Tyurin and Peter Richtárik, Optimal time complexities of parallel stochastic optimization methods under a fixed computation model, NeurIPS 2023.
> > [2] Alexander Tyurin, Marta Pozzi, Ivan Ilin, and Peter Richtárik, Shadowheart SGD: Distributed asynchronous SGD with optimal time complexity under arbitrary computation and communication heterogeneity, NeurIPS 2024.
> > >
> >
> > **Response:**
> >
> > We appreciate the reviewer's references to these recent publications. Regarding paper [1], a detailed comparison is given as follows:
> >
> > 1. *Computation Model Assumptions:* The Melania SGD in [1] achieves optimal time complexity under a fixed-computation model, which assumes uniform and predictable computation times across all workers. Despite the theoretical advancement in this paper, it might not always be practical in real-world scenarios where computational capabilities can vary significantly. In contrast, our IA$^2$SGD does not rely on such specific computation models, making it more flexible and applicable to a wider range of real-world conditions where computation time and communication delays can vary.
> > 2. *Asynchrony Mechanisms:* The asynchrony mechanism in IA$^2$SGD involves a **greedy** global update scheme, where the server updates its parameters whenever any stochastic gradient is received, regardless of the worker's delay. This approach contrasts with Malenia SGD, where the server updates are conditional on achieving a global criteria $\left( \frac{1}{n} \sum_{i=1}^{n} \frac{1}{B_i} \right)^{-1} < \frac{S}{n}$ based on the computation of $B_i$ gradients on worker $i$ and a system-wide parameter $S$. This parameter $S$, which is theoretically ideal at $\sigma^2/\epsilon$, can be large, especially when target precision $\epsilon$ is high or gradient noise $\sigma^2$ is significant. This could potentially limit the efficiency of Malenia SGD in environments where frequent updates are crucial.
> > 3. *Handling of Delays:* The theoretical analysis of IA$^2$SGD requires handling of the sophisticated delayed stochastic gradient. This is in sharp contrast with the analysis of Melania SGD, as it does not explicitly address delays in its iterations.
> >
> > Paper [2], being very recently published, extends the model in [1] by incorporating considerations of communication time, thus presenting a more comprehensive analysis under conditions of both computation and communication heterogeneity.
> >
> > Our methodology and analysis differ significantly from those employed in these works; however, we recognize the importance of these differences and their relevance to our own research. In our revised manuscript, we have included a discussion of both papers in Appendix A.

---

> > > ### Author Response · Authors · 2024-11-20
> > >
> > > > It is not clear why the analysis needs two sequences $\tau_i(t)$ and $\rho_i(t)$. Isn't there a bijection between them? I guess it would be needed if we were allowed to assign a new job to a worker different from the one who finished the computations.
> > > >
> > >
> > > **Response:**
> > >
> > > The use of these two sequences helps to distinguish between different delays that occur in our asynchronous system. The superscript of the model represents the time the server generates the model, and the superscript of the data represents the time the server first utilizes the stochastic gradient. For example, suppose that worker $j$ send a stochastic gradient to the server in iteration $t$. We use $\nabla f_{j} ( \boldsymbol{w}^{t-\tau _ {j}(t)}; \boldsymbol{\xi} _ {j}^{t} )$ to denote this stochastic gradient, because the model $\boldsymbol{w}^{t-\tau _ {j}(t)}$ was created in a prior iteration $t-\tau _ {j}(t)$ while the data sample $\boldsymbol{\xi} _ {j}^{t}$ is first utilized by the server (the data sample is still fresh at time $t$).  Note that the data sample's timestamp $t$ does not inherently inform us of when the model $\boldsymbol{w}^{t-\tau_{j}(t)}$ was received by worker $j$, meaning that data delay does not imply a corresponding model delay.
> > >
> > > As the computation progresses and another worker sends its gradient at iteration $t+1$ , the previous gradient $\nabla f_{j} ( \boldsymbol{w}^{t-\tau_{j}(t)}; \boldsymbol{\xi}_{j}^{t} )$ remains as a component in $\boldsymbol{g}^{t+1}$, rendering both the model and data used by worker $j$ increasingly stale at time $t+1$.
> > >
> > > In our implementation, we ensure that a new job is assigned to the worker immediately after it completes its computations, minimizing its idle time. Although $\tau_i(t)$ exceeds $\rho_i(t)$ by at least one iteration (i.e.,  $\tau_i(t) \geq \rho_i(t) + 1$), it is important to note that $\tau_i(t)$ and $\rho_i(t)$ do not possess a bijective relationship.
> > >
> > > Figure 1 may provide us with a clear picture: Focusing on the server's iteration $t_0+4$, when the server receives a stochastic gradient from worker 1, it does not know when worker 1 started computing this gradient. Moreover, the aggregated stochastic gradient used in IA$^2$SGD is expressed as: $\boldsymbol{g}^{t_0+4} = \frac{1}{3} \left( \nabla f_1(\boldsymbol{w}^{t_0+2};\boldsymbol{\xi}_1^{t_0+4}) + \nabla f_2(\boldsymbol{w}^{t_0-1};\boldsymbol{\xi}_2^{t_0+1}) + \nabla f_3(\boldsymbol{w}^{t_0-2};\boldsymbol{\xi}_3^{t_0+3}) \right)$. Except for $\boldsymbol{\xi}_1^{t_0+4}$ from worker 1, all the models and data samples utilized within $\boldsymbol{g}^{t_0+4}$ are stale. Besides, the staleness of the model $\boldsymbol{w}$ and data sample $\boldsymbol{\xi}$ used by each worker are different, exhibiting dual delays.
> > >
> > > We have further clarified this in the revised manuscript to ensure that the roles and relationships $\tau_i(t)$ and $\rho_i(t)$ are clearly understood. Thank you for highlighting the need for further explanation.
> > >
> > > > Could the authors show that the proposed method can have a better dependency on $\tau_{\max}$ under stronger assumptions (e.g., bounded gradients)? Can we get both better dependency on $\tau_{\max}$ and no requirement to bound the data heterogeneity simultaneously?
> > > >
> > >
> > > **Response:**
> > >
> > > Currently, our analytical framework is structured around the worst-case scenario of delay captured by $\tau_{\max}$. This conservative approach ensures that our convergence guarantees hold despite the oldest information influencing the updates. Indeed, as specified in line 341, the crux of our analysis relies on ensuring that $\mathbb{E} \left\langle \nabla F(\boldsymbol{w}^{[ t - \tau _ {\max} ] _ +}), \boldsymbol{g}^t \right\rangle	= \frac{1}{n} \sum_{i=1}^n \mathbb{E} \langle \nabla F(\boldsymbol{w}^{[t - \tau _ {\max}]_+}) , \nabla F_i ( \boldsymbol{w}^{t-\tau _ i(t)} ) \rangle$. This property is critical for the validity of our proof and is maintained by leveraging a **conditional expectation** relative to the random data samples within $\boldsymbol{g}^t$.
> > >
> > > Introducing stronger assumptions could potentially refine our analysis and improve the dependency on $\tau_{\max}$. However, our current methodological framework, which prioritizes robustness against data heterogeneity, does not exploit this avenue. Nonetheless, we acknowledge that managing both unbounded data heterogeneity and unbounded delay concurrently is a worthy topic for future research.

---

> > > > ### Author Response · Authors · 2024-11-20
> > > >
> > > > > The first line in Figure 4 is not readable. It is hard to distinguish the variance of algorithms. Could the authors make them more readable? Could you please clarify why there is no variance in accuracy plots for some algorithms (FedBuff, Synchronous SGD, Uniform ASGD)? Why does the red shaded area (variance) around the red dashed line in Figure 4 (second line) look so weird? Sometimes red dashed line is not inside red shaded area.
> > > > >
> > > >
> > > > **Response:**
> > > >
> > > > The observed subtleties in the variance plots are attributed to the implementation nuances of our experiments. Specifically, we conduct multiple repetitions of the experiments using different random seeds. This variation affects the distribution of computation times among workers, which is controlled by the parameter $\textsf{std}$. are different across different experiments. As a result, the exact moments at which losses and accuracies are computed can vary across different experiments. This variation becomes more pronounced with an increase in the number of workers, as their computation times tend to diversify further. Consequently, for certain time units, there may be **only one recorded datum**, leading to the absence of variance.
> > > >
> > > > Regarding the red dashed line occasionally appearing outside the red shaded variance area, this anomaly was due to a disparity in how we sampled the mean and variance data. In the initial figures, we downsampled the mean data without correspondingly adjusting the variance data. In the revised figure, we have corrected this issue by ensuring consistent sampling for both mean and variance data, thus rectifying the visualization inconsistency. After modification, the first row of Figure 4 becomes more readable. In addition, the following table presents specific values of both the mean and variance of our experimental results across different time units:
> > > > | Time Units | Around 1000 | Around 2000 | Around 3000 | Around 4000 | Around 5000 |
> > > > | --- | --- | --- | --- | --- | --- |
> > > > | Synchronous SGD | (2.16,0.15) | (1.95,0.29) | (1.80,0.22) | (1.74,0.22) | (1.68,0.17) |
> > > > | MIFA | (2.20,0.02) | (1.91,0.01) | (1.77,0.04) | (1.66,0.02) | (1.70,0.02) |
> > > > | FedBuff | (1.61, 0.08) | (1.47, 0.0) | (1.49, 0.0) | (1.46, 0.0) | (1.38, 0.05) |
> > > > | Vanilla ASGD | (1.66,0.05) | (1.41,0.16) | (1.20,0.09) | (1.19,0.06) | (1.10,0.12) |
> > > > | Uniform SGD | (2.15,0.0) | (2.07,0.0) | (1.96,0.0) | (1.88,0.0) | (1.85,0.13) |
> > > > | Shuffled SGD | (2.26, 0.0) | (2.10, 0.0) | (1.92, 0.0) | (1.98, 0.0) | (1.88, 0.0) |
> > > > | IA$^2$SGD | (1.74, 0.27) | (1.65,0.05) | (1.41,0.23) | (1.31,0.0) | (1.28,0.11) |
> > > >
> > > >
> > > > We appreciate the feedback and will incorporate a more detailed discussion on these points in the revised manuscript.
> > > >
> > > > > I find it weird that uniform/shuffled ASGD performs worse than vanilla ASGD. Is there any particular reason for that since from theory vanilla ASGD converges to $\zeta^2$-neighborhood of the solution while uniform/shuffled ASGD should converge exactly?
> > > > >
> > > >
> > > > **Response:**
> > > >
> > > > Theoretically, it is indeed expected that uniform/shuffled ASGD should converge more accurately compared to the $\zeta^2$-neighborhood convergence of vanilla ASGD. However, the divergence in performance observed in our experiments can potentially be attributed to several practical aspects of these algorithms' implementation and execution environments.
> > > >
> > > > In the case of uniform ASGD, while the algorithm aims to distribute the model updates uniformly across all workers, the random sampling method can inadvertently lead to scenarios where some workers are selected repeatedly before others have completed processing their current tasks. This can result in a backlog or accumulation of model updates at certain workers. When these workers have queued updates, they may not incorporate the most recent global information into their calculations, potentially leading to outdated gradients being used in the optimization process. This inefficiency can detract from the algorithm's overall effectiveness.
> > > >
> > > > The shuffled ASGD shuffles the sampling order of workers after a specified number of iterations. However, this may also leads to increased idle time of faster workers and thus insufficient utilization of computational resources.
> > > >
> > > > Both Vanilla ASGD and our IA$^2$SGD employ a highly efficient worker scheduling strategy, wherein the server dispatches the updated model to the worker immediately after it has submitted its gradient. This approach minimizes the idle time of workers by ensuring that they are continuously engaged in processing tasks without unnecessary delays.

---

> > > > > ### Comment · Reviewer_nXSQ · 2024-11-20
> > > > > **Response to authors**
> > > > >
> > > > > > Theoretically, it is indeed expected that uniform/shuffled ASGD should converge more accurately compared to the
> > > > >  neighborhood convergence of vanilla ASGD. However, the divergence in performance observed in our experiments can potentially be attributed to several practical aspects of these algorithms' implementation and execution environments.
> > > > >
> > > > > The proposed example might work if one runs experiments only once. However, if one averages the results across multiple runs uniform/shuffled ASGD should outperform vanilla ASGD. Can the reason for such empirical results be in the untuned stepsizes? Could you provide the tuning details that you used in the experiments section?

---

> > > > > ### Comment · Reviewer_nXSQ · 2024-11-20
> > > > > **Response to authors**
> > > > >
> > > > > > The observed subtleties in the variance plots are attributed to the implementation nuances of our experiments...
> > > > >
> > > > > Could you please clarify which results are presented in the table? Which of the plots in Figure 3 corresponds to the table? It seems none of them because shuffled ASGD has the loss variance at time 1000 while in the table it has zero variance. Is there a mistake somewhere?

---

> > ### Comment · Reviewer_nXSQ · 2024-11-20
> > **Response to authors**
> >
> > > Theorem 1 states that as long as the slowest worker respond once no more than $\tau_{\max}$ iterations, the algorithm converges as $T \to \infty$. Here, $\tau_{\max}$ is treated as a constant that does not scale with $T$.
> >
> > In practice, the maximum delay can scale with the number of iterations $T$ (e.g., some workers might be dead). Therefore, it is crucial to have the dependency on $\tau_{\max}$ as tight as possible, since in the extreme cases, $\tau_{\max} \sim T$. The current theory of the proposed method suggests that in such extreme cases would not converge even for homogeneous while methods like random/shuffled ASGD converge.
> >
> > As reviewer f1eX mentioned, there is a trade-off between the bounded heterogeneity and the maximum delay. Therefore, I encourage the authors to analyze the proposed algorithm under bounded heterogeneity to demonstrate possible better dependency on $\tau_{\max}$.

---

> ### Author Response · Authors · 2024-11-20
>
> > The rate of the proposed algorithm in the table looks weird: why is there $\sigma$ in the denominator of the second term?
> >
>
> **Response:**
>
> Thank you for your careful reading. In Table 1, the appearance of $\sigma$ in the denominator of the second term is an artifact of our analysis when choosing step size $\eta$. Indeed, substituting $\eta=\frac{1}{2}\sqrt{\frac{n\Delta}{L\sigma^2\tau_{\max}T}}$ into inequality (35) yields the rate of $\mathcal{O}\left(\sqrt{\frac{\sigma^2\tau_{\max}}{nT}} + \frac{\sqrt{n\tau_{\max}}}{\sigma T^{3/2}}\right)$ in the last line of Table 1, which does not exactly align with the rate $\mathcal{O}\left( (1+\sigma^2)\sqrt{\frac{\tau_{\max}}{nT}} + \frac{\sigma^2\sqrt{n\tau_{\max}}}{T^{3/2}}\right)$ in Theorem 1 (referred to as Corollary 1 in the revised manuscript).
> In our revised manuscript, we have unified the results presented in Table 1 and the main text.
>
> > Why is the accuracy so low? All algorithms barely achieved 50% accuracy which I found low for the CIFAR-10 dataset. Could you present the results when the algorithms' accuracy stops increasing after a sufficient amount of ''time''?
> >
>
> **Response:**
>
> The relatively low accuracy on the CIFAR-10 dataset primarily stems from the simplicity of the model used in our experiments—a 2-layer CNN. This is not due to insufficient runtime of the algorithms.
>
> The 2-layer CNN, due to its limited capacity, struggles to achieve high accuracy on a complex and varied dataset like CIFAR-10. Numerical results from a similar-scale model architecture in [3] show CIFAR-10 accuracies ranging approximately from 46% to 70%, achieved using different data partitioning schemes and algorithms.
>
> High-performing models on this dataset typically involve deeper architectures with more sophisticated feature extraction capabilities. Unfortunately, we are limited by computing resources to test our algorithms on more sophisticated models.
>
> However, the primary focus of our study is not to achieve state-of-the-art accuracy on CIFAR-10, but rather to analyze and compare the convergence behavior of different optimization algorithms under the constraints of a fixed model architecture. We are particularly interested in how quick the loss decreases and accuracy increases in each algorithm, even within the limits of a simpler neural network.
>
> [3] Li, Qinbin, et al. "Federated learning on non-iid data silos: An experimental study." 2022 IEEE 38th international conference on data engineering (ICDE). IEEE, 2022.
>
> **Response:**
>
> > Could the authors provide the empirical comparison against Melania SGD?
> >
>
> Thank you for your suggestion. We have implemented Melania SGD with a setting of $S=10$, as our empirical observations indicated that larger values of $S$ tend to reduce the server's update frequency and degrade convergence. The numerical results have been incorporated into the updated version of Figure in our manuscript.

---

> ### Author Response · Authors · 2024-11-21
>
> >In practice, the maximum delay can scale with the number of iterations $T$ (e.g., some workers might be dead). Therefore, it is crucial to have the dependency on $\tau_{\max}$ as tight as possible, since in the extreme cases, $\tau_{\max} \sim T$. The current theory of the proposed method suggests that in such extreme cases would not converge even for homogeneous while methods like random/shuffled ASGD converge.
> >
> >As reviewer f1eX mentioned, there is a trade-off between the bounded heterogeneity and the maximum delay. Therefore, I encourage the authors to analyze the proposed algorithm under bounded heterogeneity to demonstrate possible better dependency on $\tau_{\max}$.
>
> Thank you for your response.
>
> In the scenarios described by the reviewer, where some workers may only respond once ($\tau_{\max} = T$) or are even dead/inactive throughout the training, it is crucial to note that no existing asynchronous algorithms can guarantee convergence. This stems from the inherent requirement in optimization that every part of the function $\frac{1}{n} \sum_{i=1}^{n} F_i(\boldsymbol{w})$ must be accessible and contributive more than once for effective minimization.
>
> For uniform ASGD, as illustrated in Table 1 and discussed in Koloskova et al. (2022, Page 22), the convergence rate is contingent upon a step size upper bound of $\eta \leq \frac{1}{\sqrt{4L\tau_C\tau_{\max}}}$. Similar to the derivations in our paper, this condition implies a necessary lower bound on the number of iterations, specifically $T \geq \Omega(\tau_{C}\tau_{\max})$, which is not viable if $T$ is equivalent to $\tau_{\max}$.
>
> Similarly, the shuffled ASGD algorithm outlined by Islamov et al. (2024) assumes “*all participating workers $[n]$ are active in the training”* (Islamov et al. 2024, Page 5) and prescribes a similar step size constraint: $\eta \leq \frac{1}{20L\sqrt{\tau_C\tau_{\max}}}$ (Islamov et al. 2024, Theorem 1). This also implies a similar lower bound on $T$.
>
> Even Melania SGD, as highlighted by the reviewer, requires each worker to compute at least one stochastic gradient before each server update due to its inner-loop criterion $\left( \frac{1}{n} \sum_{i=1}^{n} \frac{1}{B_i} \right)^{-1} < \frac{S}{n}$.
>
> While there have been numerous studies on ASGD, the *unique* contribution of IA$^2$SGD is that it is the first ASGD algorithm to provably converge without the constraints of bounded data heterogeneity and specific computation models. This advancement significantly contributes to the existing literature. Addressing the challenge of improving delay dependency remains as be an open question for future research.
>
> References:
>
> - Koloskova, Anastasiia, Sebastian U. Stich, and Martin Jaggi. "Sharper convergence guarantees for asynchronous SGD for distributed and federated learning." *Advances in Neural Information Processing Systems* 35 (2022): 17202-17215.
> - Islamov, Rustem, Mher Safaryan, and Dan Alistarh. "AsGrad: A sharp unified analysis of asynchronous-SGD algorithms." *International Conference on Artificial Intelligence and Statistics*. PMLR, 2024.

---

> ### Author Response · Authors · 2024-11-21
>
> >The proposed example might work if one runs experiments only once. However, if one averages the results across multiple runs uniform/shuffled ASGD should outperform vanilla ASGD. Can the reason for such empirical results be in the untuned stepsizes? Could you provide the tuning details that you used in the experiments section?
>
> Thank you for your question.
>
> In our paper, we conducted each experiment three times and reported both the average results and variance in the figures. As detailed in our paper, we tuned the step sizes across a set of values: ${0.001, 0.005, 0.01}$.

---

> ### Author Response · Authors · 2024-11-21
>
> >Could you please clarify which results are presented in the table? Which of the plots in Figure 3 corresponds to the table? It seems none of them because shuffled ASGD has the loss variance at time 1000 while in the table it has zero variance. Is there a mistake somewhere?
>
>
>
> Thank you for your question.
>
> The Table in our response corresponds to the rightmost subfigure in the 1st row of Figure 4. In your initial review, you asked about the absence of variance in some plots in the original Figure 4, thus we have provided this table to clearly present the data for your reference.

---

> ### Comment · Reviewer_nXSQ · 2024-11-22
> **Response to authors**
>
> Of course, the workers must appear in the optimization process since we do not receive any information about the corresponding local function otherwise. My example was to show that $\tau_{\max}$ can scale with $T$.
>
> I am not sure if I understood the reason why the stepsize restriction $\eta \le \frac{1}{4L\sqrt{\tau\_C\tau\_{\max}}}$ (you have a typo in your response) implies that $T \ge \Omega(\tau_C\tau_{\max})$. Could the authors elaborate more on this point?
>
> The convergence rate of uniform ASGD is
> $$\frac{1}{T}\sum_{t=0}^{T-1}\mathbb{E}[\\|\nabla f(x^t)\\|^2] \le \frac{8F^0}{\eta T} + 6L\eta\sigma^2 + 8L\eta\zeta^2 + 8L^2\eta^2\tau_C\frac{1}{n}\sum_{j=1}^n\zeta_j^2\bar{\tau}_j.$$
> To get the rate from Theorem 11 (Koloskova et al., 2022) we need to choose sufficiently small stepsize $\eta$ such that
> $$\eta \le \min\left\\{\frac{1}{4L}, \frac{1}{4L\sqrt{\tau\_C\tau\_{\max}}}, \left(\frac{4F^0}{3L\sigma^2T}\right)^{1/2}, \left(\frac{F^0}{L\zeta^2T}\right)^{1/2}, \left(\frac{F^0}{L^2T\tau_C\frac{1}{n}\sum\_{j=1}^n \zeta_j^2\bar{\tau}_j}\right)^{1/3} \right\\}.$$
> Choosing such $\eta$ gives the rate
> $$\frac{1}{T}\sum\_{t=0}^{T-1}\mathbb{E}[\\|\nabla f(x^t)\\|^2]  \le \mathcal{O}\left(\frac{LF^0\sqrt{\tau\_C\tau\_{\max}}}{T} + \left(\frac{LF^0\sigma^2}{T}\right)^{1/2} + \left(\frac{LF^0\zeta^2}{T}\right)^{1/2} + \left(\frac{LF^0\sqrt{\tau\_C}\sqrt{\frac{1}{n}\sum\_{j=1}^n\zeta\_j^2\bar{\tau}\_j} }{T}\right)^{2/3} \right).$$
> It remains to use the fact that if the concurrency $\tau_C$ if fixed, then $\tau\_{\mathrm{avg}} = \Theta(\tau\_{C})$. One can translate this rate in terms of $\varepsilon$ to get the rate in (14) in (Koloskova et al., 2022). These derivations do not use any lower bound for $T$; rather, the stepsize should be sufficiently small. I also didn't find any discussion on page 22 (Koloskova et al., 2022). Similarly, the derivations in (Islamov et al., 2024) have neither. As I mentioned in my original review I believe that there is no lower bound for the proposed algorithm.
>
> But importantly, the maximum delay comes as $\sqrt{\tau\_{\max}}$ in the rate which is in the extreme case $\tau_{\max} \approx T$ gives the convergence.

---

> > ### Author Response · Authors · 2024-11-23
> >
> > Thanks a lot for your question. The uniform ASGD paper does not detail their stepsize selection method, and we appreciate the reviewer for providing clarification. In our previous response, we applied the stepsize selection approach used in our paper to uniform ASGD, which differs slightly from what the reviewer described. Specifically, we chose $\eta = \sqrt{\frac{8F^0}{(6L\sigma^2+8L\zeta^2)T}}$, which *minimizes* the dominant terms $\frac{8 F^0}{\eta T} + 6 L \eta \sigma^2 + 8L\eta\zeta^2$ in the convergence bound of uniform ASGD. This yields the following rate:
> > $$
> > \frac{1}{T} \sum_{t=0}^{T-1} \mathbb{E} || \nabla f ({x}^{t}) || _ 2^2 \leq \frac{8 F^0}{\eta T} + 6 L \eta \sigma^2 + 8L\eta\zeta^2 + 8 L^2 \eta^2 \tau _ {C} \frac{1}{n} \sum_{j=1}^{n} \zeta _ j^2 \bar{\tau} _ j
> > = \sqrt{\frac{8F^0(6L\sigma^2+8L\zeta^2)}{T}} + \frac{64 F^0 L^2 \tau _ {C} \frac{1}{n} \sum_{j=1}^{n} \zeta _ j^2 \bar{\tau} _ j}{(6L\sigma^2+8L\zeta^2)T}.
> > $$
> > This rate is faster than the one mentioned in your comment as the non-dominant term is improved from $\mathcal{O}\left(\frac{1}{T^{2/3}}\right)$ to $\mathcal{O}\left(\frac{1}{T}\right)$. Note that this method imposes a lower bound on $T$, because the step size condition $\eta = \sqrt{\frac{8F^0}{(6L\sigma^2+8L\zeta^2)T}} \leq \frac{1}{4L \sqrt{\tau_{C} \tau_{\max}}}$ implies that $T \geq \Omega(\tau_{C} \tau_{\max})$.
> >
> > Similarly, applying the step size selection approach described by the reviewer to our Theorem 1, we can also eliminate the lower bound on $T$, though this somewhat degrades the non-dominant rate. Specifically, our Theorem 1 provides the convergence bound:
> > $$
> > \frac{1}{T} \sum_{t=1}^{T} \mathbb{E} || \nabla F ({w}^{t-1}) || _ 2^2 \leq \frac{32 \Delta}{T\eta}  + 128 L \tau _ {\max} \eta \frac{\sigma^2}{n} + 128 L^3 \tau _ {\max}^2 \eta^3 \frac{\sigma^2}{n}.
> > $$
> > Choosing the following stepsize:
> > $$
> > \eta = \min \left \\{  \frac{1}{64L\tau_{\max}}, \frac{1}{2} \left(\frac{n \Delta}{L\sigma^2 \tau_{\max} T}\right)^{1/2}, \left( \frac{n \Delta}{4 L^3 \sigma^2 \tau_{\max}^2 T} \right)^{1/4} \right \\},
> > $$
> > we can achieve:
> > $$
> > \frac{1}{T} \sum_{t=1}^{T} \mathbb{E} || \nabla F ({w}^{t-1}) ||_2^2 \leq \mathcal{O} \left( \frac{L \Delta \tau _ {\max}}{T} + \left(\frac{L \Delta \sigma^2 \tau _ {\max}}{nT}\right)^{1/2} + \left(\frac{\sqrt[3]{L^3 \sigma^2 \tau _ {\max}^2} \Delta}{\sqrt[3]{n}T}\right)^{3/4} \right)
> > $$
> > for all $T \geq 1$. However, the non-dominant term degrades from $\mathcal{O}(1/T^{\frac{3}{2}})$ presented in our Corollary 1 to $\mathcal{O}(1/T^{\frac{3}{4}})$.
> >
> > We acknowledge that the dependence on $\tau_{\max}$ in our convergence rate presents a limitation, which we have discussed in our conclusion section. Nevertheless, our theoretical results could be the best currently known achievements for this particular research problem.

---

### Official Review · Reviewer_f1eX · 2024-10-29

**Soundness:** 2
**Presentation:** 3
**Contribution:** 1
**Rating:** 3
**Confidence:** 3

**Summary:**

The authors address a distributed learning problem with smooth nonconvex objective function under a central server orchestration, and introduce the Incremental Aggregated Asynchronous SGD (IA2SGD) algorithm. This approach uses a server-side buffer to utilize stale stochastic gradients from all workers, aiming to mitigate the negative impact of data heterogeneity. The algorithm maintains a per-iteration computational cost similar to traditional asynchronous SGD algorithms and has a convergence rate independent of the degree of data heterogeneity, distinguishing it from prior asynchronous SGD methods that often impose a bounded dissimilarity condition. Numerical experiments show that IA2SGD can outperform both asynchronous and synchronous SGD-based methods.

**Strengths:**

- The problem of distributed learning with heterogeneous data has been extensively studied, but IA2SGD appears to be a novel algorithmic approach.

- The paper is well-written, with clear visuals. However, some explanations could benefit from more technical precision (see Weaknesses).

- The problem addressed is practically relevant, especially given the challenges in distributed learning scenarios with heterogeneous client data and variable computation times. An important strength of this work is that its theoretical results are independent of data heterogeneity.

**Weaknesses:**

- The convergence result in Theorem 1 appears impractical. The stepsize $\eta = \frac{1}{2} \sqrt{\frac{n (F(w^0) - F^*)}{L \tau_{\max} T}}$ relies on unknown quantities like $F(w^0) - F^*$, $L$, and $\tau_{\max}$, and hence needs substantial tuning. I acknowledge that this is the case for many gradient-based algorithms, and hence is not a critical flaw. However, Theorem 1 contains a lower bound on $T$, requiring $T \geq 1024 L (F(w^0) - F^*) n \tau_{\max}$ regardless of the desired accuracy, making the theory seem weak.

- The paper asserts that previous asynchronous methods require bounded data heterogeneity. However, this is not entirely accurate, as the work in [1] addresses similar optimization problem without assuming data similarity. The method in [1] achieves optimal time complexity with smoothness, unbiasedness, and bounded gradient variance assumptions. Given these theoretical results, a comparison between IA2SGD and the Malenia SGD method from [1] is essential in both theoretical and practical contexts.

- The reliance on a maximum delay assumption limits the algorithm’s applicability. In practical federated learning, some clients may become unavailable, making delay bounds infinite. Some methods do relax the bounded delay assumption (e.g., [2], [3]), and instead assume bounded dissimilarity of local objective functions. The paper does not discuss the trade-off between these assumptions; namely, that one can either assume bounded data dissimilarity with arbitrary delays or bounded delays with unbounded heterogeneity. Either approach has limitations, and I do not think that one of these assumptions is superior to the other. This trade-off should be addressed more thoroughly.

- The algorithm requires additional memory allocation for a $d$-dimensional vector at both the server and each worker. While server memory constraints may not be an issue, this could limit the method’s applicability for memory-constrained clients.

[1] Tyurin, Alexander, and Peter Richtárik. "Optimal time complexities of parallel stochastic optimization methods under a fixed computation model." Advances in Neural Information Processing Systems 36 (2024).

[2] Mishchenko, Konstantin, et al. "Asynchronous SGD beats minibatch SGD under arbitrary delays." Advances in Neural Information Processing Systems 35 (2022): 420-433.

[3] Islamov, Rustem, Mher Safaryan, and Dan Alistarh. "AsGrad: A sharp unified analysis of asynchronous-SGD algorithms." International Conference on Artificial Intelligence and Statistics. PMLR, 2024.

**Questions:**

- Could the authors elaborate on the points mentioned in the Weaknesses?

- What is the origin of the dependence on $\sigma$ in the last line of Table 1, as it does not appear to follow from Theorem 1?

- Could the authors clarify why reusing outdated stochastic gradients would improve convergence rates? Intuitively, one might expect performance to degrade with increasingly stale updates.

- It is not entirely clear to me why the authors highlight the "dual-delayedness" of the updates. It seems that sample delays are already present in standard ASGD, as samples are processed after each client completes a job, creating a delay similar to that in IA2SGD. Overall, the term "dual delay" seems misleading since workers are processing freshly sampled data points.

- In lines 108-109, the authors mention that "$\xi_i^t \sim \mathbb{P}_i$ is indexed by $t$ to indicate that this particular data sample has not been utilized by the server prior to iteration $t$". Doesn't this imply single-pass data usage?

- Line 114 states that without delays, $\tau_i(t)=1$ for all $i$, implying each stochastic gradient is evaluated at the most recent model parameters. However, this would require restarts of client computations, which seems to contradict asynchronous operation.

---

> ### Author Response · Authors · 2024-11-20
>
> > The convergence result in Theorem 1 appears impractical. The stepsize $\eta = \frac{1}{2}\sqrt{\frac{n(F(w^0)-F^*)}{L\tau_{\max}T}}$ relies on unknown quantities like $F(w^0)-F^*$, $L$, and $\tau_{\max}$, and hence needs substantial tuning. I acknowledge that this is the case for many gradient-based algorithms, and hence is not a critical flaw.
> >
>
> **Response:** Your observation is correct and reflects a common challenge to obtain tight convergence bounds of gradient-based optimization methods. Actually, the formula for the stepsize $\eta$ is derived to optimize the theoretical convergence rate given in our general bound in Appendix B.3:
>
> $$
> \frac{1}{T} \sum_{t=1}^{T} \mathbb{E} \| \nabla F (\boldsymbol{w}^{t-1}) \|_2^2 \leq \frac{32 (F(\boldsymbol{w}^0) - F^*)}{T\eta}  + 128 L \tau _{\max} \eta \frac{\sigma^2}{n}  + 128 L^3 \tau _{\max}^2 \eta^3 \frac{\sigma^2}{n}.
> $$
>
> It is reasonable to set the stepsize as $\eta = \sqrt{n/T}$ in the right-hand-side of the above inequality, which does not rely on any unknown quatities. However, this yields a rate of $\mathcal{O}\left(\frac{\Delta+L\tau_{\max}\sigma^2}{\sqrt{nT}} \right)$, which is suboptimal in terms of the constant factor. Hence, we take $\eta = \frac{1}{2}\sqrt{\frac{n(F(w^0)-F^*)}{L\tau_{\max}T}}$ to obtain a tighter convergence bound $\mathcal{O}\left( (1+\sigma^2)\sqrt{\frac{L \Delta\tau_{\max}}{nT}} \right)$.
>
> Thank you for your comment and we have revised paper to display about this more "practical step size" choice.
>
> > Theorem 1 contains a lower bound on $T$, requiring $T≥1024L(F(w^0)−F^∗)n\tau_{\max}$ regardless of the desired accuracy, making the theory seem weak.
> >
>
> **Response:**
>
> The lower bound on $T$ specified in our analysis is indeed essential for ensuring the validity of the convergence rate and the step size condition $\eta \leq \mathcal{O}(1/L\tau_{\max})$.  This bound emerges specifically when substituting $\eta = \frac{1}{2}\sqrt{\frac{n(F(w^0)-F^*)}{L\tau_{\max}T}}$ into the inequality in our last response.
>
> The requirement for $T$ to be sufficiently large is a standard practice in the related papers to accommodate the effects of delayed gradient updates on convergence. For instance, Lian et al. [4] necessitate $T = \Omega(L\Delta n\tau_{\max})$ to ensure convergence under homogeneous settings, Gu et al. [5] establish $T = \Omega(L\Delta n\tau_{\max}^2)$ in their exploration of federated learning environments with arbitrary device availability, and Koloskova et al. [6] also implicitly requires $T =\Omega(L\Delta \tau_{C}^2)$  to guarantee convergence of their uniform ASGD.
>
> [4] Xiangru Lian, Yijun Huang, Yuncheng Li, and Ji Liu. Asynchronous parallel stochastic gradient for nonconvex optimization. In Advances in Neural Information Processing Systems 28, 2015.
>
> [5] Xinran Gu, Kaixuan Huang, Jingzhao Zhang, and Longbo Huang. Fast federated learning in the presence of arbitrary device unavailability. In Advances in Neural Information Processing Systems 34, pp. 12052–12064, 2021.
>
> [6] Anastasiia Koloskova, Sebastian U Stich, and Martin Jaggi. Sharper convergence guarantees for asynchronous SGD for distributed and federated learning. In Advances in Neural Information Processing Systems 35, pp. 17202–17215, 2022.

---

> > ### Author Response · Authors · 2024-11-20
> >
> > > The paper asserts that previous asynchronous methods require bounded data heterogeneity. However, this is not entirely accurate, as the work in [1] addresses similar optimization problem without assuming data similarity. The method in [1] achieves optimal time complexity with smoothness, unbiasedness, and bounded gradient variance assumptions. Given these theoretical results, a comparison between IA$^2$SGD and the Malenia SGD method from [1] is essential in both theoretical and practical contexts.
> > [1] Tyurin, Alexander, and Peter Richtárik. "Optimal time complexities of parallel stochastic optimization methods under a fixed computation model." Advances in Neural Information Processing Systems 36 (2024).
> > >
> >
> > **Response:**
> >
> > We appreciate the reviewer's reference to the work on Malenia SGD. Here, we clarify the distinctions between our IA$^2$SGD and Malenia SGD to emphasize the unique aspects and applicability of our approach:
> >
> > 1. *Computation Model Assumptions:* The Melania SGD in [1] achieves optimal time complexity under a fixed-computation model, which assumes uniform and predictable computation times across all workers. Despite the theoretical advancement in this paper, it might not always be practical in real-world scenarios where computational capabilities can vary significantly. In contrast, our IA$^2$SGD does not rely on such specific computation models, making it more flexible and applicable to a wider range of real-world conditions where computation time and communication delays can vary.
> > 2. *Asynchrony Mechanisms:* The asynchrony mechanism in IA$^2$SGD involves a **greedy** global update scheme, where the server updates its parameters whenever any stochastic gradient is received, regardless of the worker's delay. This approach contrasts with Malenia SGD, where the server updates are conditional on achieving a global criteria $\left( \frac{1}{n} \sum_{i=1}^{n} \frac{1}{B_i} \right)^{-1} < \frac{S}{n}$ based on the computation of $B_i$ gradients on worker $i$ and a system-wide parameter $S$. This parameter $S$, which is theoretically ideal at $\sigma^2/\epsilon$, can be large, especially when target precision $\epsilon$ is high or gradient noise $\sigma^2$ is significant. This could potentially limit the efficiency of Malenia SGD in environments where frequent updates are crucial.
> > 3. *Handling of Delays:* The theoretical analysis of IA$^2$SGD requires handling of the sophisticated delayed stochastic gradient. This is in sharp contrast with the analysis of Melania SGD, as it does not explicitly address delays in its iterations.
> >
> > Additionally, we have implemented Melania SGD with $S=10$ (see the revised Figure 3), as our empirical observations indicated that larger values of $S$ tend to reduce the server's update frequency and degrade convergence. The numerical results have been incorporated into the updated version of Figure in our manuscript.
> >
> > To conclude, while the Malenia SGD method presents an optimal solution under specific theoretical conditions, IA$^2$SGD offers a robust and generally applicable solution in practical, heterogeneous environments. We acknowledge the strengths of the Malenia SGD model in controlled settings and will include a detailed comparison in the revised manuscript.

---

> > > ### Author Response · Authors · 2024-11-20
> > >
> > > > The reliance on a maximum delay assumption limits the algorithm’s applicability. In practical federated learning, some clients may become unavailable, making delay bounds infinite. Some methods do relax the bounded delay assumption (e.g., [2], [3]), and instead assume bounded dissimilarity of local objective functions. The paper does not discuss the trade-off between these assumptions; namely, that one can either assume bounded data dissimilarity with arbitrary delays or bounded delays with unbounded heterogeneity. Either approach has limitations, and I do not think that one of these assumptions is superior to the other. This trade-off should be addressed more thoroughly.
> > > > [2] Mishchenko, Konstantin, et al. "Asynchronous SGD beats minibatch SGD under arbitrary delays." Advances in Neural Information Processing Systems 35 (2022): 420-433.
> > > > [3] Islamov, Rustem, Mher Safaryan, and Dan Alistarh. "AsGrad: A sharp unified analysis of asynchronous-SGD algorithms." International Conference on Artificial Intelligence and Statistics. PMLR, 2024.
> > >
> > > **Response:**
> > >
> > > While we recognize the importance of delay issues in ASGD, as detailed in the cited works [2] and [3], our approach was to focus more comprehensively on data heterogeneity. This choice was driven by our specific research questions and does not imply a general superiority of one set of assumptions over another.
> > >
> > > In fact, we do NOT claim that our algorithm is better than other algorithms in all senarios. As indicated in our results (Figure 3), IA$^2$SGD performs comparably to other algorithms under conditions of low data heterogeneity ($\alpha=0.5$). However, it excels in environments characterized by high data heterogeneity ($\alpha=0.1$).
> > >
> > > Whether our algorithmic framework is able to achieve both unbounded data heterogeneity and unbounded delay is an interesting issue that is worth further exploring. We thank the reviewer’s comment and we have included further dicussion on these points in the revised version of our manuscript.
> > >
> > > > The algorithm requires additional memory allocation for a dimensional vector at both the server and each worker. While server memory constraints may not be an issue, this could limit the method’s applicability for memory-constrained clients.
> > > >
> > >
> > > **Response:**
> > >
> > > We appreciate the reviewer's concern regarding the memory requirements of our algorithm, particularly in the context of memory-constrained clients. The additional $\mathcal{O}(d)$ memory requirement per worker, indeed, does not scale with the number of workers or the mini-batch size, which generally keeps the memory overhead manageable. However, we acknowledge that this could still be a limitation in environments where clients have extremely restricted memory capabilities.
> > >
> > > To address such scenarios, our algorithm design includes provisions for flexible implementation strategies that can accommodate varying memory capacities among the workers. For example, using the system architecture illustrated in Figure 2, we can adopt the following **server-side memory management** strategy:
> > >
> > > For a memory-restricted worker (e.g., worker 2), we can adapt the system such that the server temporarily stores the previous gradient $\boldsymbol{G}_2^{t-1}$. When worker 2 computes the new gradient $\boldsymbol{G}_2^t$, it only needs to send this new gradient to the server. The server then computes the difference $\boldsymbol{\delta}^t = \boldsymbol{G}_2^t - \boldsymbol{G}_2^{t-1}$, reducing the memory load on worker 2.
> > >
> > > These modifications illustrate our algorithm's adaptability to diverse system architectures and constraints, emphasizing that the implementation can be tailored to fit the specific needs and limitations of different practical systems. In the revised manuscript, we will expand on these potential modifications to demonstrate how our algorithm can be effectively implemented even in highly memory-constrained environments.
> > >
> > > > What is the origin of the dependence on $\sigma$ in the last line of Table 1, as it does not appear to follow from Theorem 1?
> > > >
> > >
> > > **Response:** Thank you for your attentive review. In Table 1, the dependence on $\sigma$ follows from an artifact of our analysis when choosing step size $\eta$. Indeed, substituting $\eta=\frac{1}{2}\sqrt{\frac{n\Delta}{L\sigma^2\tau_{\max}T}}$ into inequality (35) yields the rate of $\mathcal{O}\left(\sqrt{\frac{\sigma^2\tau_{\max}}{nT}} + \frac{\sqrt{n\tau_{\max}}}{\sigma T^{3/2}}\right)$ in the last line of Table 1, which does not exactly align with the rate $\mathcal{O}\left( (1+\sigma^2)\sqrt{\frac{\tau_{\max}}{nT}} + \frac{\sigma^2\sqrt{n\tau_{\max}}}{T^{3/2}}\right)$ in Theorem 1 (referred to as Corollary 1 in the revised manuscript).
> > > In our revised manuscript, we have unified the results presented in Table 1 and the main text.

---

> > > > ### Author Response · Authors · 2024-11-20
> > > >
> > > > > Could the authors clarify why reusing outdated stochastic gradients would improve convergence rates? Intuitively, one might expect performance to degrade with increasingly stale updates.
> > > > >
> > > >
> > > > **Response:**
> > > > The convergence rate of IA$^2$SGD benefits from integrating the **full aggregation** technique with reusing outdated stochastic gradients.
> > > >
> > > > In traditional ASGD, the global model update frequency from different workers varies—workers that compute faster contribute more updates, potentially leading to a bias in the model towards the data characteristics of faster workers. This is particularly problematic when local data distributions are heterogeneous, as it can skew the convergence of the model and necessitate assumptions about bounded data heterogeneity to control this bias.
> > > >
> > > > Our IA$^2$SGD tackles this issue by incorporating outdated stochastic gradients from **all** workers in every iteration, similar to a synchronous aggregation model. This method draws on the principle that, in synchronous SGD, the gradient at each iteration is the average of gradients computed concurrently across all workers: $\frac{1}{n} \sum_{i=1}^{n} \nabla f_i(\boldsymbol{w}^{t-1};\boldsymbol{\xi}_i^t)$. This full aggregation naturally mitigates the bias caused by data heterogeneity among workers and does not require the assumption of bounded heterogeneity.
> > > >
> > > > In IA$^2$SGD, the term $\boldsymbol{g}^t$ serves as an approximation of this synchronous gradient. By ensuring that every worker, regardless of their speed, contributes to the global update within a bounded delay ($\tau_{\max}$ iterations), we control the approximation error of $\boldsymbol{g}^t$. This controlled approximation allows us to effectively balance between the benefits of asynchrony and the robustness of synchronous updates, improving convergence rates despite the use of potentially outdated gradients.
> > > >
> > > > We acknowledge that this is a nuanced aspect of our approach and appreciate the opportunity to clarify this in our manuscript. We have enhanced our discussion on this topic in the revised version of our paper.
> > > >
> > > > > It is not entirely clear to me why the authors highlight the "dual-delayedness" of the updates. It seems that sample delays are already present in standard ASGD, as samples are processed after each client completes a job, creating a delay similar to that in IA$^2$SGD. Overall, the term "dual delay" seems misleading since workers are processing freshly sampled data points. In lines 108-109, the authors mention that "$\xi_i^t∼\mathbb{P}_i$ is indexed by $t$ to indicate that this particular data sample has not been utilized by the server prior to iteration $t$". Doesn't this imply single-pass data usage?
> > > > >
> > > >
> > > > **Response:**
> > > >
> > > > Thank you for your comment, which highlights a need for clearer explanation regarding the term "dual-delayedness" in our manuscript.
> > > >
> > > > In standard ASGD, there is typically a focus on the model delay. However, in our IA$^2$SGD, we explicitly consider both the delay of the model used to compute the gradient and the delay of the data samples in the computation process. This dual consideration is crucial for **rigorously characterizing asynchrony** in our system.
> > > >
> > > > The superscripts of the model and data represent the first time the server “sees” them. To clarify, we suppose that worker $j$ send a stochastic gradient to the server in iteration $t$. We use $\nabla f_{j} ( \boldsymbol{w}^{t-\tau _ {j}(t)}; \boldsymbol{\xi} _ {j}^{t} )$ to denote this stochastic gradient, because the model $\boldsymbol{w}^{t-\tau_{j}(t)}$ was created in a prior iteration $t-\tau_{j}(t)$ while the data sample $\boldsymbol{\xi} _ {j}^{t}$ is first utilized by the server (the data sample is still fresh at time $t$).  Note that the data sample's timestamp $t$ does not inherently inform us of when the model $\boldsymbol{w}^{t-\tau_{j}(t)}$ was received by worker $j$, meaning that zero data delay does not imply a corresponding model delay. As the computation progresses and another worker sends its gradient at iteration $t+1$ , the previous gradient $\nabla f_{j} ( \boldsymbol{w}^{t-\tau_{j}(t)}; \boldsymbol{\xi}_{j}^{t} )$ remains as a component in $\boldsymbol{g}^{t+1}$, rendering both the model and data used by worker $j$ **increasingly stale** at time $t+1$.
> > > >
> > > > The reviewer may wonder what if we represent the model and data in each stochastic gradient with the same delay, i.e., $\boldsymbol{g}^t = \frac{1}{n} \sum_{i=1}^{n} \nabla f_i ( \boldsymbol{w}^{t-\tau_i(t)}; \boldsymbol{\xi}_i^{t-\tau_i(t)} )$. However, this means that a worker’s data sample $\boldsymbol{\xi}_j^{t-\tau_j(t)}$ was seen by the sever **immediately** after $\boldsymbol{w}^{t-\tau_j(t)}$ was created in iteration $t-\tau_i(t)$. Such an algorithm **cannot** operate in an asynchronous manner.

---

> > > > > ### Author Response · Authors · 2024-11-20
> > > > >
> > > > > > Line 114 states that without delays, $\tau_i(t)=1$ for all $i$, implying each stochastic gradient is evaluated at the most recent model parameters. However, this would require restarts of client computations, which seems to contradict asynchronous operation.
> > > > > >
> > > > >
> > > > > **Response:**
> > > > >
> > > > > The reviewer may misunderstand our statement. Our statement aims to establish an intuitive baseline for understanding the effects of data heterogeneity. Specifically, we discuss a hypothetical scenario where the algorithm operates synchronously—that is, without delays in gradient computation and update ($\tau_i(t) = 1$ for all $i$). This scenario serves to highlight that even under such an simpler condition, non-uniform participation by workers can lead to biased gradient estimates solely due to data heterogeneity.
> > > > >
> > > > > When we transition to asynchronous operation, as in ASGD, the situation becomes more complex. The inherent delays in ASGD amplify the biases introduced by data heterogeneity. This results in gradient estimates that deviate further from the true gradient, complicating convergence.
> > > > >
> > > > > Thank you for the comment. We have made clearer explanations in Lines 114-124 in our revised manuscript.

---

> > > ### Comment · Reviewer_f1eX · 2024-11-20
> > >
> > > The authors appear to have misunderstood the mechanism behind Malenia SGD. The fixed computation model does *not* imply uniformity in computations across workers—processing times can be arbitrarily heterogeneous, similar to IA2SGD. Regarding the delays, Malenia SGD is specifically designed to avoid the introduction of delays. Its mechanism ensures that parameter updates are never performed using stale model states, effectively eliminating delays altogether. Consequently, Malenia SGD supports arbitrary heterogeneity in worker computations while maintaining optimal performance, a property IA2SGD likely does not share.

---

> > > > ### Author Response · Authors · 2024-11-20
> > > >
> > > > Thank you for your response. The word "uniform" in our response does not mean "identical" computation times. Malenia SGD of course works on heterogeneous systems with diverse computation times, which we do not misunderstood. What we wanted to express is that the "fixed-computation-model" assumes that the computation time of each worker remain constant throughout the training process. We acknowledge the strength of Melania SGD under this model.

---

### Official Review · Reviewer_a6To · 2024-11-01

**Soundness:** 3
**Presentation:** 3
**Contribution:** 2
**Rating:** 5
**Confidence:** 3

**Summary:**

This paper proposes Incremental Aggregated Asynchronous SGD, an algorithm that improves the performance of asynchronous SGD on heterogeneous environments. With a server-side buffer, the proposed algorithm can neutralize the adverse effects of data heterogeneity. The theoretical analysis show that the proposed algorithm achieves a consistent convergence rate for smooth nonconvex problems for arbitrarily heterogeneous data. The experiments show that the proposed algorithm has good performance.

**Strengths:**

1. This paper proposes Incremental Aggregated Asynchronous SGD, an algorithm that improves the performance of asynchronous SGD on heterogeneous environments. With a server-side buffer, the proposed algorithm can neutralize the adverse effects of data heterogeneity.

2.The theoretical analysis show that the proposed algorithm achieves a consistent convergence rate for smooth nonconvex problems for arbitrarily heterogeneous data.

3. The experiments show that the proposed algorithm has good performance.

**Weaknesses:**

I have 2 major concerns:

1. In overall, the proposed algorithm is very similar to SAGA (the one cited in Appendix A, additional related works). I would actually say that the proposed algorithm is indeed SAGA, except that: the participating worker is not picked in a uniformly random manner but with some bounded delay ($\tau_{max}$); and the new gradient itself could have some delay. With the assumption of bounded delay ($\tau_{max}$), the new settings do not really make too much trouble to convert the theoretical analysis of convergence from SAGA to IA2SGD. Thus, the overall novelty of the proposed algorithm and the corresponding contribution of the theoretical analysis is limited.

2. The experiment results do not show significant improvement compared to some of the baselines. According to Figure 3, in all cases IA2SGD performs same as FedBuff (at least I could not see an obvious gap at the end of training), and in some cases IA2SGD performs similar to vanilla ASGD. Thus, the empirical contribution of the proposed algorithm is weakened.

**Questions:**

1. Could the authors explain in details about the difference between the proposed algorithm and SAGA, in both the algorithm itself and the theoretical analysis of convergence?

2. Could the authors justify the experiment results compared to FedBuff and Vanilla ASGD?

---

> ### Author Response · Authors · 2024-11-20
>
> > In overall, the proposed algorithm is very similar to SAGA (the one cited in Appendix A, additional related works). I would actually say that the proposed algorithm is indeed SAGA, except that: the participating worker is not picked in a uniformly random manner but with some bounded delay ($\tau_{\max}$); and the new gradient itself could have some delay. With the assumption of bounded delay ($\tau_{\max}$), the new settings do not really make too much trouble to convert the theoretical analysis of convergence from SAGA to IA$^2$SGD. Thus, the overall novelty of the proposed algorithm and the corresponding contribution of the theoretical analysis is limited.
> >
>
> **Response:**
>
> Thank you for your comments. While both algorithms address optimization problems of the form $\min_{\boldsymbol{w}\in\mathbb{R}^d} F(\boldsymbol{w}) = \frac{1}{n}\sum_{i=1}^{n}F_i(\boldsymbol{w})$, IA$^2$SGD differs substantially from SAGA both algorithmically and theoretically.
>
> Algorithmic Differences.
>
> 1. *Worker Selection:*
>     - SAGA samples worker $j_t$ is **uniformly** from $\{1,\dots,n\}$.
>     - IA$^2$SGD allows $j_t$ to be **any** worker that finishes its computations.
> 2. *Gradient Computation:*
>     - SAGA utilizes the **true** gradient $\nabla F_{j_t}(\boldsymbol{w}^{t-1})$ evaluated on the **latest** model $\boldsymbol{w}^{t-1}$ in each iteration.
>     - IA$^2$SGD employs **stochastic** gradient $\nabla f_{j_t} ( \boldsymbol{w}^{t- \tau _ {j_t} (t)}; \boldsymbol{\xi} _ {j_t}^{t} )$ evaluated on a **delayed** model $\boldsymbol{w}^{t-\tau_{j_t}(t)}$, introducing an inherent challenge due to the asynchrony and stochasticity.
> 3. *Synchrony:*
>     - SAGA operates **synchronously**, typically on a single machine.
>     - IA$^2$SGD is designed for **asynchronous** environments, accommodating delays that are crucial in distributed systems.
>
> Theoretical Differences. The theoretical analysis of IA$^2$SGD diverges significantly from SAGA due to the non-uniform sampling and the presence of delays.
>
> 1. *Unbiased vs. Biased Estimates:*
>     - In SAGA, the model update direction $\boldsymbol{g}_{\text{SAGA}}^t$ provides an **unbiased** estimate of $\nabla F(\boldsymbol{w}^{t-1})$, greatly simplifying the convergence analysis.
>     - IA$^2$SGD works with a **biased** estimate due to asynchronous updates, requiring intricate handling of delayed stochastic gradients, as demonstrated in Proposition 1.
> 2. *Handling Delays in Analysis:* Our analysis introduces novel decomposition techniques to manage the bias caused by delays (see equation (10)). This approach addresses complexities that are not present in SAGA’s synchronous and unbiased setting.

---

> > ### Author Response · Authors · 2024-11-20
> >
> > > The experiment results do not show significant improvement compared to some of the baselines. According to Figure 3, in all cases IA$^2$SGD performs same as FedBuff (at least I could not see an obvious gap at the end of training), and in some cases IA$^2$SGD performs similar to vanilla ASGD. Thus, the empirical contribution of the proposed algorithm is weakened.
> > >
> >
> > **Response:**
> > Thank you for your observations regarding the performance comparison in Figure 3. While all algorithms converge to similar levels of loss and accuracy given enough iterations, the key differentiator and contribution of our work lies in the **rate of convergence** and **robustness** under varying conditions of data heterogeneity.
> >
> > Detailed Observations from Experimental Results.
> >
> > 1. *High Data Heterogeneity ($\alpha=0.1$):*
> >     - In the regimes with high data heterogeneity, IA$^2$SGD demonstrates a faster rate of convergence in both training loss and test accuracy compared to all baselines, including FedBuff. This is evident in the 1st and 2nd columns of Figure 3. The faster convergence of IA$^2$SGD in these conditions is a significant empirical finding as it suggests better efficiency in more challenging and realistic distributed environments.
> > 2. *Low Data Heterogeneity ($\alpha=0.5$):*
> >     - In scenarios with lower data heterogeneity, IA$^2$SGD performs comparably to vanilla ASGD, and remains competitive or superior to FedBuff. This is shown in the 3rd and 4th columns of Figure 3. The performance of vanilla ASGD in this setting can be interpreted by its theoretical convergence rate of $\mathcal{O}\left( \sqrt{\frac{\sigma^2}{T}} + \frac{n}{T} + \zeta_{\max}^2 \right)$ noted in [1], where the adverse effect of the $\zeta^2_{\max}$ term becomes negligible under low heterogeneity. This highlights the robustness of IA$^2$SGD across both high and low heterogeneity scenarios.
> >
> > Our findings validate the effectiveness of IA$^2$SGD in managing the intrinsic challenges of asynchronous updates and data diversity. Thus, while the end-of-training performance may appear similar across algorithms, the journey to that endpoint—especially under challenging conditions—highlights the practical value and contribution of IA$^2$SGD.
> >
> > [1] Konstantin Mishchenko, Francis Bach, Mathieu Even, and Blake E Woodworth. Asynchronous SGD beats minibatch SGD under arbitrary delays. In Advances in Neural Information Processing Systems 35, pp. 420–433, 2022.

---

> ### Comment · Reviewer_a6To · 2024-11-22
> **the authors' response about the experiments makes sense. thanks for clarification**
>
> the authors' response about the experiments makes sense. thanks for clarification

---

> ### Comment · Reviewer_a6To · 2024-11-22
> **about the theoretical analysis**
>
> for the novelty of the algorithm and theoretical analysis, I do understand there are some differences between IA2SGD and SAGA.
>
> However:
>
> 1. Adding stochastic gradient to SAGA is easy. We basically just need to insert the variance term in the error bound.
>
> 2. Adding delay gradient is also not that difficult. Given the assumption of $L$-smoothness, we could always setup the upper-bound like (please forgive my typos and simplicity here) $|| \nabla F(w^{t}) - \nabla F(w^{t-\tau}) ||^2 \leq L^2 || w^{t} -  w^{t-\tau})||^2 = L^2 || \sum_{i \in [\tau]} -\eta \nabla f(w^{t-i}) ||^2 \leq \eta^2 L^2 \tau \sum_{i \in [\tau]}  || \nabla f(w^{t-i}) ||^2$, since there is a factor of $\eta^2$ where $\eta$ is the learning rate, with the diminishing or small enough learning rate, I think we should be able to make this term diminish (converge to 0 or absorbed by some other gradient square terms) or at least get upper-bounded. Then we could just insert this upper-bound in the theoretical analysis of some SAGA-like algorithm, which seems to be a common practice to me.
>
> I'm not saying that the proposed algorithm and the corresponding theoretical analysis has no novelty at all, but just the novelty and contribution seems limited or weak to me.

---

> ### Author Response · Authors · 2024-11-23
>
> Many thanks for your response.
>
> The high-level idea highlighted by the reviewer is a standard intermediate step in analyzing most algorithms that involve delays, such as Stich and Karimireddy (2020, Lemma 27), Koloskova el al. (2022, Lemmas 16 & 17), Islamov el al. (2024, Theorem 1), and ours. However, the actual theoretical analyses of our paper and all these cited references, incorporate unique complexities that are not immediately apparent.
>
> For instance, obtaining a bound on the iterate difference $||w - w^{t-\tau}||^2$ while maintaining the scaling factor $\frac{1}{n}$ ahead of the gradient noise $\sigma^2$, as detailed in Lemma 2, requires careful treatment of the conditional expectations inherent in our method. This may not be as easy as the reviewer suggests. Indeed, the overall analysis of our algorithm requires meticulous management of the errors introduced by delayed gradients and delayed iterates, which diverge substantially from the succinct two-page proof of nonconvex SAGA (Reddi et al., 2016). Thus, it could be an oversimplification to perceive these technical challenges as common.
>
> Conceptually, SAGA behaves more like the IAG method (Blatt el al., 2007) (see our Appendix A for a discussion). Despite their algorithmic similarity, the convergence rate of IAG under strong convexity were not resolved until 2017 (Gurbuzbalaban et al., 2017), well after the introduction of SAGA in 2014 (Defazio et al., 2014). This timeline may indicate that integrating delays into algorithms is not that trivial, otherwise the rich body of literature on asynchronous optimization could be easily extended.
>
> Given these explanations and the referenced literature, we kindly ask the reviewer to reconsider the novelty and technical contributions of our work. Our theoretical advancements in handling arbitrarily heterogeneous data in ASGD have not been addressed by previous algorithms like SAGA.
>
> Reference:
> - Sebastian U Stich and Sai Praneeth Karimireddy. The error-feedback framework: SGD with delayed gradients. Journal of Machine Learning Research, 21(237):1–36, 2020.
> - Anastasiia Koloskova, Sebastian U Stich, and Martin Jaggi. Sharper convergence guarantees for asynchronous SGD for distributed and federated learning. In Advances in Neural Information Processing Systems 35, pp. 17202–17215, 2022.
> - Rustem Islamov, Mher Safaryan, and Dan Alistarh. AsGrad: A sharp unified analysis of asynchronous-SGD algorithms. In Proceedings of the 27th International Conference on Artificial Intelligence and Statistics, pp. 649–657. PMLR, 2024.
> - Reddi, Sashank J., et al. Fast incremental method for nonconvex optimization. arXiv preprint arXiv:1603.06159 (2016).
> - Doron Blatt, Alfred O Hero, and Hillel Gauchman. A convergent incremental gradient method with a constant step size. SIAM Journal on Optimization, 18(1):29–51, 2007.
> - Mert Gurbuzbalaban, Asuman Ozdaglar, and Pablo A Parrilo. On the convergence rate of incremental aggregated gradient algorithms. SIAM Journal on Optimization, 27(2):1035–1048, 2017.
> - Aaron Defazio, Francis Bach, and Simon Lacoste-Julien. SAGA: A fast incremental gradient method with support for non-strongly convex composite objectives. In Advances in Neural Information Processing Systems 27, 2014.

---

> > ### Comment · Reviewer_a6To · 2024-11-24
> > **Thanks for the further clarification**
> >
> > Thanks for the further clarification. After reading the proof of Lemma 2, I do agree it's different from the error bound used SAGA paper or other asynchronous SGD papers. However, the "incorporate unique complexities" and "meticulous management of the errors" you mentioned above seems to be the carefully chosen plus-minus terms inserted in the inequalities. Honestly, I do understand that this kind of proof techniques requires effort and elaboration because I myself write this kind of proofs all the time. However, whether this is really difficult and complex is kind of subjective, and in overall its contribution seems kind of incremental to me.
> >
> > To be frank, the algorithm and the theoretical analysis together seems to be borderline, and I'm struggling between score 5 and 6.
> >
> > I will make a decision whether to raise the score after reading the other reviews and discussion with other reviewers.

---

### Official Review · Reviewer_TPj7 · 2024-11-01

**Soundness:** 3
**Presentation:** 3
**Contribution:** 2
**Rating:** 6
**Confidence:** 4

**Summary:**

This paper borrows the idea from IGD and proposes an asynchronous algorithm for distributed optimization accompanied by the convergence guarantee under non-convex settings. This algorithm is featured by using historical gradient information.

**Strengths:**

This paper borrows the idea from IGD and proposes an asynchronous algorithm for distributed optimization accompanied by the convergence guarantee under non-convex settings. This algorithm is featured by using historical gradient information.

The analysis doesn't rely on bounded data heterogeneity assumption.

The convergence rate enjoys a speedup in the number of clients.

**Weaknesses:**

The analysis is conservative in terms of the step size. According to my understanding, the proof follows the strategy of FedAvg, which restricts them to small step sizes.
From an optimization perspective, I was curious about the convergence rate under a deterministic setting. Based on my understanding of your proof framework, I think it cannot achieve a good rate under a deterministic setup. In addition, I think the learning rate used here in a deterministic setup will be less than the conventional distributed optimization.

**Questions:**

Can you show how the step size will look like when you reduce your analysis to a deterministic setup? Can you compare the step size choice used in your work to the one used in the existing works? e.g., Freya PAGE: First Optimal Time Complexity for Large-Scale Nonconvex Finite-Sum Optimization with Heterogeneous Asynchronous Computations.

---

> ### Author Response · Authors · 2024-11-20
>
> > The analysis is conservative in terms of the step size. According to my understanding, the proof follows the strategy of FedAvg, which restricts them to small step sizes.
> >
>
> **Response:**
>
> Thank you for your comments. Our analysis indeed differs significantly from that of FedAvg. While FedAvg is a synchronous parallel algorithm primarily challenged by its local update strategy, as noted in [1], our approach addresses different aspects. The main technical challenges in our proof stem from the sophisticated handling of delayed terms, which are not a factor in FedAvg's synchronous setup.
>
> For instance, when dealing with the inner product term $\mathbb{E} \left\langle \nabla F(\boldsymbol{w}^{t-1}), \frac{1}{nK} \sum_{i=1}^{n}\sum_{k=1}^{K} \nabla f(\boldsymbol{w}^{t-1};\xi_i^t) \right\rangle$ in the analysis of FedAvg, we can simply taking expectation relative to the random data samples by conditioning on $\boldsymbol{w}^{t-1}$, which yields $\mathbb{E} \left\langle \nabla F(\boldsymbol{w}^{t-1}), \boldsymbol{g}^t \right\rangle = \mathbb{E} \left\langle \nabla F(\boldsymbol{w}^{t-1}), \frac{1}{nK} \sum_{i=1}^{n}\sum_{k=1}^{K} \nabla F(\boldsymbol{w}^{t-1}) \right\rangle$. However, such a nice property does not apply to the analysis of IA$^2$SGD due to the delays of the models and data samples. Thus, we introduce a novel decomposition technique, as described in Lines 324-341, to manage the expectation of the inner product.
>
> [1] Sai Praneeth Karimireddy, Satyen Kale, Mehryar Mohri, Sashank Reddi, Sebastian Stich, and Ananda Theertha Suresh. Scaffold: Stochastic controlled averaging for federated learning. In Proceedings of the 37th International Conference on Machine Learning, pp. 5132–5143. PMLR, 2020.
>
> > From an optimization perspective, I was curious about the convergence rate under a deterministic setting. Based on my understanding of your proof framework, I think it cannot achieve a good rate under a deterministic setup. In addition, I think the learning rate used here in a deterministic setup will be less than the conventional distributed optimization.
> >
>
> **Response:**
>
> Our theoretical framework does indeed cover the deterministic setting (i.e., $\sigma^2=0$) as a special case. We demonstrate a unified convergence bound for both deterministic and stochastic settings in Appendix B.3:
> $$
> \frac{1}{T} \sum_{t=1}^{T} \mathbb{E} || \nabla F (\boldsymbol{w}^{t-1}) ||_2^2 \leq \frac{32 (F(\boldsymbol{w}^0) - F^*)}{T\eta} + 128 L \tau _{\max} \eta \frac{\sigma^2}{n} + 128 L^3 \tau _{\max}^2 \eta^3 \frac{\sigma^2}{n}.
> $$
>
> By setting $\sigma^2 = 0$ and choosing $\eta = \frac{1}{64L\tau_{\max}}$, inequality (35) simplifies to:
>
> $$
> \frac{1}{T}\sum_{t=1}^{T} \mathbb{E}[\|\nabla F(\boldsymbol{w}^{t-1}\|_2^2)] \leq \mathcal{O}\left(\frac{(F(\boldsymbol{w}_0)- F^*)L \tau _{\max}}{T}\right).
> $$
>
> This rate surpasses the rate of $\mathcal{O} \left( \sqrt{\frac{(1+\sigma^2)\tau_{\max}}{nT}} \right)$ achieved by IA$^2$SGD in the stochastic setting. Furthermore, if $\tau_{\max} = \mathcal{O}(n)$, then the sample complexity of this deterministic algorithm to reach an $\epsilon$-stationary point is $\mathcal{O}(n/\epsilon)$, which aligns with that of the nonconvex gradient descent (see, e.g., [2]).
>
> We appreciate your insight on this matter and have included a more detailed discussion in Remark 2 of our revised manuscript.
>
> [2] Sashank J. Reddi, Suvrit Sra, Barnabas Poczos, Alex Smola. Fast Incremental Method for Nonconvex Optimization, 2016.
>
> > Can you compare the step size choice used in your work to the one used in the existing works? e.g., Freya PAGE: First Optimal Time Complexity for Large-Scale Nonconvex Finite-Sum Optimization with Heterogeneous Asynchronous Computations.
> >
>
> **Response:**
>
> Thank you for your inquiry. In the Freya PAGE approach, the step size is defined as $\gamma = \left(L_-+L_{\pm}\sqrt{\frac{1-p}{pS}}\right)^{-1}$, This formulation is contingent upon the smoothness parameters $L_-,L_{\pm}$, as well as the probability $p$ of executing a particular subroutine. It is important to note that the Freya PAGE paper focuses on scenarios involving **homogeneous** data, meaning all workers share access to the same dataset. In contrast, our work addresses the complexities of a heterogeneous data environment, where different workers may have access to diverse subsets of data. This results in a step size formulation that may differ significantly from that used in Freya PAGE.

---

> ### Comment · Reviewer_TPj7 · 2024-11-21
>
> Thanks for your response. You can definitely recover deterministic case by setting the sgd variance to 0. What I want to say is that when it recovers to a deterministic case, can you enjoy a large step size. It seems not based on my understanding (In corollary 1, you need 1/(L \tau)). The paper I showed to you can. I think that's a shortcoming of your paper. Is it common to have the stepsize inversely proportional to the maximum of the delay?
>
> In addition, I don't think your analysis significantly differs from fedavg. The technique is quite similar from my perspective. The bounded delay ($\tau_{\max}$) plays a similar role as the number of local updates as in the analysis of FedAVG.
> You claim that you "introduce a novel decomposition technique", I agree with the author that's a difference. But from my perspective, I feel most of the proof in the appendix is similar.
>
> Anyway, I am still positive about this work. I appreciate the effort made by the authors.

---

> > ### Author Response · Authors · 2024-11-23
> >
> > Thanks a lot for your questions and positive feedback.
> >
> > It is well established that the step sizes of ASGD algorithms are often inversely proportional to delays, as evidenced by works such as Stich and Karimireddy (2020, Theorem 24), Koloskova et al. (2022, Lemma 19), Mishchenko el al. (2022, Theorem 4), and Islamov et al. (2024, Theorem 1).
> >
> > Our chosen step size of $\mathcal{O}\left(\frac{1}{L\tau_{\max}}\right)$ in the deterministic setting may seem conservative due to the presence of delays. However, when $\tau_{\max}$ scales with $n$, the sample complexity of $\mathcal{O}\left(\frac{n}{\epsilon}\right)$ aligns with that of full gradient descent, which uses a larger step size of $\mathcal{O}\left(\frac{1}{L}\right)$. This observation suggests that the conservative step size might not significantly impede the convergence rate when addressing deterministic problems.
> >
> > For the analysis, while there may be similarities in foundational techniques between FedAvg and ours (such as the critical use of the descent lemma for $L$-smooth functions), the key elements contributing to the final convergence rates differ significantly. The crux of the FedAvg analysis lies in bounding the client drift arising from local updates (Karimireddy et al., 2020, Lemma 8), which is orthogonal to our current approach. Our claim of introducing a novel decomposition technique is aimed at highlighting our unique method for managing the nuanced conditional expectations. This feature, which is not present in FedAvg, is essential to our analysis and has not been rigorously addressed in some previous works (see footnote 1 in our manuscript).
> >
> > We appreciate the reviewer's comments and hope that our clarification addresses them effectively.
> >
> >
> > References:
> > - Sebastian U Stich and Sai Praneeth Karimireddy. The error-feedback framework: SGD with delayed gradients. Journal of Machine Learning Research, 21(237):1–36, 2020.
> > - Anastasiia Koloskova, Sebastian U Stich, and Martin Jaggi. Sharper convergence guarantees for asynchronous SGD for distributed and federated learning. In Advances in Neural Information Processing Systems 35, pp. 17202–17215, 2022.
> > - Konstantin Mishchenko, Francis Bach, Mathieu Even, and Blake E Woodworth. Asynchronous SGD beats minibatch SGD under arbitrary delays. In Advances in Neural Information Processing Systems 35, pp. 420–433, 2022.
> > - Rustem Islamov, Mher Safaryan, and Dan Alistarh. AsGrad: A sharp unified analysis of asynchronous-SGD algorithms. In Proceedings of the 27th International Conference on Artificial Intelligence and Statistics, pp. 649–657. PMLR, 2024.
> > - Karimireddy, Sai Praneeth, et al. "Scaffold: Stochastic controlled averaging for federated learning." International conference on machine learning. PMLR, 2020.

---

### Meta-Review · Area_Chair_MQBF · 2024-12-17

**Metareview:**

This paper proposes a novel asynchronous algorithm, IA²-SGD, for solving distributed optimization problems. While naïve implementations of asynchronous SGD are often affected by data heterogeneity, the proposed algorithm incorporates variance-reduction techniques, similar to SAGA, to mitigate these adverse effects. The authors provide a convergence bound for non-convex smooth functions to support their approach.

One reviewer noted similarities between the algorithm design and the (centralized) SAGA method but acknowledged key differences in methodology and the convergence proofs. Additionally, two reviewers raised concerns about the related work "Melania SGD," designed for heterogeneous hardware. The authors convincingly addressed these concerns in their rebuttal, clarifying the differences and including Melania SGD in their empirical comparisons.

The paper is in the borderline range even when considering an adjustment of the reviewer scores after the rebuttal (considering an increase for reviewer f1eX  - as some of their concerns seem addressed). However, two key concerns remain. First, the worst-case guarantee provided does not demonstrate uniform improvement over prior methods in all settings, requiring further discussion or evaluation. Second, the numerical evaluation is insufficient to fully demonstrate the strengths of the proposed method across diverse heterogeneous settings.

Given these considerations, the paper remains borderline, with room for improvement in theoretical discussion and empirical validation.

**Additional Comments On Reviewer Discussion:**

Two reviewers raised the related work "Melania SGD," designed for heterogeneous hardware. The authors addressed these concerns in their rebuttal, clarifying the differences and including Melania SGD in their empirical comparisons. While reviewer f1eX maintained their score after the rebuttal, I found the concerns regarding this aspect convincingly and sufficiently addressed.

---

### Decision · Program_Chairs · 2025-01-22

Reject